# When to Update Your Model: Constrained Model-based Reinforcement Learning

**Tianying Ji**[1], **Yu Luo**[1], **Fuchun Sun**[*,1], **Mingxuan Jing**[2], **Fengxiang He**[3], **Wenbing Huang**[4,5]

[1] Department of Computer Science and Technology, Tsinghua University
[2] Science & Technology on Integrated Information System Laboratory,
Institute of Software Chinese Academy of Sciences
[3] JD Explore Academy, JD.com Inc
[4] Gaoling School of Artificial Intelligence, Renmin University of China
[5] Beijing Key Laboratory of Big Data Management and Analysis Methods, Beijing, China
`{jity20, luoyu19}@mails.tsinghua.edu.cn; fcsun@tsinghua.edu.cn;`
`jingmingxuan@iscas.ac.cn; fengxiang.f.he@gmail.com; hwenbing@126.com`

## Abstract

Designing and analyzing model-based RL (MBRL) algorithms with guaranteed monotonic improvement has been challenging, mainly due to the interdependence between policy optimization and model learning. Existing discrepancy bounds generally ignore the impacts of model shifts, and their corresponding algorithms are prone to degrade performance by drastic model updating. In this work, we first propose a novel and general theoretical scheme for a non-decreasing performance guarantee of MBRL. Our follow-up derived bounds reveal the relationship between model shifts and performance improvement. These discoveries encourage us to formulate a constrained lower-bound optimization problem to permit the monotonicity of MBRL. A further example demonstrates that learning models from a dynamically-varying number of explorations benefit the eventual returns. Motivated by these analyses, we design a simple but effective algorithm CMLO (Constrained Model-shift Lower-bound Optimization), by introducing an event-triggered mechanism that flexibly determines when to update the model. Experiments show that CMLO surpasses other state-of-the-art methods and produces a boost when various policy optimization methods are employed.

## 1 Introduction

Reinforcement learning (RL) has driven impressive advances in many complex decision-making problems in recent years [36, 48]. Many of these advances are obtained by model-free (MFRL) methods, whose desirable asymptotic performance yet comes at the cost of sample efficiency. Hence their applications are mostly limited to simulation scenarios [45, 33, 35]. In contrast, Model-Based RL (MBRL) methods, which learn a transition model directly from orders-of-magnitude fewer samples and then derive the optimal policy from the learned model, have become an appealing alternative in small-data and more practical cases [37, 9, 41, 18].

In general, MBRL methods alternate between the two stages: model learning and policy optimization (*e.g.* the general Dyna-style [50, 51]). A more accurate model will lead to a better policy. Various attempts have been proposed to improve model accuracy by investigating high-capacity models (the model ensemble technique [27, 8] and better function approximators [15, 38]) or amending the policy optimization stage based on model bias [24, 39, 28, 20, 7, 57]. Even so, their resultant models

---

[*]Corresponding authors: Fuchun Sun.

36th Conference on Neural Information Processing Systems (NeurIPS 2022).

are just accurate in a local and relative sense, since the learning is conditional on a fixed number of state-action tuples explored by the policy at the current step, other than the full transition dynamics of the environment. Indeed, it is tricky to determine how much we should explore. Insufficient exploration would trap the model and the following policy optimization, whereas excessive newly-encountered state-action pairs would confuse the model and later cause policy chattering. To derive a "truly" accurate model, we need a smarter scheme to choose different numbers of explorations at different times, instead of the unchanged setting in current methods.

From an optimization point of view, the thinking of how to derive an accurate model for MBRL in a global sense also motivates us to investigate the monotonicity of the optimization target (*i.e.* the return of the learned model and policy in MBRL), which, unfortunately, is less explored and not well guaranteed in current works. However, discussing the monotonicity guarantee for MBRL is challenging by any means, arising from the coupling of the model learning and policy optimization processes. Although there has been recent interest in related subjects, most of the theoretical works seek to characterize the monotonicity in terms of a fixed model of interest [49, 34, 20, 12, 28], which does not naturally fit our case when the model is dynamically shifted.

In this paper, we study how to guarantee the optimization monotonicity theoretically, upon which we then develop an event-triggered strategy that learns the model from a dynamically-varying number of explorations. In particular, we interestingly find that the lower bound of the performance improvement between two adjacent alternation steps in MBRL is dependent on the one-step model accuracy plus the constraint of the model shifts under certain mild assumptions. This discovery encourages us to formulate a constrained optimization problem, in order to permit positive performance improvement and thus the optimization monotonicity for MBRL. We also give a feasible solution example to show that dynamical alternation between model learning and policy exploration does benefit performance monotonicity. To resolve the constrained optimization problem, we design a simple but effective algorithm CMLO (Constrained Model-shift Lower-bound) equipped with an event-triggered mechanism. This mechanism first estimates whether the model shifts meet the constraint and then decides when to train the model.

We evaluate CMLO on several continuous control benchmark tasks. The results show that CMLO learns much faster than other state-of-the-art MBRL methods and yields promising asymptotic performance compared with the model-free counterparts. Note that our optimization framework is general and can be applied to different backbones of policy optimization algorithms, which is also ablated in our experiments.

## 2 Related works

Model-based reinforcement learning methods have shown great potential for sequential decision-making both in simulation and in the real world due to their sample efficiency [9, 21]. Generally, these MBRL algorithms can be grouped into several categories to highlight the range of uses of predictive models [55]. And our work falls into the Dyna-style category. In Dyna-style algorithms, training alternates between model learning under policy iterations with the real environments, and policy optimization using the model rollouts [50–52, 13]. Many attempts have been devoted to improving these two stages.

For model learning stage, previous main concerns are function approximators and training objectives. The dynamics approximator has advanced from Gaussian processes [26, 9], time-varying linear dynamics [29, 30] to neural network predictive models [15, 38]. And training objectives vary from Mean Square Error (MSE) [38, 34], Negative Log Likelihood (NLL) [8, 20], *etc*. Moreover, the deep ensemble technique is appealing for improving the robustness to model error. Our method adopts an ensemble of probabilistic networks similarly as in [8, 20].

The policy optimization stage allows Dyna-style algorithms to leverage various off-the-shelf model-free methods, such as SAC [16], TRPO [44], and TD3 [14]. Much owing to the progress of model-free methods, our method is to invoke any reasonable optimization oracle for the empirical models, rather than entangle a particular policy optimization algorithm.

A consensus of MBRL is that a smart policy requires an accurate model. However, model bias cannot be eliminated because the state-action distribution of the samples in the model learning stage and policy optimization stage is quite different. Many prior works attend to this distribution mismatch

issue and then tailor the data used for policy optimization according to the model bias. Janner et al. [20], Buckman et al. [7] encourage truncated rollout lengths. Besides, the ratio of real to model-generated data can be dynamically tuned according to the model uncertainty [24, 39, 28]. Yu et al. [57] penalizes rewards by the model uncertainty. Our method incorporates the truncated model rollouts mechanism. Moreover, we further explore how real interactions affect overall performance which is rarely studied before. We construct an event-triggered mechanism to cope with overfitting in a small-data regime and suffering generalization error when facing a drastic distribution shift.

Monotonic improvement guarantee has been a fundamental concern in both model-free and model-based avenues. In MFRL methods, both CPI [22] and TRPO [44] can be understood as approximating and optimizing the performance gap by forcing the new policy to be not too far away from the current policy. However, in Model-based settings, their trust-region constraints cannot directly be satisfied because the policies highly depend on the randomness of the models. While constructing such a bound for performance gap is straightforward, it has not been explored in previous MBRL theoretical analyses, instead they [49, 34, 20, 12, 28] turn to bound the discrepancy between returns under a model and those in the real environment. Although they could guarantee that the lower bound of policy performance improves under a certain model, this guarantee may face several issues regarding model updating. In contrast, we construct the performance difference scheme for MBRL algorithms and perform monotonicity analysis under this scheme.

Another line of theoretical works focus on regret bounds [10, 23] or sample complexity properties [1, 5, 47], focusing on the convergence performance and sample complexity for model-based approaches.

## 3 Preliminaries

**Markov Decision Process** A discounted Markov Decision Process (MDP) is a quintuple $M = (\mathcal{S}, \mathcal{A}, P_M, r_M, \gamma)$, where $\mathcal{S}$ is the state space, $\mathcal{A}$ represents the action space, $P_M$ denotes the transition function, $r : \mathcal{S} \times \mathcal{A} \to [-R, R]$ stands for the reward function, and $\gamma \in (0, 1)$ is the discount factor. For a fixed policy $\pi$ and model $M$, we define $V_M^\pi(\mu)$ as the return of the model $M$ with the starting state distribution $\mu$, and $V^\pi(\mu)$ denotes the returns under the real environment,

$$V_M^\pi(\mu) = \mathbb{E}_{\substack{a_t \sim \pi(\cdot|s_t) \\ s_{t+1} \sim P_M(\cdot|s_t, a_t)}} \Big[ \sum_{t=0}^\infty \gamma^t r_M(s_t, a_t) | \pi, s_0 \Big]. \tag{3.1}$$

We make a mild assumption that the model $M$ to be identical to the real MDP except the transition function. Let $d_{M_i}^{\pi_k}(s, a; \mu)$ denote the visitation probability $s, a$ when starting at $s_0 \sim \mu$ and following $\pi_k$ under the dynamics $P_{M_i}$. We will omit it as $d_{M_i}^{\pi_k}$ henceforth for brevity. Besides, we denote $\mathcal{M}$ as a (parameterized) family of models of interest, and let $\Pi$ be a family of policies.

**Generative Model** Many previous works [31, 25, 1] focus on a stylized generative model. By assuming an access to the generative model, we collect $N$ samples for each state-action pair $(s, a) \in \mathcal{S} \times \mathcal{A}$: $s_{s,a}^i \overset{i.i.d}{\sim} P(\cdot|s, a)$ which allows us to construct an empirical model defined as follows:

$$\forall s' \in \mathcal{S}, \quad \hat{P}(s'|s, a) = \frac{1}{N} \sum_{i=1}^N \mathbb{1}\{s_{s,a}^i = s'\}. \tag{3.2}$$

where $\mathbb{1}\{\cdot\}$ is the indicator function. This leads to an empirical MDP $\hat{M} = (\mathcal{S}, \mathcal{A}, \hat{P}, r, \gamma)$.

## 4 Monotonic improvement under model shifts

This section provides theoretical analyses for monotonic improvement of MBRL, factoring in the interdependence between policies and models. We first construct a general scheme for a non-decreasing performance guarantee and follow it up by characterizing the lower bound when shifting the model. Towards a non-negative lower bound, we restrict the model shifts and then obtain a refined bound. These discoveries encourage us to translate the bound maximization to a constrained optimization problem to permit monotonicity. By deriving an instance solution under the generative model setting, we demonstrate the merits of the dynamic model learning interval.

## 4.1 Monotonic improvement with policy optimization oracle

Our goal is to construct a general recipe for a monotonicity guarantee. Naturally, we seek to build a performance difference scheme for model-based algorithms.

**Definition 4.1** (**Performance Difference Bound Scheme**). *$V^{\pi_i|M_i}(\mu)$ denotes the return of the policy $\pi_i \in \Pi$ in the real environment, whereas $\pi_i$ is derived from the dynamical model $M_i \in \mathcal{M}$. Then, the lower bound on the true return gap of $\pi_1$ and $\pi_2$ can be stated in the form,*

$$V^{\pi_2|M_2}(\mu) - V^{\pi_1|M_1}(\mu) \geq C. \tag{4.1}$$

*Such a statement guarantees that the policy allows non-decreasing performance in the real environment when $C$ is non-negative.*

Although there has been interest in non-decreasing performance guarantee, previous works [34, 20] commonly derive under a "discrepancy bound" scheme disregarding the model shifts (*i.e.* $M_1 = M_2$). Their results imply that once a policy update $\pi_1 \rightarrow \pi_2$ increases the returns under the same model ($V_{M_1}^{\pi_2} > V_{M_1}^{\pi_1}$), the lower bound on the policy performance evaluated in the real environment improves accordingly, $\inf\{V^{\pi_2|M_1}\} > \inf\{V^{\pi_1|M_1}\}$.

When the model shift is introduced, establishing the performance difference bound turns out to be rather difficult, mainly due to the coupling of the policy optimization and model learning: the estimated model is generated from the policy explorations, while the policy derives from the model rollouts. Hence, we need to consider the performance gap arising from the integration of the two stages, which, unfortunately, has never been explored before. Since the performance of the policy with a fixed model is already well guaranteed, it is natural to make the following assumptions.

**Assumption 4.2** (**Policy Optimization Oracle**). *The policy optimization oracle is defined as the one that takes as input a model $M$ and returns a $\epsilon_{opt}$-optimal policy $\pi$ satisfying: $V_M^*(\mu) - \epsilon_{opt} \leq V_M^\pi(\mu) \leq V_M^*(\mu)$. We assume that our policy optimization stage always meets the policy optimization oracle given its corresponding estimated model $M$.*

Such assumption usually holds in practice when we explore existing off-the-shelf model-free algorithms [43, 1, 16, 44]. With this assumption, we can focus directly on the eventual performance difference under the real environment encountering the model updating.

The bound $C$ we seek can be expressed in terms of two kinds of gaps: the inconsistency gap between the model and the environment, and the optimal returns gap between the two models. When a policy $\pi_k$ samples in a model $M_i$, it will encounter states not consistent with those generated by the real environment. We denote this inconsistency by $\epsilon_{M_i}^{\pi_k} = \mathbb{E}_{s,a\sim d^{\pi_k}}[\mathcal{D}_{\mathrm{TV}}(P(\cdot|s,a)\|P_{M_i}(\cdot|s,a))]$. Besides, for each model $M_i \in \mathcal{M}$, a remarkable property is that there always exists a policy that maximizes the value function $V_{M_i}^\pi(\mu)$ [53]. Hence, we define the ceiling performance of model $M$ as $V_{M_i}^*(\mu) = \sup_{\pi \in \Pi} V_{M_i}^\pi(\mu)$. We will omit $\mu$ henceforth for simplicity unless confusion exists. With these two terms well-defined, we now present our bound.

**Theorem 4.3** (**Performance Difference Bound for Model-based RL**). *Let $M_i \in \mathcal{M}$ be the estimated models and $\pi_i$ be the $\epsilon_{opt}$-optimal policy for $M_i$. Recalling $\kappa = \frac{2R\gamma}{(1-\gamma)^2}$ where $R$ is the bound of the reward function, we have the performance difference of $\pi_1$ and $\pi_2$ evaluated under the real environment be bounded as below,*

$$V^{\pi_2|M_2} - V^{\pi_1|M_1} \geq \kappa \cdot (\epsilon_{M_1}^{\pi_1} - \epsilon_{M_2}^{\pi_2}) + V_{M_2}^* - V_{M_1}^* - \epsilon_{opt}. \tag{4.2}$$

*Proof.* See Appendix B, Theorem B.1. $\qquad\qquad\qquad\qquad\qquad\qquad\qquad\qquad\qquad\qquad\qquad\square$

This bound implies that if the model update $M_1 \rightarrow M_2$ can (1) shorten the divergence between the estimated dynamics and the true dynamics and (2) improve the ceiling performance on the model, it may guarantee overall performance improvement under the true dynamics.

## 4.2 Lower-bound optimization with model shift constraints

Theorem 4.3 provides a general performance difference bound that suggests models with higher ceiling performance and lower bias would raise the overall performance. However, finding a non-negative lower bound may face several issues regarding a drastic model shift in practice. On the one

hand, performing an abrupt model update could potentially lead to a tumble in ceiling performance and then fail to make incremental improvements. On the other hand, huge distribution divergence between model rollouts would confound the policy optimization stage and hamper its access to optimal policies. Thus, we shoot for refining the bound upon adding model shifts constraint.

First, we seek to further unfold the ceiling performance gap, which serves as the main building block toward the desired bound. During derivation, we additionally introduce the $L$-Lipschitz assumption.

**Assumption 4.4** ($L$-**Lipschitzness of Value Function**). *We call a value function $V_M^\pi$ on the estimated dynamical model $M$ is $L$-Lipschitz w.r.t to some norm $\|\cdot\|$ in the sense that*

$$\forall s, s' \in \mathcal{S}, |V_M^\pi(s) - V_M^\pi(s')| \leq L \cdot |s - s'|. \tag{4.3}$$

We assume that our estimated model $M \in \mathcal{M}$ satisfies the Lipschiz character, inspired from [4, 34]. Under the $L$-Lipschitzness assumption, we can derive a bound for the ceiling return gap.

**Theorem 4.5** (**Ceiling Return Gap under Model Shift**). *For an estimated model $M_i \in \mathcal{M}$, the ceiling return gap is bounded as:*

$$V_{M_2}^* - V_{M_1}^* \geq -\frac{\gamma}{1-\gamma} L \cdot \sup_{\pi \in \Pi} \mathbb{E}_{s, a \sim d_{M_2}^\pi} \left[ |P_{M_2}(\cdot|s, a) - P_{M_1}(\cdot|s, a)| \right]. \tag{4.4}$$

*Proof.* See Appendix B, Theorem B.2. $\qquad\square$

This conclusion reveals the connection between the ceiling return gap and models' disagreement. In a benign scenario, the term of ceiling performance gap in the RHS of Eq. 4.2 should be dominated by the term $\kappa \cdot (\epsilon_{M_1}^{\pi_1} - \epsilon_{M_2}^{\pi_2})$ when the model shift is sufciently small. A sharp model shift, however, risks a massive reduction in ceiling performance that is hardly bridged by the other parts in Eq. 4.2, and therefore corrupts monotonicity. This inspires us to introduce the constraint of the model shift. We further refine our performance difference bound to better characterize the relationship between performance gap and model shift.

**Theorem 4.6** (**Refined Bound with Constraint**). *Let policy $\pi_i \in \Pi$ denotes the $\epsilon_{opt}$ optimal policy under the dynamical model $M_i \in \mathcal{M}$, and $\sigma_{M_1, M_2}$ be the constraint threshold for $M_1$ and $M_2$. Note that $L \geq \frac{R}{1-\gamma}$. Then we can refine performance difference lower bound under the model shifts constraint as,*

$$V^{\pi_2 | M_2} - V^{\pi_1 | M_1} \geq \kappa \cdot \left\{ \mathbb{E}_{s, a \sim d^{\pi_1}} \mathcal{D}_{\text{TV}} \left[ P(\cdot|s, a) \| P_{M_1}(\cdot|s, a) \right] \right.$$
$$\left. - \mathbb{E}_{s, a \sim d^{\pi_2}} \mathcal{D}_{\text{TV}} \left[ P(\cdot|s, a) \| P_{M_2}(\cdot|s, a) \right] \right\} - \frac{\gamma}{1-\gamma} L \cdot (2\sigma_{M_1, M_2}) - \epsilon_{opt}, \tag{4.5}$$

$$s.t. \quad \mathcal{D}_{\text{TV}}(P_{M_2}(\cdot|s, a) \| P_{M_1}(\cdot|s, a)) \leq \sigma_{M_1, M_2}, \quad \forall (s, a) \in \mathcal{S} \times \mathcal{A}. \tag{4.6}$$

*Proof.* See AppendixB, Theorem B.3. $\qquad\square$

Theorem 4.6 implies that policy $\pi_2$ is guaranteed to outperform policy $\pi_1$ once it makes the RHS in Eq. 4.5 greater than zero under the constraint Eq. 4.6. More specifically, to guarantee a non-decreasing performance, $M_2$ and $\pi_2$ should meet the following two requirements,

$$\mathbb{E}_{s, a \sim d^{\pi_2}} \left[ \sum_{s' \in \mathcal{S}} |P(s'|s, a) - P_{M_2}(s'|s, a)| \right] \leq 2\epsilon_{M_1}^{\pi_1} - \frac{(1-\gamma)L}{R}(2\sigma_{M_1, M_2}) - \frac{2}{\kappa} \cdot \epsilon_{opt}, \tag{R1}$$

$$\mathcal{D}_{\text{TV}}(P_{M_2}(\cdot|s, a) \| P_{M_1}(\cdot|s, a)) \leq \sigma_{M_1, M_2}, \quad \forall (s, a) \in \mathcal{S} \times \mathcal{A}. \tag{R2}$$

R1 encourages us to alleviate model bias as much as possible. Moreover, when the policy $\pi_1$ samples too many states not encountered by the model $M_1$, it may lead to excessive generalization errors during model learning, i.e. the estimated model will suffer extrapolation error in these unexperienced regions, which further causes instability or crashes in the following derived sub-optimal policies. Thus, the introduction of the R2 constraint can help solve the above problem. Finally, we abstract these two requirements to a constrained optimization problem.

**Proposition 4.7** (**Constrained Lower-Bound Optimization Problem**). *We reduce the issue of finding a non-negative $C$ to the following constrained optimization problem. Here, $\pi_i$ is still the suboptimal policy under model $M_i$. The minimal objective in E.q. 4.7 leads to the maximum of $C$. Then the overall optimization problem can be formalized as,*

$$\min_{\substack{M_2 \in \mathcal{M} \\ \pi_2 \in \Pi}} \mathbb{E}_{s,a \sim d^{\pi_2}} \Big[ \sum_{s' \in \mathcal{S}} |P(s'|s,a) - P_{M_2}(s'|s,a)| \Big],$$

$$s.t. \quad \sup_{s \in \mathcal{S}, a \in \mathcal{A}} \mathcal{D}_{\mathrm{TV}}(P_{M_1}(\cdot|s,a)\|P_{M_2}(\cdot|s,a)) \leq \sigma_{M_1,M_2}. \tag{4.7}$$

### 4.3 A feasible example for constrained optimization problem

We first remark that, Proposition 4.7 provides a useful guide for acquiring performance improvement through restricting the upcoming model shift into a safe zone. In this section, we give an instance for a feasible solution of the constrained optimization problem under the generative model setting [1, 31]. Specifically, we construct an empirical model $M_1$ from the N samples per state-action pair that stems from the generative model. Upon encountering another $k$ samples on each state-action pair, we segue into the model updating stage, which outputs $M_2$ given these newly collected samples as input. Towards obtaining a non-decreasing performance, we yield the following feasible solution for model training interval through satisfying requirements R1 and R2.

**Corollary 4.8.** *Here, $vol(\mathcal{S})$ denotes the volume of the state coverage simplex. For simplicity, we denote $\delta_{M_i}(\cdot|s,a) = |P(\cdot|s,a) - P_{M_i}(\cdot|s,a)|$ for each model $M_i \in \mathcal{M}$. Under the generative model setting, with a probability larger than $1 - \xi$, we can provide a non-negative $C$ when given,*

$$k = \frac{2}{\epsilon^2} \log \frac{2^{vol(\mathcal{S})} - 2}{\xi} - N. \tag{4.8}$$

*Here, $\epsilon = \delta_{M_1}(\cdot|s,a) - \frac{(1-\gamma)L}{R} \cdot (2\sigma_{M_1,M_2}) - \frac{(1-\gamma)^2}{R\gamma} \cdot \epsilon_{opt}$ and $\xi \in (0,1)$ is a constant.*

*Proof.* See Appendix B, Corollary B.4. □

One can deduce from Corollary 4.8 that dynamically adjusting the model training interval according to the model bias and the model shifts constraints threshold does benefit monotonicity. Along with the model bias $\delta_{M_1}(\cdot|s,a)$ decaying, the model training interval $k$ requires to be scaled up to obtain adequate newly encountered samples for training $M_2$. Besides, once gathering excessive samples, we risk violating the model shifts constraint, thus impairing performance. This instance supports the insight that determining "when to update your model" is vital for performance improvement and motivates a smarter scheme to choose different numbers of explorations at different times instead of the unchanged setting in current methods.

## 5 CMLO framework

It is nontrivial to tackle the proposed constrained optimization problem in Proposition 4.7, since one cannot directly assign a value to the optimization variable $M_2$. We decouple the optimization objective and the constraint as "how to train the model" and "when to train the model" through an event-triggered mechanism towards dynamic alternation.

**Objective minimization.** Minimizing the objective function involves improving the model accuracy. Specifically, we adopt the model-ensemble technique to reduce model bias. For practical implementation, the probabilistic models $\{\hat{f}_{\phi_1}, \hat{f}_{\phi_2}, \dots, \hat{f}_{\phi_K}\}$ are fitted on shared but differently shuffled replay buffer $\mathcal{D}_e$, and the target is to optimize the Negative Log Likelihood (NLL).

**Constraint estimation.** The unobserved model $M_2$ makes the constraint function incalculable, and here we seek to design an estimator for it. Recall that $\mathcal{D}_{\mathrm{TV}}(P_{M_1}(\cdot|s,a)\|P_{M_2}(\cdot|s,a)) = \frac{1}{2}\sum_{s' \in \mathcal{S}} |P_{M_1}(s'|s,a) - P_{M_2}(s'|s,a)|$, then we can find that the distance arises from two parts, the state space coverage, and the models' disagreement. We estimate the policy coverage (state-space coverage) by computing the volume $vol(\mathcal{S}_\mathcal{D})$ of the convex closure $\mathcal{S}_\mathcal{D} = \big\{ \sum_{s_i \in \mathcal{D}} \lambda_i S_i :$

$\lambda_i \geq 0, \sum_i \lambda_i = 1\}$ constructed on the replay buffer $\mathcal{D}$. We exploit the average prediction error on these new samples data to estimate the disagreement on newly encountered data $(\Delta\mathcal{D})$ and get $\mathcal{L}(\Delta\mathcal{D}) = \mathbb{E}_{(s,a,s') \in \Delta_{\mathcal{D}}} \left[ \frac{1}{K} \sum_{i=1}^{K} \|s' - \hat{f}_{\phi_i}(s, a)\| \right]$. Combining these two components, we can obtain an estimation for the model shift, $i.e.$, $vol(\mathcal{S}_\mathcal{D}) \cdot \mathcal{L}(\Delta\mathcal{D})$. This practical overestimation for the model shifts makes the constraint stay satisfied during objective optimization.

**Event-triggered mechanism.** We design an event-triggered mechanism to determine the occasion to pause collecting and turn to solve the optimization objective. Remark that although diverting to train models as long as not to violate the constraint is theoretically reasonable, we refrain from doing so in practice because performing an update on data with a minor shift in coverage and distribution is wasteful and may risk overfitting. Therefore, we trigger when the constraint boundary is touched to reduce computational cost and escape overfitting. The event-triggering mechanism is based on the following condition:

$$\frac{vol(\mathcal{S}_{\mathcal{D} \cup \Delta D(\tau)})}{vol(\mathcal{S}_\mathcal{D})} \cdot \mathcal{L}(\Delta D(\tau)) \geq \alpha. \tag{5.1}$$

Here, $\tau$ is the event-triggering time, and $\alpha$ is a given constant.

**Policy optimization oracle.** Clearly, we can leverage many model-free RL methods (SAC [16], TRPO [44], PPO [46] $etc.$) as our policy optimization oracle. Besides, we adopt a truncated short model rollouts technique to mitigate compounding error while encouraging model usage. Based on the fresh model rollouts, we perform the policy optimization oracle, employing SAC as an example.

**Algorithm Overview.** We briefly give an overview of our proposed CMLO in algorithm 1. Notably, the event-triggered mechanism subtly determines the occasion to perform model updating, promoting performance monotonicity and reducing computation load.

---

**Algorithm 1:** CMLO

---

**initialize :** policy $\pi_\theta$, ensemble models $\hat{f}_{\phi_1}, \hat{f}_{\phi_2}, \ldots, \hat{f}_{\phi_K}$, environment buffer $\mathcal{D}_e$ and model buffer $\mathcal{D}_m$

**repeat**

    Sample $\Delta\mathcal{D}_e \sim \pi_\theta$ from real environment; add to $\mathcal{D}_e$

    Estimate model shifts by $vol(\mathcal{S}_{D_e})\mathcal{L}(\Delta\mathcal{D}_e)$

    **if** *Event-triggered condition (E.3) is reached* **then**

        | Train all models $\hat{f}_{\phi_1}, \hat{f}_{\phi_2}, \ldots, \hat{f}_{\phi_K}$ on $\mathcal{D}_e$

    Perform $h$-step model rollouts using policy $\pi_\theta$; add to $\mathcal{D}_m$

    Update $\pi_\theta$ on $\mathcal{D}_m$ through SAC [16]

**until** *the policy performs well in the environment*;

---

## 6 Experiments

Our experimental evaluation aims to investigate the following questions: (1) How well does our algorithm perform on standard reinforcement learning benchmarks compared to prior state-of-the-art model-based and model-free algorithms? (2) Does the performance with or without the constraint consistent with previous theoretical analyses?

### 6.1 Comparative evaluation

To illustrate the effectiveness of our method, we contrast several popular model-based and model-free baselines. Model-free counterparts include: (1) SAC [16], the state-of-the-art in terms of asymptotic performance. (2) PPO [46] that explores monotonic improvement as well. Model-based baselines include: (3) PETS [8], which employs models directly for planning, different from the Dyna-style. (4) SLBO [34], that explores monotonicity under the discrepancy bound scheme. (5) MBPO [20], that employs a similar design of model ensemble technique (ensemble of probabilistic dynamics networks) and policy optimization oracle (SAC) as we do. (6) AutoMBPO [28], a variant of MBPO, that uses an automatic hyperparameter controller to tune the model-training frequency but suffers from high pre-training cost and lacks theoretical analysis on parameters rationality.

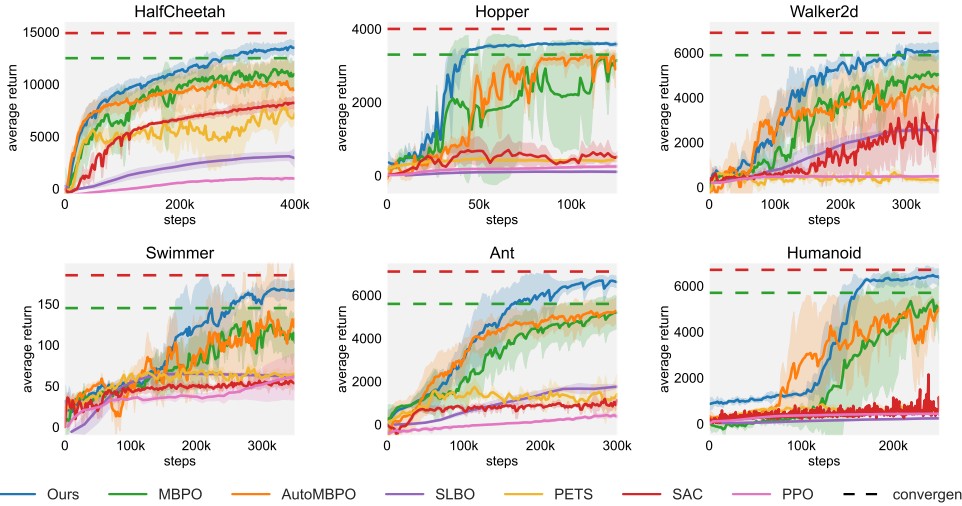

Figure 1: Comparison of learning performance on continuous control benchmarks. We evaluate each algorithm on the standard 1000-step versions. Solid curves indicate the average performance among seven trials under different random seeds, while the shade corresponds to the standard deviation over these trails. The dashed lines are the asymptotic performance of SAC (at 5M steps) and MBPO.

We evaluate CMLO and these baselines on six continuous control tasks in OpenAI Gym [6] with the MuJoCo [54] physics simulator, including HalfCheetah, Hopper, Walker2d, Swimmer, Ant, Humanoid. For fair comparison, we adopt the standard full-length version of these tasks and align the same environment settings.

Figure 1 shows the learning curves of all compared methods, along with the asymptotic performance. These results show that our algorithm is far ahead of the model-free method in terms of sample efficiency, coupled with an asymptotic performance comparable to that of the state-of-the-art model-free counterparts SAC. Compared to model-based baselines, our method gains faster convergence speed and better eventual performance. Notably, credit to the event-triggered mechanism, our method enjoys a more stable training curve. The better monotonic property of the learning curve agrees with our previous analyses.

## 6.2 Ablation studies

Next, we make ablations and modifications to our method to validate the effectiveness and generalizability of the mechanism we devised.

**The necessity of event-triggered mechanism.** To verify the necessity of event-triggered mechanism, we compare to three unconstrained cases (given fixed model training interval) under two environments. The training curve and triggered times are shown in Figure 2. Clearly, our mechanism improves the performance while reducing the total times of model training. Besides, we notice that the performance is comparable to other MBRL baselines (MBPO *etc.*) when fixing our model training interval at 250. Still it performs worse than that equipped with a smart mechanism to decide whether to train the model at current exploration step.

To better understand why the event-triggered mechanism brings up our outperformance, we asses its main bricks. Model shift, which reflects the current ability to digest new data, is the basis of triggered condition. And we estimate model shift from two parts, policy coverage and prediction error. In Figure 3 we observe that, gradually, as the training progresses, the policy coverage increases, which reflects our policy has new explorations at every stage without falling into a local optimum prematurely. Also, the prediction error gradually decreases, which implies that our estimated dynamics come closer to the true dynamics in the explored region. Figure 4 implies that the number of samples required to hit the constraint tends to grows and the model training frequency then goes down. This is also consistent with our intuition that, as the model refines and the exploration novelty fades, then the sample size required to reach a certain level of model shifts grows up. The result

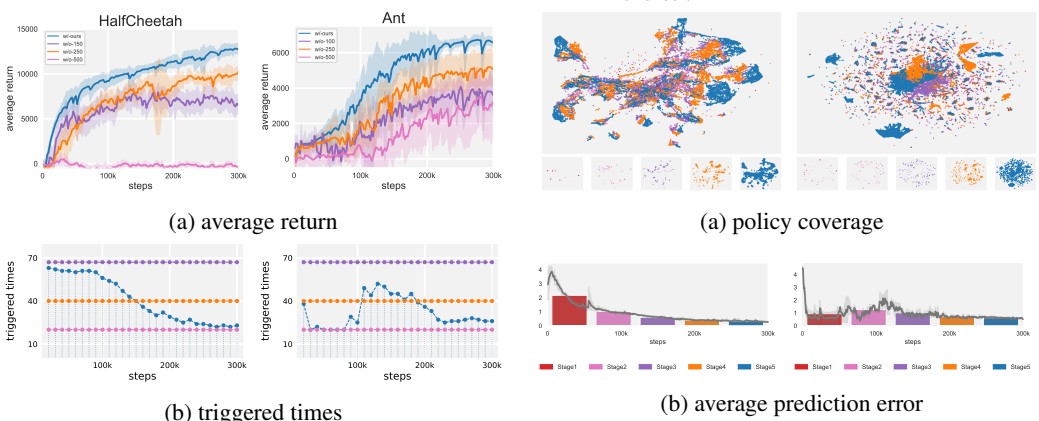

(a) average return

(a) policy coverage

(b) triggered times

(b) average prediction error

Figure 2: Ablation on event-triggered mechanism. These experiments are average over 5 random seeds. (a) shows the average return with or with-out event-triggered mechanism in HalfCheetah and Ant benchmarks. (b) shows the number of triggered times per 10k step.

Figure 3: Visualization of the policy coverage and prediction error on HalfCheetah and Ant. Each stage contains 60k steps. (a) implies that policy coverage expands over stages. (b) shows the prediction error over newly collected data. The bars are average values of each stage.

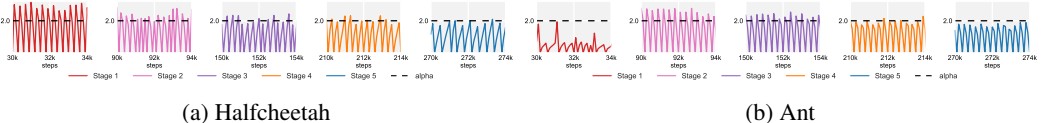

(a) Halfcheetah

(b) Ant

Figure 4: Visualization of our event-triggered mechanism on HalfCheetah and Ant benchmarks. Solid lines show the model shifts estimation and dotted lines reveals the threshold of triggered condition. Note that here we apply log value.

agrees with our theoretical analyses which describe that a dynamic alternation subject to the model shifts constraint does benefit to monotonicity rather than an assigned one.

**The generalizability of event-triggered mechanism.** We further investigate the generalizability of our proposed mechanism through ablation on policy optimization oracle, and results are shown in Figure 5.

**1) under Dyna-style.** We adopt TRPO [44] as the policy optimization oracle and test the performance with or without event-triggered mechanism in Halfcheetah and Ant benchmarks. Observably, our mechanism can effectively guarantee the overall performance improvement and alleviate the local optimization issue.

**2) jumping off Dyna-style.** We utilize iLQR [3] and then conduct ablation experiments in the DKitty-Stand [2] and Panda-Reaching [19] scenarios. The result shows that adding event-triggered mechanism has a comparable asymptotic performance as the most-frequently triggered case but enjoys a lower computation cost.

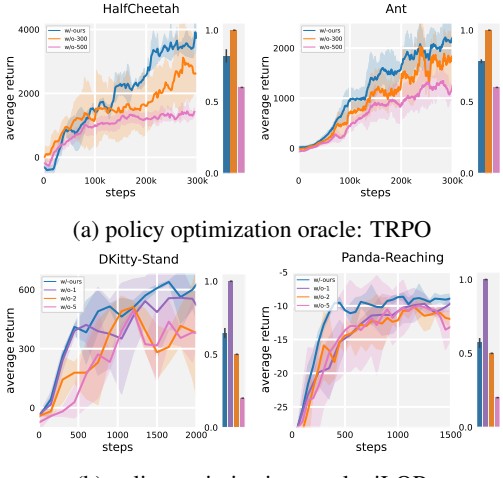

(a) policy optimization oracle: TRPO

(b) policy optimization oracle: iLQR

Figure 5: Ablation on generalizability via various policy optimization oracles. Line plots reflect the average return and the standard deviation over 5 trails. Bar plots indicate the total triggered times and the y-axis is scaled to [0,1].

# 7 Conclusion

We have investigated the role of the decision on "when to update the model" in joint optimization procedures through theoretical and empirical lens. We devise a general novel scheme for exploring the monotonicity of MBRL methods, distinguished from the existing discrepancy bound scheme. We then derive lower bounds under the scheme, suggesting that models with higher ceiling performance and lower bias guarantee a non-decreasing performance evaluated in the real environment. An effective constrained optimization problem comes from the follow-up refined bound to seek a non-negative lower bound. Further, the instance under the generative model setting further verifies the effectiveness of learning models from a dynamically varying number of explorations. The algorithm CMLO, stemming from these analyses, has asymptotic performance rivaling the best model-free algorithms and boasts better monotonicity. Further ablation studies reveal that the proposed mechanism scales to various policy optimization oracles and benefits computation cost reduction. Currently, we observed empirically that the event-triggered condition is usually related to specific environments, which will cost a little time for tuning. Thus, one direction that merits further investigation is to construct the dual problem of our constrained optimization problem for better exploring optimality and monotonicity.

## Acknowledgments and Disclosure of Funding

This work was jointly supported by the Sino-German Collaborative Research Project "Crossmodal Learning" (NSFC 62061136001/ DFG TRR169), the CAS Project for Young Scientists in Basic Research (Grant No.YSBR-040), the National Natural Science Foundation of China (No.62006137) and Beijing Outstanding Young Scientist Program (No.BJJWZYJH012019100020098). The authors would also like to thank the anonymous reviewers for their careful reading and their many insightful comments.

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
