# Appendices

## A  Sketch of Theoretical Analyses

Here, we present our sketch of theoretical analyses. We first construct a general scheme (Definition 4.1) for non-decreasing performance guarantee and follow it up by characterizing the lower bound under model shifts (Theorem 4.3). Towards seeking a non-negative lower bound, we restrict model shift and refine the bound (Theorem 4.6), then further reduce this issue to a constrained optimization problem (Proposition 4.7). With an instance under the generative model setting (Corollary 4.8), we demonstrate the merits of the dynamic model learning interval.

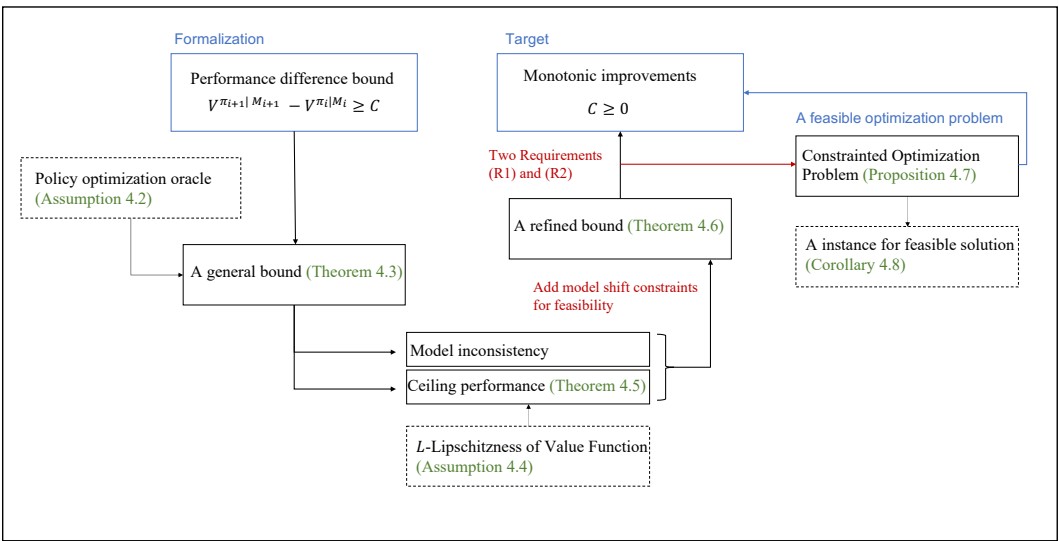

Figure 6: Theoretical sketch of CMLO

## B  Omitted Proofs

**Theorem B.1** (Performance difference bound for Model-based RL). *Let $\epsilon_{M_i}^{\pi_j}$ denote the inconsistency between the learned dynamics $P_{M_i}$ and the true dynamics, i.e. $\epsilon_{M_i}^{\pi_j} = \mathbb{E}_{s,a \sim d^{\pi_j}(s,a;\mu)}[\mathcal{D}_{\mathrm{TV}}(P(\cdot|s,a)\|P_{M_i}(\cdot|s,a))]$, where $d^\pi(s,a;\mu)$ is the probability of visiting $s,a$ after starting at state $s_0 \sim \mu$ and following $\pi_j \in \Pi$ thereafter under the true dynamics. Let policy $\pi_i$ be the $\epsilon_{opt}$-optimal policy under model $M_i$. Assume the performance discrepancy of policy $\pi$ between the estimated model $M_1$ and the true dynamics be approximated as $V_{M_1}^{\pi_1}(\mu) - V^{\pi_1}(\mu) = \kappa \cdot \epsilon_{M_1}^{\pi_1}$. Recall $\kappa = \frac{2R\gamma}{(1-\gamma)^2}$, then the performance gap between $\pi_1$ and $\pi_2$ evaluated in the true MDP can be bounded by:*

$$V^{\pi_2|M_2} - V^{\pi_1|M_1} \geq -\kappa \cdot (\epsilon_{M_2}^{\pi_2} - \epsilon_{M_1}^{\pi_1}) + V_{M_2}^* - V_{M_1}^* - \epsilon_{opt}$$

*Proof.* We overload notation $V^{\pi_i|M_i}$ and write $V^{\pi_i}$ for simplicity.

$$V^{\pi_2} - V^{\pi_1} = \underbrace{V^{\pi_2} - V_{M_2}^{\pi_2}}_{L_1} + \underbrace{V_{M_2}^{\pi_2} - V_{M_1}^{\pi_1}}_{L_2} - \underbrace{(V^{\pi_1} - V_{M_1}^{\pi_1})}_{L_3}$$

We can bound $L_1 - L_3$ using Lemma C.2, and bound $L_2$ using the property of $\epsilon_{opt}$-optimal.

For $L_1 - L_3$, with the performance gap approximation of $M_1$ and $\pi_1$, we apply Lemma C.2, and obtain: $L_1 - L_3 \geq -\kappa \cdot (\epsilon_{M_2}^{\pi_2} - \epsilon_{M_1}^{\pi_1})$.

We call a policy $\pi$ $\epsilon$-optimal under the dynamical model $M$, if $V_M^*(s) \geq V_M^\pi(s) \geq V_M^*(s) - \epsilon_{opt}$ for all $s \in \mathcal{S}$. Under the assumptions of black-box optimization oracle, we obtain:

$$L_2 \geq V_{M_2}^* - V_{M_1}^* - \epsilon_{opt}$$

Adding these two bounds together yields the desired result.

Remark that, when $V_{M_1}^{\pi_1} - V^{\pi_1}$ gets a value far away from $\kappa \cdot \epsilon_{M_1}^{\pi_1}$, it indicates the performance discrepancy that evaluated between the model $M_1$ and the environment is near to zero, which further indicates that the inconsistency between $P_{M_1}$ and the true dynamics is quite small and thus the optimization process has reached a stopping point. $\qquad\square$

**Theorem B.2** (Ceiling return gap Under model shifts). *For a dynamical model $M_i \in \mathcal{M}$, $V_{M_i}^*(\mu)$ denotes the maximal returns on dynamics $P_{M_i}$. Then the gap of optimal returns under these two models $M_1$, $M_2$ can be bounded as:*

$$V_{M_2}^* - V_{M_1}^* \geq -\frac{\gamma}{1-\gamma} L \cdot \sup_{\pi \in \Pi} \mathbb{E}_{s,a \sim d_{M_2}^\pi} \Big[ |P_{M_2}(\cdot|s,a) - P_{M_1}(\cdot|s,a)| \Big]$$

*Proof.* Let $G_{M_i,M_j}^\pi(s,a)$ be the discrepancy between $M_i$ and $M_j$ on a single state-action pair $(s,a)$, i.e. $G_{M_i,M_j}^\pi(s,a) = \mathbb{E}_{\tilde{s}' \sim P_{M_j}(\cdot|s,a)}[V_{M_j}^\pi(\tilde{s}')] - \mathbb{E}_{s' \sim P_{M_i}(\cdot|s,a)}[V_{M_j}^\pi(s')]$. We construct $Z_k$ be the discounted return when using $\pi$ to sample in model $M_i$ for $k$ steps and then in $M_j$ for the rest with a starting point $s_0 = s$, that is,

$$Z_k = \mathbb{E}_{\substack{\forall t, a_t \sim \pi(\cdot|s_t) \\ \forall t < k, s_{t+1} \sim P_{M_i}(\cdot|s_t,a_t) \\ \forall t \geq k, s_{t+1} \sim P_{M_j}(\cdot|s_t,a_t)}} \left[ \sum_{t=0}^{\infty} \gamma^t R(s_t,a_t) s_0 = s \right]$$

Base on this definition, we have that $V_{M_i}^\pi(s) = Z_\infty$ and $V_{M_j}^\pi(s) = Z_0$. Then, we can decompose $V_{M_i}^\pi(s) - V_{M_j}^\pi(s)$ into a sum of $Z_k$:

$$V_{M_i}^\pi(s) - V_{M_j}^\pi(s) = \sum_{k=0}^{\infty}(Z_{k+1} - Z_k)$$

We can find that, $Z_{k+1}$ and $Z_k$ only differ in their dynamical model used in the $k$-th step rollout. And we can rewrite them to be :

$$Z_k = r_k + \mathbb{E}_{s_k,a_k \sim \pi, P_{M_i}}\Big[ \mathbb{E}_{\tilde{s}_{k+1} \sim P_{M_j}(\cdot|s_k,a_k)}\big[ \gamma^{k+1} V_{M_j}^\pi(\tilde{s}_{k+1}) \big] \Big]$$

$$Z_{k+1} = r_k + \mathbb{E}_{s_k,a_k \sim \pi, P_{M_i}}\Big[ \mathbb{E}_{s_{k+1} \sim P_{M_i}(\cdot|s_k,a_k)}\big[ \gamma^{k+1} V_{M_j}^\pi(s_{k+1}) \big] \Big]$$

Here, $r_k$ denotes the reward from the first $j$ steps from the real environment. Combine the two equations above together and we get:

$$Z_{k+1} - Z_k = \gamma^{k+1} \mathbb{E}_{s_k,a_k \sim \pi, P_{M_i}} \left[ \mathbb{E}_{\substack{s_{k+1} \sim P_{M_i}(\cdot|s_k,a_k) \\ \tilde{s}_{k+1} \sim P_{M_j}(\cdot|s_k,a_k)}} \left[ V_{M_j}^\pi(s_{k+1}) - V_{M_j}^\pi(\tilde{s}_{k+1}) \right] \right]$$

Then, we can obtain the following conclusion by adding up all $Z_{k+1} - Z_k$:

$$V_{M_i}^\pi - V_{M_j}^\pi = \frac{\gamma}{1-\gamma} \mathbb{E}_{\substack{s \sim d_{M_i}^\pi \\ a \sim \pi(\cdot|s)}} \left[ \mathbb{E}_{s' \sim P_{M_i}(\cdot|s,a)}\left[ V_{M_j}^\pi(s') \right] - \mathbb{E}_{\tilde{s}' \sim P_{M_j}(\cdot|s,a)}\left[ V_{M_j}^\pi(\tilde{s}') \right] \right]$$

Here, $d_{M_i}^\pi$ denotes the distribution of state-action pair induced by policy $\pi$ under the dynamical model $M_i$.

For a starting state $s_0 = s$, and two dynamical models $M_2$, $M_1$, with the definition of $G_{M_1,M_2}^\pi$ we have:

$$V_{M_2}^\pi(s) - V_{M_1}^\pi(s) = \frac{\gamma}{1-\gamma} \mathbb{E}_{s,a \sim d_{M_1}^\pi} \left[ G_{M_1,M_2}^\pi(s,a) \right]$$

From the definition: $G_{M_1,M_2}^\pi(s,a) = \mathbb{E}_{\tilde{s}' \sim P_{M_2}(\cdot|s,a)}[V_{M_2}^\pi(\tilde{s}')] - \mathbb{E}_{s' \sim P_{M_1}(\cdot|s,a)}[V_{M_2}^\pi(s')]$.

In the case of deterministic dynamics: for clarity, we write $s' = M_i(s,a)$ instead of $s' \sim P_{M_i}(s'|s,a)$, then we rewrite $G_{M_1,M_2}^\pi(s,a)$ as: $G_{M_1,M_2}^\pi(s,a) = V_{M_2}^\pi(M_2(s,a)) - V_{M_2}^\pi(M_1(s,a))$, with the Lispchitzness, we have that $|G_{M_1,M_2}^\pi(s,a)| \leq L \cdot |M_2(s,a) - M_1(s,a)|$.

In the case of stochastic dynamics: when $L \geq \frac{R}{1-\gamma}$, we have

$$
\begin{aligned}
|G_{M_1,M_2}^\pi(s,a)| &= |\mathbb{E}_{\tilde{s}' \sim P_{M_2}(\cdot|s,a)}[V_{M_2}^\pi(\tilde{s}')] - \mathbb{E}_{s' \sim P_{M_1}(\cdot|s,a)}[V_{M_2}^\pi(s')]| \\
&= |\sum_{\tilde{s}' \in \mathcal{S}} (P_{M_2}(s'|s,a) - P_{M_1}(s'|s,a))V_{M_2}^\pi(s')| \\
&\leq |\max_{s'} V_{M_2}^\pi(s')| \cdot |P_{M_2}(\cdot|s,a) - P_{M_1}(\cdot|s,a)| \\
&\leq L \cdot |P_{M_2}(\cdot|s,a) - P_{M_1}(\cdot|s,a)|.
\end{aligned}
$$

Note that we require $L$ to be larger than $\frac{R}{1-\gamma}$ in the infinite horizon settings. Previous work [11] has found that $L$ has a dependence on $R \cdot H$ when equipped with a maximum horizon of $H$.

Thus, with the Lipschitzness of $V_{M_1}^\pi$ and $V_{M_2}^\pi$, we have that $|G_{M_1,M_2}^\pi(s,a)| \leq L \cdot |P_{M_2}(\cdot|s,a) - P_{M_1}(\cdot|s,a)|$.

$$
\begin{aligned}
|V_{M_2}^\pi(s) - V_{M_1}^\pi(s)| &= \frac{\gamma}{1-\gamma} \cdot |\mathbb{E}_{s,a \sim d_{M_2}^\pi}[G_{M_1,M_2}^\pi(s,a)]| \\
&\leq \frac{\gamma}{1-\gamma} \cdot \mathbb{E}_{s,a \sim d_{M_2}^\pi}[|G_{M_1,M_2}^\pi(s,a)|] \\
&\leq \frac{\gamma}{1-\gamma} L \cdot \mathbb{E}_{s,a \sim d_{M_2}^\pi}[|P_{M_2}(\cdot|s,a) - P_{M_1}(\cdot|s,a)|]
\end{aligned}
$$

Observe that $|\sup_x f(x) - \sup_x g(x)| \leq \sup_x |f(x) - g(x)|$, where $f$ And $g$ are real valued functions. This implies:

$$
|V_{M_2}^*(s) - V_{M_1}^*(s)| = |\sup_{\pi \in \Pi} V_{M_2}^\pi(s) - \sup_{\pi \in \Pi} V_{M_1}^\pi(s)| \leq \sup_{\pi \in \Pi} |V_{M_2}^\pi(s) - V_{M_1}^\pi(s)|
$$

Thus, we have that for starting state $s_0 \sim \mu$:

$$
V_{M_2}^*(\mu) - V_{M_1}^*(\mu) \geq -\frac{\gamma}{1-\gamma} L \cdot \sup_{\pi \in \Pi} \mathbb{E}_{s,a \sim d_{M_2}^\pi} \left[ |P_{M_2}(\cdot|s,a) - P_{M_1}(\cdot|s,a)| \right]
$$

$\square$

**Theorem B.3** (Refined bound with constraints). *Here, $\mathcal{S}$ denotes the state space simplex. Here, policy $\pi_i \in \Pi$ is the $\epsilon_{opt}$ optimal policy under the dynamical model $M_i \in \mathcal{M}$. For a dynamical model $M_i \in \mathcal{M}$, $V_{M_i}^*(\mu)$ denotes the maximal returns on dynamics $P_{M_i}$. We give the model shift constraints under the TV-distance. Then the performance difference bound can be refined under the model shift constraints as:*

$$
\begin{aligned}
V^{\pi_2|M_2} - V^{\pi_1|M_1} \geq &\kappa \cdot (\mathbb{E}_{s,a \sim d^{\pi_1}} \mathcal{D}_{\mathrm{TV}}[P(\cdot|s,a)\|P_{M_1}(\cdot|s,a)] \\
&-\mathbb{E}_{s,a \sim d^{\pi_2}} \mathcal{D}_{\mathrm{TV}}[P(\cdot|s,a)\|P_{M_2}(\cdot|s,a)]) - \frac{\gamma}{1-\gamma} L \cdot 2\sigma_{M_1,M_2} - \epsilon_{opt} \\
s.t. \quad &\mathcal{D}_{\mathrm{TV}}(P_{M_2}(\cdot|s,a)\|P_{M_1}(\cdot|s,a)) \leq \sigma_{M_1,M_2}, \quad \forall(s,a) \in \mathcal{S} \times \mathcal{A}
\end{aligned}
$$

*Proof.* Let $\mu$ and $v$ be two probability distributions on the configuration space $\mathcal{X}$, according to Lemma C.1, then we have $\mathcal{D}_{\mathrm{TV}}(\mu\|v) = \frac{1}{2} \sum_{x \in \mathcal{X}} |\mu(x) - v(x)|$.

Recall $\epsilon_{M_i}^\pi$ denote the discrepancy between the learned dynamics $P_{M_i}$ and the true dynamics, i.e. $\epsilon_{M_i}^\pi = \mathbb{E}_{s,a \sim d^\pi}[\mathcal{D}_{\mathrm{TV}}(P(\cdot|s,a)\|P_{M_i}(\cdot|s,a))]$.

Under these definitions, we can yield the following intermediate outcome by applying the results from B.2 and B.1

$$
\begin{aligned}
V^{\pi_2|M_2} - V^{\pi_1|M_1} \geq &-\kappa \cdot (\epsilon_{M_2}^{\pi_2} - \epsilon_{M_1}^{\pi_1}) + V_{M_2}^* - V_{M_1}^* - \epsilon_{opt} \\
\geq &-\kappa \cdot \left\{ \mathbb{E}_{s,a \sim d^{\pi_2}} \mathcal{D}_{\mathrm{TV}}[P(\cdot|s,a)\|P_{M_2}(\cdot|s,a)] \right. \\
&\left. - \mathbb{E}_{s,a \sim d^{\pi_1}} \mathcal{D}_{\mathrm{TV}}[P(\cdot|s,a)\|P_{M_1}(\cdot|s,a)] \right\} \\
&- \frac{\gamma}{1-\gamma} L \cdot \sup_{\pi \in \Pi} \mathbb{E}_{s,a \sim d_{M_2}^\pi} \left[ |P_{M_2}(\cdot|s,a) - P_{M_1}(\cdot|s,a)| \right] - \epsilon_{opt}
\end{aligned}
$$

Recall the following constraint on model shift that we subject to:

$$\mathcal{D}_{\mathrm{TV}}(P_{M_2}(\cdot|s,a)\|P_{M_1}(\cdot|s,a)) \le \sigma_{M_1,M_2}, \quad \forall (s,a) \in \mathcal{S} \times \mathcal{A}$$

Then, the ceiling performance difference can be further bounded as:

$$
\begin{aligned}
V_{M_2}^* - V_{M_1}^* &\ge -\frac{\gamma}{1-\gamma}L \cdot \sup_{\pi \in \Pi} \mathbb{E}_{s,a \sim d_{M_2}^\pi}\Big[|P_{M_2}(\cdot|s,a) - P_{M_1}(\cdot|s,a)|\Big] \\
&\ge -\frac{\gamma}{1-\gamma}L \cdot \sup_{\pi \in \Pi} \mathbb{E}_{s,a \sim d_{M_2}^\pi}\Big[2\sum_{s' \in \mathcal{S}_2}\frac{1}{2}|P_{M_2}(s'|s,a) - P_{M_1}(s'|s,a)|\Big] \\
&\ge -\frac{\gamma}{1-\gamma}L \cdot \sup_{\pi \in \Pi} \mathbb{E}_{s,a \sim d_{M_2}^\pi}\Big[2\mathcal{D}_{\mathrm{TV}}(P_{M_2}(\cdot|s,a)\|P_{M_1}(\cdot|s,a))\Big] \\
&\ge -\frac{\gamma}{1-\gamma}L \cdot (2\sigma_{M_1,M_2})
\end{aligned}
$$

And finally, we have the following refined bound:

$$
\begin{aligned}
V^{\pi_2|M_2} - V^{\pi_1|M_1} \ge &\kappa \cdot (\mathbb{E}_{s,a \sim d^{\pi_1}}\mathcal{D}_{\mathrm{TV}}[P(\cdot|s,a)\|P_{M_1}(\cdot|s,a)] - \mathbb{E}_{s,a \sim d^{\pi_2}}\mathcal{D}_{\mathrm{TV}}[P(\cdot|s,a)\|P_{M_2}(\cdot|s,a)]) \\
&- \frac{\gamma}{1-\gamma}L \cdot (2\sigma_{M_1,M_2}) - \epsilon_{opt}
\end{aligned}
$$

This refined bound is subject to the model shift constraint we set.

$\square$

**Corollary B.4.** *Under the generative model setting, let model $M_1$ has already trained on $N$ samples of each $(s,a)$ pair and model $M_2$ on $N+k$ samples per $(s,a)$ pair. Policy $\pi_1$ is the $\epsilon_{opt}$-optimal policy on $M_1$ and so is $\pi_2$ on $M_2$. Recall $\epsilon = \delta_{M_1}(\cdot|s,a) - \frac{(1-\gamma)L}{R} \cdot (2\sigma_{M_1,M_2}) - \frac{(1-\gamma)^2}{R\gamma} \cdot \epsilon_{opt}$ and $\xi \in (0,1)$ is a constant. As the model is trained on true interaction samples, we can work out the amount of samples we need to satisfy the monotonic improvement requirements:*

$$k = \frac{2}{\epsilon^2} \log \frac{2^{vol(\mathcal{S})} - 2}{\xi} - N$$

*Proof.* For simplicity, we denote $\delta_{M_i}(s'|s,a) = |P(s'|s,a) - P_{M_i}(s'|s,a)|$.

Recall the lower bound for performance difference:

$$
\begin{aligned}
C_{M_1,\pi_1}(M_2,\pi_2) = \frac{2R\gamma}{(1-\gamma)^2}\Big\{ &\mathbb{E}_{s,a \sim d^{\pi_1}}\Big[\frac{1}{2}\sum_{s' \in \mathcal{S}_2}(\delta_{M_1}(s'|s,a))\Big] \\
&- \mathbb{E}_{s,a \sim d^{\pi_2}}\Big[\frac{1}{2}\sum_{s' \in \mathcal{S}_2}(\delta_{M_2}(s'|s,a))\Big]\Big\} - \frac{\gamma}{1-\gamma}L \cdot (2\sigma_{M_1,M_2}) - \epsilon_{opt}
\end{aligned}
$$

Towards this lower bound, we give the assumption that similar models derives similar sub-obptimal policies. To be specific, when $M_1$ and $M_2$ are close to each other in terms of $L_1$-norm distance at any transition pair $(s,a)$ : $|P_{M_2}(\cdot|s,a) - P_{M_1}(\cdot|s,a)|_1 \le \delta$, then the sub-optimal policies derived from them are close as well, *i.e.*, there exists $\alpha$, subject to $|\pi_2(\cdot|s)-\pi_1(\cdot|s)|_1 \le \alpha \cdot \delta$ for all transition pairs. Here, we show the feasibility of given the $\alpha$ in the control theory perspective. Let a trajectory $s_{1:N}, a_{1:N-1}$ be generated from the dynamical model $M_1$ that satisfies:

$$s_{t+1} = f_{M_1}(x_t, a_t, 0)$$

We call it a nominal trajectory. Here, $f_{M_1}$ is a nonlinear function that represents the dynamics of model $M_1$. Then, we can formalize the inconsistency between $M_2$ and $M_1$ by disturbance, $w_i \in \mathcal{W}$. By entering $w_i$ into $f_{M_1}$ in a general nonlinear way we can get the dynamic of model $M_2$ as:

$$s_{t+1} = f_{M_1}(s_t, a_t, w_t)$$

Further, the deviations from the nominal trajectory can be calculated as when giving a disturbance sequence, $w_{1:N-1}$. Let $\delta y$ denote the deviations of variable $y$.

$$\delta s_t = f_{M_1}(s_t + \delta x_t, u_t, +\delta u_t, w_t) - s_{t+1}$$

Assume that the deviations are computed with a linear feedback controller, that is,

$$\delta a_t = -K_t \delta s_t$$

Actually, we can utilize any reasonable linear controller. Here, we take the time-varying linear quadratic regulator as an instance for illustrating the rationality of our assumption on $\alpha$. Based on the dynamic Riccati equation, we have the solution as:

$$K_t = (R + B_t^T P_{t+1} B_t)^{-1} (B_t^T P_{t+1} A_t)$$
$$P_{t-1} = Q_t + A_t^T P_t A_t - A_t^T P_t B_t (R + B_t^T P_t B_t)^{-1} (B_t^T P_t A_t)$$

where $A_t = \partial f_{M_1}/\partial s|_{s_t,a_t,0}$ and $B_t = \partial f_{M_1}/\partial a|_{s_t,a_t,0}$. And, $Q_i \succeq 0$ and $R \succeq 0$ are state and input cost matrices. That is, we have a feasible solution for $\alpha$. We then seek to explain the state-action distribution is similar thus we can use $d^{\pi_1}$ as an approximation of $d^{\pi_2}$. First of all, we have the distance between two policies,

$$|\pi_2(\cdot|s) - \pi_1(\cdot|s)|_1 \le \alpha \cdot \delta \triangleq \beta$$

Denote $\mathbb{P}_\pi^h$ as the state distribution resulting from $\pi$ at time step $h$ with $\mu$ as the initial state distribution. We consider bounding $|\mathbb{P}_h^{\pi_2} - \mathbb{P}_h^{\pi_1}|_1$ with $h > 1$.

$$\mathbb{P}_h^{\pi_2}(s') - \mathbb{P}_h^{\pi_1}(s')$$
$$= \sum_{s,a} \left( \mathbb{P}_{h-1}^{\pi_2}(s)\pi_2(a|s) - \mathbb{P}_{h-1}^{\pi_1}(s)\pi_1(a|s) \right) P(s'|s,a)$$
$$= \sum_{s,a} \left( \mathbb{P}_{h-1}^{\pi_2}(s)\pi_2(a|s) - \mathbb{P}_{h-1}^{\pi_2}(s)\pi_1(a|s) + \mathbb{P}_{h-1}^{\pi_2}(s)\pi_1(a|s) - \mathbb{P}_{h-1}^{\pi_1}(s)\pi_1(a|s) \right) P(s'|s,a)$$
$$= \sum_s \mathbb{P}_{h-1}^{\pi_2}(s) \sum_a (\pi_2(a|s) - \pi_1(a|s)) P(s'|s,a) + \sum_s \left( \mathbb{P}_{h-1}^{\pi_2}(s) - \mathbb{P}_{h-1}^{\pi_1}(s) \right) \sum_a \pi_1(a|s) P(s'|s,a).$$

Apply absolute value on both sides, we then get:

$$\sum_{s'} |\mathbb{P}_h^{\pi_2}(s') - \mathbb{P}_h^{\pi_1}(s')| \le \sum_s \mathbb{P}_{h-1}^{\pi_2}(s) \sum_a |\pi_2(a|s) - \pi_1(a|s)| \sum_{s'} P(s'|s,a)$$
$$+ \sum_s |\mathbb{P}_{h-1}^{\pi_2}(s) - \mathbb{P}_{h-1}^{\pi_1}(s)| \sum_{s'} \sum_a \pi_1(a|s) P(s'|s,a)$$
$$\le \beta + \|\mathbb{P}_{h-1}^{\pi_2} - \mathbb{P}_{h-1}^{\pi_1}\|_1 \le 2\beta + |\mathbb{P}_{h-2}^{\pi_2} - \mathbb{P}_{h-2}^{\pi_1}|_1 = h\beta.$$

Under the definition of $d_\mu^\pi$, we have:

$$|d_\mu^{\pi_2} - d_\mu^{\pi_1}|_1 = |(1-\gamma) \sum_{h=0}^\infty \gamma^h (\mathbb{P}_h^{\pi_2} - \mathbb{P}_h^{\pi_1})| \le \beta\gamma/(1-\gamma).$$

Upon these analyses, we find that similar model derives similar policy, which invokes similar state-action distribution. Thus we can approximate state-action visitation density $d^{\pi_2}$ by previous visitation density $d^{\pi_1}$. Also, we assume the same state space $\mathcal{S}$. Then, we get a approximation of $C_{M_1,\pi}(M_2, \hat{\pi})$ as following:

$$\tilde{C}_{M_1,\pi_1}(M_2, \pi_2)$$
$$= \frac{2R\gamma}{(1-\gamma)^2} \mathbb{E}_{s,a} \left[ \frac{1}{2} \sum_{s'\in\mathcal{S}} (\delta_{M_1}(s'|s,a) - \delta_{M_2}(s'|s,a)) \right] - \frac{\gamma}{1-\gamma} L \cdot (2\sigma_{M_1,M_2}) - \epsilon_{opt}$$
$$= \frac{\gamma}{(1-\gamma)} \left\{ \frac{R}{1-\gamma} \mathbb{E}_{s,a} \left[ \sum_{s'\in\mathcal{S}} (\delta_{M_1}(s'|s,a) - \delta_{M_2}(s'|s,a)) \right] - L \cdot \mathbb{E}_{s,a} \left[ \sum_{s'\in\mathcal{S}} (\frac{1}{vol(\mathcal{S})} 2\sigma_{M_1,M_2}) \right] \right.$$
$$\left. - \frac{(1-\gamma)}{\gamma} \mathbb{E}_{s,a} \left[ \frac{1}{vol(\mathcal{S})} \sum_{s'\in\mathcal{S}} (\epsilon_{opt}) \right] \right\}$$

Thus, when meeting the following requirements for each $(s,a)$ pair, we can guarantee the monotonic improvement for that $V^{\pi_2|M_2} - V^{\pi_1|M_1} \ge \tilde{C}_{M_1,\pi_1}(M_2, \pi_2) \ge 0$.

$$\delta_{M_2}(\cdot|s,a) \le \delta_{M_1}(\cdot|s,a) - \frac{(1-\gamma)L}{R} \cdot (2\sigma_{M_1,M_2}) - \frac{(1-\gamma)^2}{R\gamma} \cdot \epsilon_{opt}$$

By applying Lemma C.3, the $L_1$ deviation of the empirical distribution $P_{M_2}(\cdot|s, a)$ and true $P(\cdot|s, a)$ over $vol(\mathcal{S})$ distinct events from $n$ samples is bounded by:

$$\Pr(|P(\cdot|s, a) - P_{M_2}(\cdot|s, a)|_1 < \epsilon) \geq 1 - (2^{vol(\mathcal{S})} - 2) \exp(-\frac{(N + k)\epsilon^2}{2})$$

Then for a fixed $(s, a)$, with probability greater than $1 - \xi$, we have:

$$|P(\cdot|s, a) - P_{M_2}(\cdot|s, a)|_1 \leq \sqrt{\frac{2}{N + k} \cdot \log \frac{2^{vol(\mathcal{S})} - 2}{\xi}}$$

Let $\epsilon = \delta_{M_1}(\cdot|s, a) - \frac{(1-\gamma)L}{R} \cdot (2\sigma_{M_1, M_2}) - \frac{(1-\gamma)^2}{R\gamma} \cdot \epsilon_{opt}$. Our requirements can be further shown in this form:

$$\sqrt{\frac{2}{N + k} \cdot \log \frac{2^{vol(\mathcal{S})} - 2}{\xi}} = \delta_{M_1}(\cdot|s, a) - \frac{(1 - \gamma)L}{R} \cdot (2\sigma_{M_1, M_2}) - \frac{(1 - \gamma)^2}{R\gamma} \cdot \epsilon_{opt}$$

Finally, with probability greater than $1 - \xi$, we can guarantee the monotonic improvement when having:

$$k = \frac{2}{\epsilon^2} \log \frac{2^{vol(\mathcal{S})} - 2}{\xi} - N$$

$\square$

## C   Toolbox

**Lemma C.1** (Total variation distance). *Let $\mu$ and $v$ be two probability distributions on the configuration space $\mathcal{X}$. Then*

$$\mathcal{D}_{\mathrm{TV}}(\mu\|v) = \frac{1}{2} \sum_{x \in \mathcal{X}} |\mu(x) - v(x)|$$

*Proof.* Let $B = \{x \in \mathcal{X} : \mu(x) \geq v(x)\}$, and $A \subseteq \mathcal{X}$ be any event. Since $\mu(x) - v(x) < 0$ for any $x \in A \cap B^c$, we have

$$\mu(A) - v(A) \leq \mu(A \cap B) - v(A \cap B) \leq \mu(B) - v(B)$$

For all events $A$, $|\mu(A) - v(A)|\mu(B) - v(B)$, and the equality is achieved for $A = B$ or $A = B^c$. Thus, we get that

$$\mathcal{D}_{\mathrm{TV}}(\mu\|v) = \frac{1}{2}[\mu(B) - v(B) + v(B^c) - \mu(B^c)] = \frac{1}{2} \sum_{x \in \mathcal{X}} |\mu(x) - v(x)|$$

$\square$

**Lemma C.2** (Relationship between true returns and model returns). *Let $\epsilon_M^\pi$ denote the inconsistency between the learned dynamics $P_M$ and the true dynamics, $\mathbb{E}_{s,a \sim d^\pi}[\mathcal{D}_{\mathrm{TV}}(P(\cdot|s, a)\|P_M(\cdot|s, a))]$, where $d^\pi \doteq d^\pi(s, a; \mu)$ is the probability of visiting state-action pair $(s, a)$ after starting at state $s_0 \sim \mu$ and following $\pi$ thereafter under the true dynamics. Then the true returns can be represented as below:*

$$V^\pi(\mu) \geq V_M^\pi(\mu) - \frac{2R\gamma}{(1 - \gamma)^2} \epsilon_M^\pi$$

*Proof.* Given policy $\pi$ and dynamics $P(\cdot|s, a)$, we denote the density of state-action visitation after $h$ steps from starting state $s_0 \sim \mu$ as $\rho_h^\pi(\mu; P) = \mathbb{E}_{s_0 \sim \mu}[\rho_h^\pi(s_h, a_h|s_0; P)]$.

Then the discounted returns are bounded as:

$$V^\pi(\mu) - V_M^\pi(\mu) = \sum_{h=0}^\infty \mathbb{E}_{s,a\sim\rho_h^\pi(\mu;P)}[\gamma^h r(s,a)] - \sum_{h=0}^\infty \mathbb{E}_{s,a\sim\rho_h^\pi(\mu;P_M)}[\gamma^h r(s,a)]$$

$$\geq -\sum_{h=0}^\infty \gamma^h |\mathbb{E}_{s,a\sim\rho_h^\pi(\mu;P)}[r(s,a)] - \mathbb{E}_{s,a\sim\rho_h^\pi(\mu;P_M)}[\gamma^h r(s,a)]|$$

$$\geq -\sum_{h=0}^\infty \gamma^h \sum_{s,a\in\mathcal{S}\times\mathcal{A}} R|\rho_h^\pi(\mu;P) - \rho_h^\pi(\mu;P_M)|$$

$$= -2R\cdot\sum_{h=0}^\infty \gamma^h \frac{1}{2}\sum_{s,a\in\mathcal{S},\mathcal{A}} |\rho_h^\pi(\mu;P) - \rho_h^\pi(\mu;P_M)|$$

$$= -2R\cdot\sum_{h=0}^\infty \gamma^h \mathcal{D}_{\mathrm{TV}}(\rho_h^\pi(\mu;P)\|\rho_h^\pi(\mu;P_M))$$

Using the property of Markov chain TV distance bound, then,

$$V^\pi(\mu) - V_M^\pi(\mu) \geq -2R\cdot\sum_{h=0}^\infty \gamma^h \mathcal{D}_{\mathrm{TV}}(\rho_h^\pi(\mu;P)\|\rho_h^\pi(\mu;P_M))$$

$$\geq \sum_{h=1}^\infty -2R\cdot\gamma^h\Big\{\mathcal{D}_{\mathrm{TV}}(\rho_{h-1}^\pi(\mu;P)\|\rho_{h-1}^\pi(\mu;P_M))$$

$$+ \mathbb{E}_{s,a\sim d^\pi}[\mathcal{D}_{\mathrm{TV}}[(P(\cdot|s,a)\|P_M(\cdot|s,a))]] + \mathcal{D}_{\mathrm{TV}}(\pi\|\pi)\Big\}$$

By plugging the results back, we then get,

$$V^\pi(\mu) - V_M^\pi(\mu) \geq -2R\sum_{h=0}^\infty \gamma^h h\cdot\epsilon_M^\pi = -\frac{2R\gamma}{(1-\gamma)^2}\cdot\epsilon_M^\pi$$

$\square$

**Lemma C.3** (Inequalities for the $L_1$ deviation of the empirical distribution). *Let $P$ be a probability distribution on the set $\mathcal{A} = \{1,\ldots,a\}$. For a sequence of samples $x_1,\ldots,x_m \sim P$, let $\hat{P}$ be the empirical probability distribution on $\mathcal{A}$ defined by $\hat{P}(j) = \frac{1}{m}\sum_{i=1}^m \mathbb{1}(x_i = j)$. The $L_1$-deviation of the true distribution $P$ and the empirical distribution $\hat{P}$ over $\mathcal{A}$ from $m$ independent identically samples is bounded by,*
$$Pr(|P - \hat{P})|_1 \geq \epsilon) \leq (2^{|\mathcal{A}|} - 2)e^{-m\epsilon^2/2}.$$

*Proof.* For a probability distribution $P$ on $\mathcal{A}$, we define
$$\pi_p = \max_{A\subseteq\mathcal{A}} \min(P(A), 1 - P(A)).$$

And for $p \in [0, 1/2)$, we define
$$\varphi(p) = \frac{1}{1-2p}\log\frac{1-p}{p}.$$
and, by continuity, set $\varphi(1/2) = 2$.

According to Weissman et al. [56], the $L_1$-deviation of the true distribution $P$ and the empirical distribution $\hat{P}$ is bound by,
$$Pr(|P - \hat{P})|_1 \geq \epsilon) \leq (2^{|\mathcal{A}|} - 2)e^{-m\varphi(\pi_P)\epsilon^2/4}.$$

Firstly, for any $P$, we have
$$\pi_p = \max_{A\subseteq\mathcal{A}} \min(P(A), 1 - P(A)) \leq \max_{A\subseteq\mathcal{A}}(\frac{P(A) + 1 - P(A)}{2}) = 1/2.$$

and note that $\pi_P = 1/2$ when $P(A) = 1/2$.

Then, we claim that the function $\varphi(p)$ is strictly decreasing for $p \in [0, 1/2]$. Differentiating $\varphi(p)$ with respect to $p$ yields

$$\varphi'(p) = \frac{1}{(1-2p)^2}\left[-\frac{1-2p}{1-p} - \frac{1-2p}{p} + 2\log\frac{1-p}{p}\right].$$

For $p \in (0, 1/2)$, there always exists $\frac{1}{(1-2p)^2} > 0$. Thus, to show that $\varphi'(p) < 0$ for $p \in (0, 1/2)$, it suffices to show that

$$g(p) = -\frac{1-2p}{1-p} - \frac{1-2p}{p} + 2\log\frac{1-p}{p} < 0.$$

The derivative of $g(p)$ is

$$g'(p) = (\frac{1}{1-p} - \frac{1}{p})^2 > 0, \quad p \in (0, 1/2).$$

Note that $g(1/2) = 0$, thus we have $\varphi'(p) < 0$ for $p \in (0, 1/2)$. And continuity arguments complete the claim for $p = 1/2$ and $p = 0$.

It is then no difficult to see that for any probability distribution $P$,

$$\varphi(\pi_P) \geq \varphi(1/2) = 2.$$

Therefore

$$Pr(|P - \hat{P})|_1 \geq \epsilon) \leq (2^{|\mathcal{A}|} - 2)e^{-m\epsilon^2/2}.$$

$\square$

## D  Comparison with Prior Works

To begin with, an important fact is that the effect of model shifts on trajectories is drastic. For example, even when the system dynamics satisfy $L$- Lipschitz continuity, along with the policy and the initial state are the same, the difference in trajectories sampled in $M_1, M_2$ grows at $e^{LH}$ with the length $H$ of the trajectory [32]. As the model shifts decay, the trajectory discrepancy will also decrease sharply. It implies that model shift stays a substantial influence during the MBRL training process.

There are two main trends of local view analysis:

**API [22] class.**  Their recipe for monotonicity analysis is $V^{\pi_{n+1}}(\mu) - V^{\pi_n}(\mu) \geq C(\pi_n, \pi_{n+1}, \epsilon_m)$. If policies update $\pi_n \to \pi_{n+1}$ could provide a non-negative $C(\pi_n, \pi_{n+1}, \epsilon_m)$, then the performance is guaranteed to increase. Here, $\epsilon_m = \max_{\pi \in \Pi, M \in \mathcal{M}} \mathbb{E}_{s,a\sim d^\pi}[\mathcal{D}_{TV}(P(\cdot|s,a)\|P_M(\cdot|s,a))]$. Most previous works [44, 22] were derived under model-free settings ($\epsilon_m = 0$). they use conservative policy iteration, for example, by forcing $\mathcal{D}_{TV}(\pi_n\|\pi_{n+1}) \leq \alpha$), then the state-action distribution are close as well $\mathcal{D}_{TV}(d^{\pi_n}\|d^{\pi_{n+1}}) \leq \frac{\alpha\gamma}{1-\gamma}$, so that they can optimize over their performance difference lemma, e.g., $C(\pi_n, \pi_{n+1}, 0) \approx \frac{1}{1-\gamma}\mathbb{E}_{s,a\sim d^{\pi_n}}[A^{\pi_n}(s,a)]$. When $\epsilon_m > 0$, this approximation $C(\pi_n, \pi_{n+1}, 0) \approx \frac{1}{1-\gamma}\mathbb{E}_{s,a\sim d^{\pi_n}}[A^{\pi_n}(s,a)]$ fails. It is non-trivial to apply the results to model-based settings.

DPI [49] tries to force $\pi_{n+1}$ and $\pi_n$ to be close, which will result in a high similarity of the data they sample. Thus, a risk arises from it, this approach would limit the growth of the policy exploration in the real environment, thereby leading the inferred models to stay optimized in a restrictive local area. For example, in the Humanoid environment, the agent struggles to achieve balance at the beginning of training. By then, an updated restricted policy will cause the exploration space to be limited in such an unbalanced distribution for a long time, and the learned model in such highly repetitive data will converge quickly with a validation loss be zero. However, the success trajectory has not been explored yet, implying that both the policy and the learned model will fall into a poor local optimum.

Besides, the definition of model accuracy (Eq.3) is a local view in DPI, i.e., $\hat{P}$ is $\delta$-opt under $d^{\pi_n}$. If we replace model accuracy with a more general, global definition (for example, $\hat{P}$ is $\delta$-opt under

$d^{\pi_{n+1}}$, or $\hat{P}$ is $\delta$-opt under all $(s, a, s')$ tuples), we find that the $\delta$ in (Eq. 3) will be large at the initial steps, making it difficult to obtain a local optimal solution in Theorem 3.1.

Finally, theoretical analysis in DPI can only guide the policy iteration process, while the update of the model is passive, which is different from our global view theory.

**Discrepancy bound class [34, 20].** They mostly derive upon $V^{\pi_n}(\mu) \geq V_M^{\pi_n}(\mu) - C(\epsilon_m, \epsilon_\pi)$. As guaranteed in them, once a policy update $\pi_n \to \pi_{n+1}$ has improved returns under the same model $M$, i.e., $V_M^{\pi_{n+1}}(\mu) > V_M^{\pi_n}(\mu) + C(\epsilon_m, \epsilon_\pi)$, it would improve the lower bound on the performance evaluated in the real environment, i.e., $\inf\{V^{\pi_2|M}(\mu)\} > \inf\{V^{\pi_1|M(\mu)}\}\}$.

Their theory is based on a fixed model $M$, or an upper bound on the distribution shift of all models $\epsilon_m$. It does not concern the change in model dynamics during updating, nor the performance varying due to the model shift. Moreover, The solution would be very coarse if only the upper bound of the model shift is given. Even worse, the given upper bound is likely to be too large, then it will fail to find a feasible solution for $V_M^{\pi_{n+1}}(\mu) - V_M^{\pi_n}(\mu) \geq C(\epsilon_m, \epsilon_\pi)$ in practice, thus making the monotonicity guarantee fails.

# E   Experimental Details

## E.1   Environment Setup

We evaluate all algorithms on a set of MuJoCo [54] continuous control benchmark tasks. We adopt the standard full-length version of all these tasks. Among then, we truncate some redundant observations for Hopper, Ant and Humanoid as our model-based baselines (MBPO[20], AutoMBPO[28]) do. The details of the experimental environments are provided in Table 1.

Table 1: Overview on Environment settings. Here, $\theta_t$ denotes the joint angle at time $t$. and $z_t$ denotes the height.

|  | State Space Dimension | Action Space Dimension | Horizon | Terminal Function |
|---|---|---|---|---|
| Hopper-v2 | 11 | 3 | 1000 | $z_t \leq 0.7$ or $\theta_t \geq 0.2$ |
| Swimmer-v2 | 8 | 2 | 1000 | None |
| Walker2d-v2 | 17 | 6 | 1000 | $z_t \geq 2.0$ or $z_t \leq 0.8$ or $\theta_t \leq -1.0$ or $\theta_t \geq 1.0$ |
| HalfCheetah-v2 | 17 | 6 | 1000 | None |
| Ant-v2 | 27 | 8 | 1000 | $z_t < 0.2$ or $z_t > 1.0$ |
| Humanoid-v2 | 45 | 17 | 1000 | $z_t < 1.0$ or $z_t > 2.0$ |

The environment settings for the ablation study on the generalizability of event-triggered mechanism are presented in Table 2

Table 2: Overview on Environment settings in Ablation.

|  | State Space Dimension | Action Space Dimension | Horizon | Terminal Function |
|---|---|---|---|---|
| Kitty Stand | 61 | 12 | 50 | $u_{t,kitty} \leq 0$ |
| Panda Reach | 20 | 7 | 50 | None |

## E.2 Baselines and implementation

**MFRL algorithms.** We compare to two state-of-the-art model-free baselines, SAC [16] and PPO [46]. The hyperparameters are kept the same as the authors. Regarding the low sample efficiency of MFRL methods, we ran 5M steps for them, which is an order of magnitude more than in MBRL, to fairly evaluate the asymptotic performance of these MFRL algorithms. The implementation of SAC is based on the opensource repo (pranz24 [42], MIT License).

**MBRL algorithms.** As for model-based methods, we compare with several algorithms including PETS [8], SLBO [34], MBPO [20] and AutoMBPO [28]. Our algorithm CMLO is implemented based on the opensource toolbox for MBRL algorithms, MBRL-LIB [40] (MIT License). The implementation of SLBO mainly follows Wang et al. [55]. To ensure a fair comparison, we run CMLO and MBPO with the same network architectures and training configurations based on MBRL-LIB.

We report the asymptotic performance on six benchmark tasks in Table 3. Results show that our method has comparable asymptotic performance in each benchmarks to both MBRL and MFRL baselines. Each result is averaged over seven trials using different random seeds. For MBRL baselines, the performances on different tasks are capped at different timesteps when the learning curves come to converge, we choose 125k for Hopper, 350k for Walker2d and Swimmer, 400k for HalfCheetah, 300k for Ant and 250k for Humanoid.

Table 3: Comparative results. The results show the average and standard deviation on the **maximum average returns** among different trails.

|  |  | Hopper | Walker2d | Swimmer |
|---|---|---|---|---|
| MFRL (@5M steps) | SAC | $4257.92 \pm 100.23$ | $7898.01 \pm 563.45$ | $195.60 \pm 5.97$ |
|  | PPO | $3114.76 \pm 1039.06$ | $5740.75 \pm 500.89$ | $129.66 \pm 9.78$ |
| MBRL | PETS | $571.25 \pm 71.14$ | $1174.79 \pm 471.39$ | $92.61 \pm 4.29$ |
|  | SLBO | $278.82 \pm 65.83$ | $3129.70 \pm 154.16$ | $71.02 \pm 1.98$ |
|  | AutoMBPO | $3534.46 \pm 77.53$ | $6276.99 \pm 1878.56$ | $184.89 \pm 58.84$ |
|  | MBPO | $2831.23 \pm 1109.63$ | $6285.64 \pm 538.32$ | $145.70 \pm 18.14$ |
|  | Ours | $\mathbf{3666.90 \pm 22.71}$ | $\mathbf{7749.90 \pm 523.27}$ | $\mathbf{185.14 \pm 1.73}$ |
|  |  | HalfCheetah | Ant | Humanoid |
| MFRL (@5M steps) | SAC | $16015.64 \pm 351.21$ | $7105.49 \pm 169.61$ | $8036.12 \pm 480.60$ |
|  | PPO | $6733.45 \pm 1528.87$ | $4427.39 \pm 836.02$ | $3068.95 \pm 1600.89$ |
| MBRL | PETS | $12023.84 \pm 3340.02$ | $3558.99 \pm 140.76$ | $1335.84 \pm 292.27$ |
|  | SLBO | $3993.43 \pm 127.17$ | $2492.19 \pm 92.02$ | $644.16 \pm 237.00$ |
|  | AutoMBPO | $12044.35 \pm 1550.21$ | $5792.35 \pm 415.46$ | $5780.14 \pm 245.01$ |
|  | MBPO | $13171.53 \pm 937.65$ | $5894.45 \pm 702.39$ | $5905.68 \pm 420.64$ |
|  | Ours | $\mathbf{14623.45 \pm 612.10}$ | $\mathbf{6798.39 \pm 196.84}$ | $\mathbf{6967.54 \pm 317.07}$ |

## E.3 Implementation details of CMLO

**Modeling and learning the dynamical models.** As inferred from the optimization objective, the minimization of the objective function can be achieved when we try to minimize the difference between $M_2$ and the real environment. To reduce model bias, we chose to use NLL as a loss function in our implementation, which has been shown an effective way to learn model dynamics. More specifically, CMLO adopts a bootstrap ensemble of dynamical models $\{\hat{f}_{\phi_1}, \hat{f}_{\phi_2}, \ldots, \hat{f}_{\phi_K}\}$. Specifically, each forward dynamical model $f_{\phi_i}$ approximates the transition function of the real environment, that is $\hat{s}_{t+1} \sim f_{\phi_i}(s_t, a_t)$. The probabilistic models are fitted on shared but differently shuffled replay buffer $\mathcal{D}_e$, and the target is to optimize the Negative Log Likelihood (NLL).

$$\mathcal{L}^H(\phi) = \sum_t^H [\mu_\phi(s_t, a_t) - s_{t+1}]^T \Sigma_\phi^{-1}(s_t, a_t)[\mu_\phi(s_t, a_t) - s_{t+1}] + \log \det \Sigma_\phi(s_t, a_t)$$

And the prediction for these ensemble models is, $\hat{s}_{t+1} = \frac{1}{K}\sum_{i=1}^{K} f_{\phi_i}(s_t, a_t)$. More details on network settings are presented in Table 4.

**Model shifts estimation.** Recall that we partition the incalculable model shifts into two components for estimation, one for state-space coverage and the other for model divergence.

- *state-space coverage*. State coverage (policy coverage) is the range of state spaces that our algorithm can explore in the real environment under the current policy $\pi_i$ (derived from the learned model $M_i$). In the existing works, [1] defined the return set for two state sub-space as $\bar{R}_{ret} = \lim_{n\to\infty} R_{ret}^n(X, \bar{X})$, where $R_{ret}^n(X, \bar{X})$ means an n-step returnability from $X$ to $\bar{X}$. Referring to this definition, the state coverage of $\pi_i$ can be defined as $\mathcal{S}_{pc}^{\pi_i} : \forall s \in \mathcal{S}_{pc}^{\pi_i}, a \sim \pi_i(\cdot|s), s' \sim P(\cdot|s, a) \in \mathcal{S}_{pc}^{\pi_i}$. Besides, in the description of La Salle's Invariance Principle [2], we verify the equivalence of Invariant Set and state coverage. Intuitively, the Humanoid example in our response to your major concerns also shows that the variation of state coverage in the different training stages.

  We estimate the policy coverage (state-space coverage) by computing the volume $vol(\mathcal{S}_{\mathcal{D}})$ of the convex closure $\mathcal{S}_{\mathcal{D}}$ constructed on the replay buffer $\mathcal{D}$. Since estimation on the full historical experiences involves a huge computational burden, we instead sample $N$ tuples (*e.g.* 1000 tuples) from the replay buffer upon each estimation. As for the convex hull, we first perform Principal Component Analysis on the states to reduce the dimension and then leverage the Graham-Scan algorithm to construct a convex hull of these $N$ points, which only takes $\mathcal{O}(N \log N)$ for time complexity.

- *model divergence*. We estimate the model divergence by computing the average prediction error on newly encountered data. Upon it, we get the estimation for the model divergence from the $K$ ensemble models, $\mathcal{L}(\Delta\mathcal{D}) = \mathbb{E}_{(s,a,s')\in\Delta_{\mathcal{D}}}\left[\frac{1}{K}\sum_{i=1}^{K}\|s' - \hat{f}_{\phi_i}(s, a)\|\right]$.

**Event-triggered mechanism.** Recall our proposed optimization problem:

$$\min_{\substack{M_2\in\mathcal{M}\\\pi_2\in\Pi}} \mathbb{E}_{s,a\sim d^{\pi_2}} \left[ \sum_{s'\in\mathcal{S}} |P(s'|s, a) - P_{M_2}(s'|s, a)| \right],$$

$$s.t. \quad \sup_{s\in\mathcal{S}, a\in\mathcal{A}} \mathcal{D}_{\text{TV}}(P_{M_1}(\cdot|s, a)\|P_{M_2}(\cdot|s, a)) \leq \sigma_{M_1, M_2}.$$

We design an event-triggered mechanism to determine the interval instant $\tau$ on the condition that the optimization problem is solved at step $t + \tau$. The mechanism is developed based on the difference between the current model $M_1$ (trained at step $t$) and the upcoming model $M_2$, which is estimated on the newly encountered data. If their model shifts reaches a certain value, it stands to reason that a new dynamic model is required to be trained. Thus, the event-triggered mechanism is based on the condition:

$$\frac{vol(\mathcal{S}_{\mathcal{D}_t\cup\Delta D(\tau)})}{vol(\mathcal{S}_{\mathcal{D}_t})} \cdot \mathcal{L}(\Delta D(\tau)) \geq \alpha.$$

Here, we adopt the fraction form for the triggered condition. Denominator $vol(\mathcal{S}_{\mathcal{D}\cup\Delta\mathcal{D}(\tau)}) \cdot \mathcal{L}(\Delta\mathcal{D}(\tau))$ is used to obtain an estimation for the model shift, as detailed in Line 239-248. The numerator $vol(\mathcal{S}_{\mathcal{D}})$, on the one hand, is to reduce numerical errors; on the other hand, this fraction reflects the relative change of the policy coverage and model shift if we turn to train $M_2$ under different $\tau$ starting from $M_1$. This fraction reflects the current ability to digest new data. It can facilitate the setting of threshold, for we do not need to tune $\alpha$ once the policy coverage updates.

In CMLO, the condition estimation execute per $F$ steps because excessively frequently estimation doesnt make huge difference but bring up the computation load. In order to reuse the result of intermediate computation, we apply a log value to the result of each estimation so that the log value condition function can be approximated by the sum of each estimation value within the interval. We additionally append a constant $\beta = 1.0$ for the penalty of accumulated interval steps.

$$\sum_{i=0}^{[\tau/F]} \log\left(\frac{vol(\mathcal{S}_{\mathcal{D}_t\cup\Delta\mathcal{D}(Fi)})}{vol(\mathcal{S}_{\mathcal{D}_t})} \cdot \mathcal{L}(\Delta\mathcal{D}(Fi)) + \beta\right) \geq \alpha$$

*Remark 1:* We observe from the event-triggered condition that the interevent time can be enlarged by increasing the threshold $\alpha$, which implies more exploration samples will be collected by current policy and the optimization objective will be solved less frequently. In other words, the triggered threshold is environment-specific. Notably, the tuning load required for our event-triggered mechanism is not heavier than in those fixed settings, due to that only a hyperparameter $\alpha$ is introduced for model training frequency in CMLO, while those algorithms with fixed settings need to tune the fixed model training interval. We claim that it is crucial to dynamically adapt the numbers of explorations to update the model according to the current training and exploration status.

*Remark 2:* Zeno behavior [17] is common in the event-triggered mechanism, which leads to a most frequently triggering. The zeno behavior is naturallt alleviated by the introdution of $\beta$, which acts as a penalty for interevnet time. We addition-

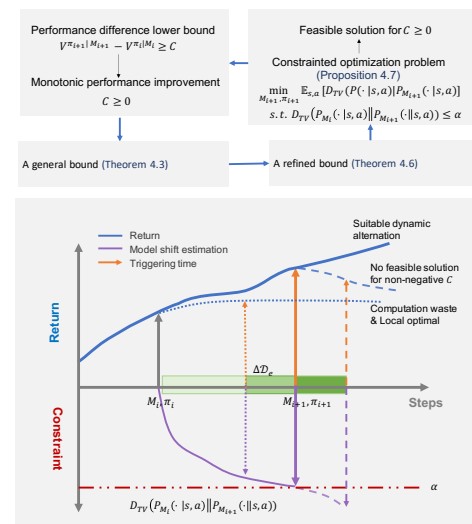

Figure 7: Illustration of the proposed event-triggered mechanism.

aly add the lower and upper bounds of the interevent time to further keep the interval in a safe zone to aviod zeno behavior caused by some extreme situations. The minimal and maximal interevent time are given by $\inf\{\tau\} = \underline{T}$ and $\sup\{\tau\} = \overline{T}$.

**Policy optimization and model rollouts.** We can adopt a standard off-policy model-free RL method SAC [16] as the policy optimization oracle of CMLO. Another key concern is the way of training data generation. We adopt the truncated short model rollouts strategy inspired by some current MBRL works [20, 39, 28], which helps to escape from compounding error and encourage model usage. The main difference from the general rollouts mechanism is that we restrict our rollouts to be generated from fresh models, rather than using outdated models to generate rollout data as MBPO [20] and AutoMBPO [28] do in their implementations. Based on the dataset $\mathcal{D}_m$ of the fresh model rollouts, we perform SAC. In the policy evaluation step, SAC repeatedly apply a Bellman backup operator $\mathcal{T}^\pi$ to the soft Q-value, $\mathcal{T}^\pi Q^\pi(s,a) \triangleq r(s,a) + \gamma \underset{s'}{\mathbb{E}}[V^\pi(s')]$, and in the policy improvement step, SAC updates the policy according to $\pi = arg\min_{\pi \in \Pi} \mathbb{E}_{s_t \in \mathcal{D}_m} \mathcal{D}_{KL}(\pi(\cdot|s_t) \| \exp(Q^\pi - V^\pi))$.

*Remark 3:* The data distribution introduced by the outdated model has a drift from the data distribution introduced by the fresh model. This data shift will somehow mislead policy training and, in addition, the policy trained on the outdated model suffers from the limited sampling coverage during interacting with the real environment, which might in return cause the following models to fall into a local trap. In other words, the less the model differs from the real dynamics, the data it rolls out is more valuable.

### E.4 Hyperparameters

**Hyperparameters for Main Experiments.** Table 4 lists the hyperparameters used in training CMLO. Here, $x \to y$ over epochs $a \to b$ denotes a threshold linear function, *i.e.*, at epoch $t$, $f(t) = \min(\max(x + \frac{t-a}{b-a} \cdot (y-x), x), y)$.

**Hyperparameters for Ablation Studies.** Note that other hyperparameters we do not mention below are the same as hyperparameter settings in Table 4.

**(1) Policy optimization oracle: TRPO.** For the TRPO part, the key parameters are listed below:

- Ant: $horizon = 1000, \gamma = 0.99, gae = 0.97, step\_size = 0.01, iterations = 40$

- HalfCheetah: $horizon = 1000, \gamma = 0.99, gae = 0.95, step\_size = 0.01, iterations = 40$

Table 4: Hyperparameter Settings for CMLO.

| | Hopper | Walker | Swimmer | HalfCheetah | Ant | Humanoid |
|---|---|---|---|---|---|---|
| epochs | 300 | 125 | 300 | 300 | 400 | 250 |
| environment steps per epoch | 1000 | | | | | |
| dynamical models network | Gaussian MLP with 4 hidden layers of size 200 | | | | | |
| ensemble size | 5 | | | | | |
| model rollouts per policy update | 400 | | | | | |
| rollout schedule | $1 \to 15$ over epochs $20 \to 100$ | 1 | | | $1 \to 25$ over epochs $20 \to 100$ | $1 \to 25$ over epochs $20 \to 300$ |
| SAC policy network | Gaussian with hidden size 512 | | | | Gaussian with hidden size 1024 | |
| policy updates per step | 40 | 20 | 20 | 10 | 20 | 20 |
| event-triggered threshold $\alpha$ | 1.2 | 3.0 | 2.0 | | | 2.5 |
| computing frequency $F$ | 20 | 50 | | | | |
| minimal interevent time $\underline{T}$ | 150 | | | | | |
| maximal interevent time $\overline{T}$ | 500 | | | | | |

About Legend w/o-n, we use a data sampler with batchsize=20, thus we get $20 \times n$ real interactions during the model training interval. We compute the total triggered times and scale them to [0,1], which is shown in the bar plots.

**(2) Policy optimization oracle: iLQR.** Dynamical models network: Gaussian MLP with 3 hidden layers of size 200, batch size is 64, and the learning rate is 0.0001. For the iLQR part: $LQR\_ITER = 10, R = 0.001, Q = 1, horizon = 5$. About Legend w/o-n, we get n real interactions during the model training interval. And $\alpha = 0.5$ in w/-ours. We compute the total triggered times and scale them to $[0, 1]$, as shown in the bar plots.

### E.5 Additional Ablation Study

**Estimation on model shifts.** The constraint function based on model shifts is incalculable due to the unobserved model $M_2$. We design a practical predictor for the model shifts by computing the state-space coverage and the model prediction error. Also, the decoupling of the constraint and the objective is enabled partly owing to the slightly overestimation over the true value. Recall our constraint function:

$$\mathcal{D}_{\mathrm{TV}}(P_{M_1}(\cdot|s,a)\|P_{M_2}(\cdot|s,a)) = \frac{1}{2} \sum_{s' \in \mathcal{S}} \left[ |P_{M_1}(s'|s,a) - P_{M_2}(s'|s,a)| \right]$$

$$\leq \sum_{s' \in \mathcal{S}} \frac{1}{2} \left[ |P_{M_1}(s'|s,a) - P_M(s'|s,a)| + |P_{M_2}(s'|s,a) - P_M(s'|s,a)| \right]$$

The updated dynamics $P_{M_2}$ usually comes closer to the true dynamics than the previous one $P_{M_1}$, thus we turn to estimate $\sum_{s' \in \mathcal{S}}[|P_{M_1}(s'|s,a) - P_M(s'|s,a)|]$. Once the model $M_2$ is trained, we can actually conduct a more realistic calculation for the constraint function. To show the connection between our predicted value in the absence of $M_2$ and the estimated value after obtaining $M_2$, we perform experiments on four environments and results are shown in Figure 8. The results demonstrate that our prediction is higher than the true estimation and their trends stand consistent, which indicates that our predictor is well designed. This gap helps to decouple the constrained optimization problem, and this overestimation part can be bridged by adjusting the $\alpha$.

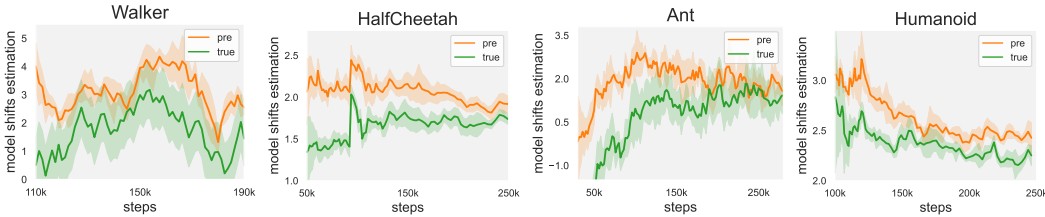

Figure 8: Estimation of model shifts in four environments. Green lines imply the estimation on model shifts after updating, which could be considered as a real value of our constraint function. And orange lines show our pre-estimation on model shifts before model updating.

**Model accuracy.** Figure 9 shows the one-step model error during the training under four benchmark tasks. We find that CMLO achieves a more accurate model than the state-of-the-art baseline MBPO. This result agrees with our insight that, a smarter scheme to choose different numbers of explorations at different steps instead of the unchanged setting in current methods, will promote a better model.

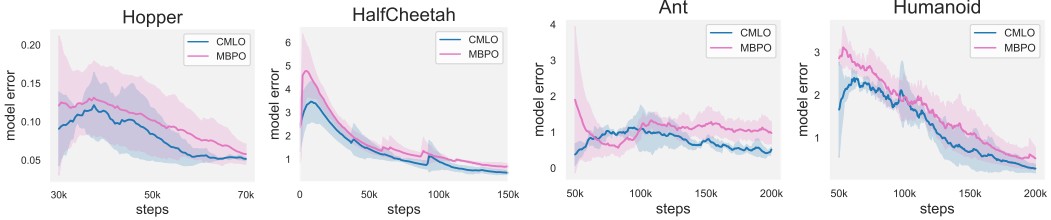

Figure 9: One-step model error in four benchmark tasks.

**Policy Coverage Comparison.** Policy coverage represents the exploration ability of the policy. The policy coverage increasing with the stages means that the policy has new explorations at every stage and may not fall into a local optimum. Here, we present the numerical comparison to MBPO in Table 5.

Table 5: Policy Coverage Comparison to MBPO in different stages.

| | | Stage1 | Stage2 | Stage3 | Stage4 | Stage5 |
|---|---|---|---|---|---|---|
| HalfCheetah | CMLO | 138.57 | 182.28 | 243.47 | 302.82 | 344.36 |
| | MBPO | 129.25 | 173.09 | 242.49 | 264.85 | 338.55 |
| | | Stage1 | Stage2 | Stage3 | Stage4 | Stage5 |
| Ant | CMLO | 354.15 | 744.92 | 849.47 | 876.12 | 909.80 |
| | MBPO | 342.13 | 729.30 | 821.66 | 864.93 | 880.25 |

Here, each stage $i$ contains $(60 \times (i-1), 60 \times i]k$ steps. In HalfCheetah, we find that our policy achieves higher coverage especially in first 4 stages than MBPO. Consistently, we find that our

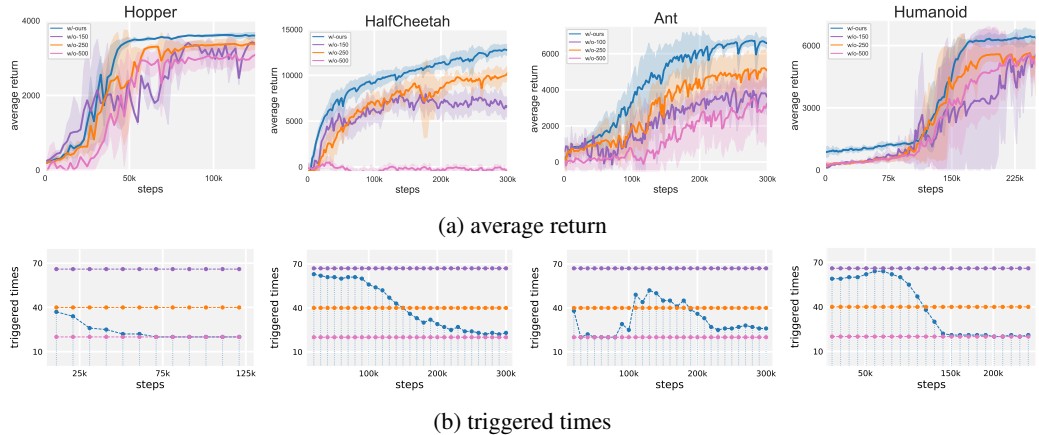

(a) average return

(b) triggered times

Figure 10: Ablation on event-triggered mechanism. (a,c) shows the average return with or with-out event-triggered mechanism in HalfCheetah and Ant benchmarks. (b,d) shows the average number of triggered times per 10k step. All the experiments are average over 4 random seeds.

policy enjoys higher performance, with an average return lead of about 1855.29 over MBPO in the first 300k steps. Likewise, the growth of policy coverage in Ant is also consistent with the rise in average return. The increase in policy coverage helps the policy to refrain from falling into a local optimum, thus improving performance.

**Effectiveness of event-triggered mechanism.** We compare applying model shift constraints to the unconstrained cases and the results are shown on Figure 10. To verify the effectiveness of adding suitable constraints, we invalidate the event-triggered mechanism and keep the other part unchanged in our method. As observed, our model is accurate enough when fixing the model training interval at 250, but it still performs worse than applying the model shift constraints. We attribute our model's out-performance to our rational model shift design. It improves the performance while minimizing the training cost of the model. Adding such a model shift can protect the model from overfitting on under-explored data, and can also save the model from fitting large data shifts.

To determine whether the event-triggered mechanism has an effect on the training process, we conduct a t-test to compare the average returns of CMLO with or without the mechanism. We compare the original CMLO to its variant with a fixed setting (w/o-250) and list the p-values in Table 6. Our p-values are much smaller than 0.05, so we say with a high degree of confidence that the smartly choosing dynamically varying number of explorations does make a difference in the overall performance.

Table 6: t-test to the average returns of CMLO with or without event-triggered mechanism.

|         | Hopper | Walker   | Swimmer | HalfCheetah | Ant      | Humanoid  |
|---------|--------|----------|---------|-------------|----------|-----------|
| p-value | 0.0141 | 4.74e-10 | 2.47e-5 | 7.48e-33    | 2.78e-26 | 1.983e-15 |

Besides, we provide visualization of event-triggered mechanism on HalfCheetah and Ant in Figure 11. The y-axis is our estimation of the triggered condition. This figure shows the constraint estimation and whether it reaches the triggered threshold (when the peak is above the threshold (dashed line), the primary trigger condition is satisfied) within different stages. Note that in the paper we have shown 4k steps for each stage, and here we present for the whole 60k.

## E.6 Computing Infrastructure

Table 7 lists our computing infrastructure and the corresponding computational time used for training CMLO on the six benchmark tasks.

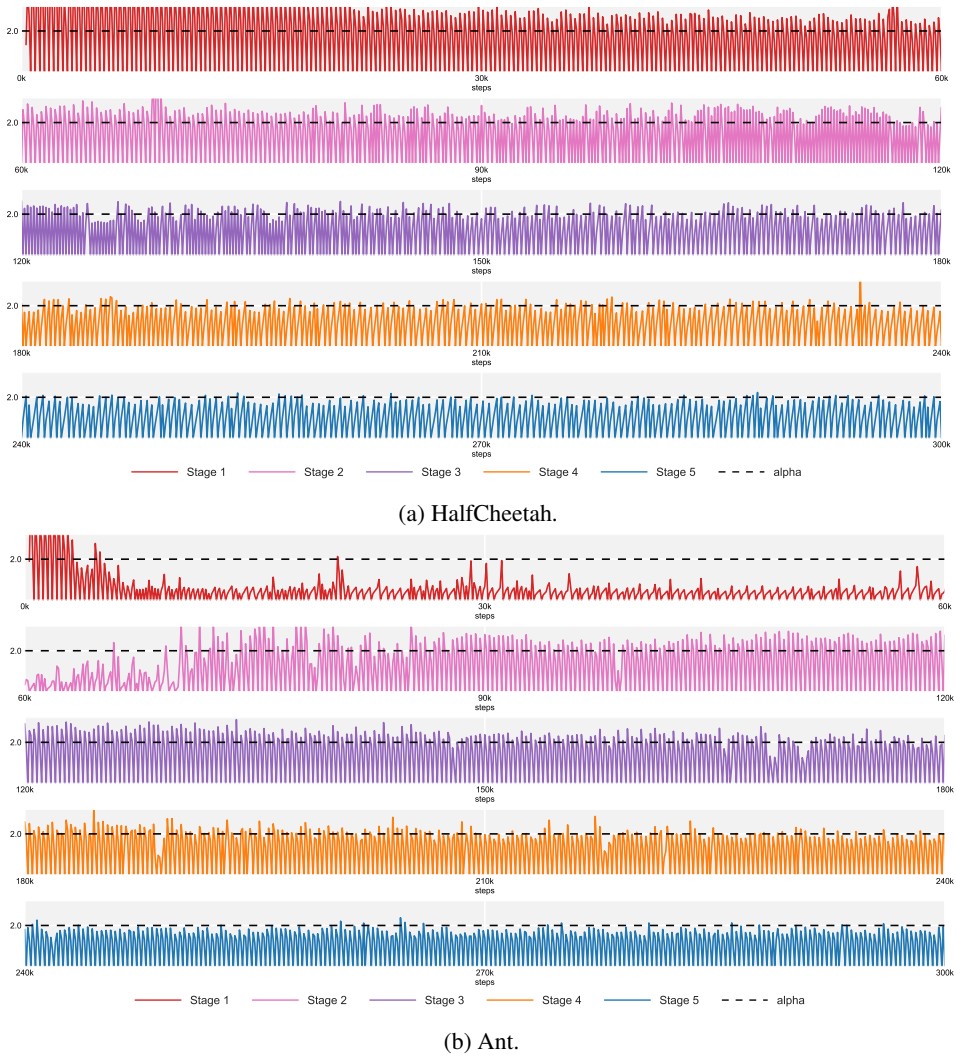

(a) HalfCheetah.

(b) Ant.

Figure 11: Visualization of event-triggered mechanism on HalfCheetah and Ant. Solid lines show the model shift estimation and dotted lines are the triggered threshold. Note that here we apply log value.

Table 7: Computing infrastructure and the computational time for each benchmark task.

|  | Hopper | Walker | Swimmer | HalfCheetah | Ant | Humanoid |
|---|---|---|---|---|---|---|
| CPU | Intel Core i7-6900K (16 threads) | | | | | |
| GPU | NVIDIA TITAN X (Pascal) x 3 | | | | | |
| computation time in hours | 20.15 | 19.21 | 31.58 | 35.97 | 29.35 | 33.31 |