# OpenReview forum: "When to Update Your Model: Constrained Model-based Reinforcement Learning"
_NeurIPS.cc/2022/Conference — NeurIPS 2022 Accept_

### Official Review · Reviewer_3kr9 · 2022-07-09

**Rating:** 5
**Confidence:** 5
**Soundness:** 3 good
**Presentation:** 2 fair
**Contribution:** 2 fair

**Summary:**

This paper studies the relationship between the shift brought by model updates and policy performance. The authors proposed a model shift constraint and the CMLO algorithm for a monotonic improvement guarantee.

**Questions:**

1. It would be better to have more comparisons between CMLO and previous works (e.g. DPI). Although they have different proof structures (from the local and global view?), the monotonic improvement results are similar in my view.

2. Why do we care about the *optimal* value under a model and what is the advantage compared to the local view? I am assuming that it is the primary reason that leads to different algorithms between CMLO and previous works.

3. The definition of state coverage and the event-triggered equation is not so clear.

My concerns are closely related. So it might be the case that I missed something important. I would like to change my score depending on the authors' rebuttal.

**Limitations:**

No.

**Strengths And Weaknesses:**

pros: 1. The authors consider an important problem in RL, the inconsistency of policy updates under shifted models. The main results and proofs are largely correct in my view.

2. The ablations verify the proposed algorithm.

cons: 1. The monotonic results are not novel in MBRL, for example, the authors cite DPI [1], which established similar monotonic results. But the authors claim that previous works "characterize the monotonicity in terms of a fixed model of interest". I don't see the reason for such a claim and the exact advantage of CMLO.

2. From the introduction and related work, the partial reasons for cons 1 might include that the authors derive the theorems from a global optimal view. But I can't judge the necessity of doing this. E.g., in theorem 4.5, why do we care about the *optimal* value under a model? Notably, the monotonic property rather than global optimality is of interest. Therefore, according to the performance difference lemma, shouldn't we focus more on the value-under-model evaluated using the *current policy*?

3. Another confusion I have is the notation of state coverage. As it is introduced from a global point of view, I can't judge its necessity before understanding the necessity of the global view. Besides, it is better to include the exact definition of state coverage in the context of RL, so that we can see why it is estimated with the replay buffer.

4. The intuition behind the event-triggered equation 5.1 is also not very clear. What will the fraction of state coverage give?

Minor: how much additional time does it cost for estimating model shifts?

[1] Dual Policy Iteration, Wen Sun et al.

---

> ### Author Response · Authors · 2022-08-02
> **Response to Reviewer 3kr9 (Q4-Q6)**
>
> > **Q4:** "Another confusion I have is the notation of state coverage. As it is introduced from a global point of view, I can't judge its necessity before understanding the necessity of the global view. Besides, it is better to include the exact definition of state coverage in the context of RL, so that we can see why it is estimated with the replay buffer."
>
> Definition: State coverage (policy coverage) is the range of state spaces that our algorithm can explore in the real environment under the current policy $\pi_i$ (derived from the learned model $M_i$). In the existing works, [1] defined the return set for two state sub-space as $\overline R_{ret} = \lim_{n\rightarrow \infty} R^n_{ret} (X,\bar{X} )$,
>
>
> where $R_{ret}^n(X,\bar X) $ means an n-step returnability from $X$ to $\bar X$. Referring to this definition, the state coverage of $\pi_i$ can be defined as $\mathcal S_{pc}^{\pi_i}: \forall s\in \mathcal S_{pc}^{\pi_i}, a\sim\pi_i(\cdot|s), s'\sim P(\cdot|s,a)\in\mathcal S_{pc}^{\pi_i}$. Besides, in the description of La Salle's Invariance Principle [2], we verify the equivalence of Invariant Set and state coverage. Intuitively, the Humanoid example in our response to your major concerns also shows that the variation of state coverage in the different training stages.
>
> Necessity: As we discussed before, state coverage reflects the ability of policy exploration, which affects the final optimal value. When state coverage is improved, the policy $\pi_n$ can obtain more unseen samples in the exploration phase from the real environment, which further improves the model accuracy of $M_{n+1}$ and the optimization value of the derived policy $\pi_{n+1}$. This is also reflected by the consistent change in our Figure 3(a) state coverage and the performance in Figure 1.
>
> Estimation: We use the replay buffer $\mathcal{D}$ to store the explored samples by policy $\pi_n$ from the real environment. Notice that these samples are a subset of the policy coverage $\mathcal{S}_{pc}^{\pi_i}$,  thus we can utilize $\mathcal{D}$ to estimate the state coverage as a practical implementation.
>
> Sorry for the insufficient explanation, and we have polished our description of state coverage per your concerns in the rebuttal revision.
>
> [1] Wachi A, Sui Y. Safe reinforcement learning in constrained Markov decision processes[C]//International Conference on Machine Learning. PMLR, 2020: 9797-9806.
>
> [2] Slotine J J E, Li W. Applied nonlinear control[M]. Englewood Cliffs, NJ: Prentice hall, 1991.
>
>
> > **Q5:** "The intuition behind the event-triggered equation 5.1 is also not very clear. What will the fraction of state coverage give?"
>
> About intuition: Equation 5.1 is the event-triggering condition motivated by Proposition 4.7. Proposition 4.7 implies that, when the model shift constraint boundary is touched according to the estimation, we need to pause data collecting and turn to solve the optimization objective. Besides, although turning to train models once not to violate the constraint is theoretically reasonable, we avoid doing so in practice because performing an update on data with a minor shift in coverage and distribution is wasteful and may risk overfitting (Line 250-253).
>
> About fraction: The numerator $vol (\cal S_D)$, on the one hand, is to reduce numerical errors; on the other hand, this fraction reflects the relative change of the policy coverage and model shift if we turn to train $M_2$ under different $\tau$  starting from $M_1$. This fraction reflects the current ability to digest new data. It can facilitate the setting of threshold $\alpha$, for we do not need to tune $\alpha$ once the policy coverage updates.
>
> > **Q6:** "How much additional time does it cost for estimating model shifts?"
>
> Actual computation time: compared to the unconstrained cases (ablation studies in Figure 2), our total training time increased by an average of 3.24h in the HalfCheetah environment (300k steps) and 3.96h in the Ant environment (300k steps). And our computing infrastructure and CMLO computational time was listed in Appendix Table 5.
>
> Time complexity analysis:
>
> * For state-space coverage: we perform Principal Component Analysis to reduce the dimension and then leverage the Graham-Scan algorithm to construct a convex hull of these $N$ points, which only takes $O(N \log N)$ for time complexity.
>
> * For model divergence: We estimate the model divergence ($K$ ensemble models) by computing the average prediction error on $N$ newly encountered data, which only takes $O(KN)$ for time complexity.
>
> Finally, we hope we resolve all of your concerns and will continue to polish our language and the clarity in our revision. Thanks again for your comments and we wish you could reconsider your score.

---

> ### Author Response · Authors · 2022-08-02
> **Response to Reviewer 3kr9 (Q1-Q3)**
>
> We have detailed above the advantages of our theory, and the comparison with previous approaches. We now respond to your other concerns one by one.
>
> > **Q1:** "The monotonic results are not novel in MBRL, for example, the authors cite DPI [1], which established similar monotonic results. But the authors claim that previous works "characterize the monotonicity in terms of a fixed model of interest". I don't see the reason for such a claim and the exact advantage of CMLO."
>
> Our theoretical analysis is quite different from DPI, we have a different scheme, assumptions, proof structures, and results. For example, DPI forces $\pi_n$ and $\pi_{n+1}$ to be close, while we do not have such assumptions. Instead, we derive under the policy optimization oracle, which requires the $\pi_n$ to be $\epsilon_{opt}$-optimal under its corresponding model $M_n$.  Under Dyna-style MBRL, our assumptions are more relaxed. Our theory framework allows for the replacement of different policy optimization algorithms, and thus we are free to exploit the advantages of advanced model-free algorithms.
>
> We have detailed our advantages over DPI  in the response to your **major concerns**. Besides, the practical instance of DPI may fail if the reward function could not be approximated by the quadratic function.  DPI focused on the policy optimization in a fixed model, cf. Theorem 3.1,  is why we call it "a fixed model perspective".
>
> > **Q2:** "It would be better to have more comparisons between CMLO and previous works (e.g. DPI). Although they have different proof structures (from the local and global view?), the monotonic improvement results are similar in my view."
>
> Thanks for your suggestion. In our response to your **major concerns**, we elaborate on the differences between our work and the previous two classes of work on monotonicity. We are distinct from them in terms of scheme, assumptions, proof structures, results, and guidance to the algorithm.  And we have included the comparison in our rebuttal revision.
>
> > **Q3:**  "Why do we care about the *optimal* value under a model and what is the advantage compared to the local view?  But I can't judge the necessity of doing this. E.g., in theorem 4.5, why do we care about the *optimal* value under a model? Notably, the monotonic property rather than global optimality is of interest.  According to the performance difference lemma, shouldn't we focus more on the value-under-model evaluated using the *current policy*?"
>
> We have detailed the advantage of the global view analyses in our response to your **major concerns**. It has rarely been explored in previous MBRL.
>
> As we discussed above, in MBRL, model quality is the bottleneck of policy eventual performance. The sub-optimal value under a model is a novel and effective way to measure the model quality. Therefore, our analysis helps to guide the model improvement, whereas previous local view monotonicity analysis is hard to do so.
>
> Besides, focusing on the optimal value could be regarded as marginating on policy iteration within a model so that our analysis is generic under Dyna-style without worrying about different policy optimization methods (as shown in our ablation study, see Figure 5.). Owing to our proposed theoretical scheme, our algorithm does not conflict with the local view, and we marginate the policy iterations under a fixed model by our policy optimization oracle. And it allows us to employ many local view methods to improve the monotonicity of single-step policy iterations.
>
> Finally, as we analyzed in our response to your major concerns, the inevitable model shifts in MBRL have a significant impact on policy performance and trajectory bias. Those works that care only about "value-under-model evaluated using the current policy”(e.g. MBPO), usually give an upper bound on the model bias, which is a crude approach and may lead to monotonicity equations that do not have feasible solutions.

---

> ### Author Response · Authors · 2022-08-02
> **Response to Reviewer 3kr9 (Major Concerns 2.2)**
>
> (2) **Discrepancy bound class:**  They derive upon $V^{\pi_n}(\mu)\geq V_M^{\pi_n}(\mu) - C(\epsilon_m, \epsilon_\pi)$.  As guaranteed in them,  once a policy update $\pi_n \rightarrow \pi_{n+1}$ has improved returns under the same model $M$, i.e., $V_{M}^{\pi_{n+1}}(\mu) > V_{M}^{\pi_n}(\mu) + C(\epsilon_m, \epsilon_\pi)$ , it would improve the lower bound on the performance evaluated in the real environment, i.e., $\inf\{V^{\pi_2\vert M}(\mu)\} > \inf \{ V^{\pi_1\vert M(\mu)}\} $\}.
>
> Their theory is based on a fixed model $M$, or an upper bound on the distribution shift of all models $\epsilon_m$.  It does not concern the change in model dynamics during updating, nor the performance varying due to the model shift. Moreover, The solution would be very coarse if only the upper bound of the model shift is given. Even worse, the given upper bound is likely to be too large, then it will fail to find a feasible solution for $V_M^{\pi_{n+1}}(\mu) - V_M^{\pi_n}(\mu) \geq C(\epsilon_m, \epsilon_\pi)$ in practice, thus making the monotonicity guarantee fails.
>
> In summary, our proposed theoretical framework provides a new perspective on model-based RL monotonicity which considers the entangled nature of model learning and policy optimization. It will be useful for guiding the optimization model updates and might be helpful to understand several perspectives of model-based RL that have been rarely explored before.
>
>  [1] Li H, Shi Y. Event-triggered robust model predictive control of continuous-time nonlinear systems[J]. Automatica, 2014.
>
>  [2] Sham Kakade and John Langford. Approximately optimal approximate reinforcement learning.In ICML, 2002.
>
>  [3] Thanard Kurutach et al. Model-ensemble trust-region policy optimization. In ICLR, 2018.
>
>  [4] Wen Sun et al. Dual policy iteration. In NeurIPS, 2018.
>
>  [5] Michael Janner et al.When to trust your model: Model-based policy optimization. In NeurIPS, 2019.

---

> ### Author Response · Authors · 2022-08-02
> **Response to Reviewer 3kr9 (Major Concerns 2.1)**
>
> **2. Detailed Analysis**:
> To begin with, an important fact is that the effect of model shifts on trajectories is drastic. For example, even when the system dynamics satisfy $L$- Lipschitz continuity, along with the policy and the initial state be the same,  the difference in trajectories sampled in $M_1, M_2$ grows at $e^{LH}$ with the length $H$ of the trajectory [1]. As the model shifts decay, the trajectory discrepancy will also decrease sharply. It implies that model shift stays a substantial influence during the MBRL training process. There are two main trends of local view analysis (for readability, the conclusions in previous work will be rewritten with the notation of our paper).
>
> (1) **API [2] class:**  Their recipe for monotonicity analysis is $V^{\pi_{n+1}} (\mu)-  V^{\pi_n}(\mu) \geq C(\pi_n, \pi_{n+1}, \epsilon_m)$. If policies update $\pi_n\rightarrow \pi_{n+1}$  could provide a non-negative $C(\pi_n, \pi_{n+1}, \epsilon_m)$ , then the performance is guaranteed to increase.  Here, $\epsilon_m = \max_{\pi\in \Pi, M\in \mathcal M} E_{s,a\sim d^{\pi}}[\mathcal D_{TV}(P(\cdot\vert s,a)\Vert P_M(\cdot\vert s,a))]$ .
> * Most previous works ([2] [3]) were derived under model-free settings ($\epsilon_m=0$) through conservative policy iteration, e.g., by forcing $\mathcal D_{TV}(\pi_n\Vert \pi_{n+1})\leq \alpha$), then the state-action distribution are close as well $\mathcal D_{TV}(d^{\pi_n}\Vert d^{\pi_{n+1}})\leq \frac{\alpha\gamma}{1-\gamma} $, so that they can optimize over their performance difference lemma  $C(\pi_n, \pi_{n+1}, 0) \approx \frac{1}{1-\gamma}E_{s,a\sim d^{\pi_n}}[A^{\pi_n}(s,a)] $.
> * When $\epsilon_m>0$, this approximation $C(\pi_n, \pi_{n+1}, \epsilon_m) \approx \frac{1}{1-\gamma}\mathbb{E}_{s,a\sim d^{\pi_n}}[A^{\pi_n}(s,a)] $ fails.
>
> * **DPI**: Firstly, DPI [4] focused on policy optimization in a fixed model, cf. Theorem 3.1, is why we call it "a fixed model perspective".
>
> Second, DPI tries to force $\pi_{n+1}$ and $\pi_n$ to be close, which will result in a high similarity of the data sampled. Then a risk arises from it, this approach would limit the growth of the policy exploration in the real environment, thus leading the inferred models to stay optimized in a restrictive local area. For example, in the Humanoid environment, the agent struggles to achieve balance at the beginning of training. An updated restricted policy will cause the exploration space to be limited in such an unbalanced distribution for a long time, and the learned model in such highly repetitive data will converge quickly with a validation loss be zero. However, the success trajectory has not been explored yet, causing both the policy and the learned model to fall into a poor local optimum.
>
> Besides, the definition of model accuracy (Eq.3) is a local view in DPI, i.e., $\hat{P}$ is $\delta$-opt under $d^{\pi_n}$. If we replace model accuracy with a more general, global definition (for example, $\hat{P}$ is $\delta$-opt under $d^{\pi_{n+1}}$ , or $\hat{P}$ is $\delta$-opt under all $(s,a,s')$ tuples), we find that the $\delta$ in (Eq. 3) will be large at the initial steps, making it difficult to obtain a local optimal solution in Theorem 3.1. Finally, theoretical analysis in DPI can only guide the policy iteration process, while the update of the model is passive, which is different from our global view theory.

---

> ### Author Response · Authors · 2022-08-02
> **Response to Reviewer 3kr9 (Major Concerns 1)**
>
> **Response summary.**  Thanks for your review of our work. Based on your view, it seems that you have two main concerns and several other minor concerns.
>
> 1. **The advantage of global view monotonicity**: Why do we care about the *optimal* value under a model and what is the advantage compared to the local view?
>
> 2. **Comparison between CMLO and previous works**: It would be better to have more comparisons between CMLO and previous works (e.g. DPI), although they have different proof structures.
>
> We provide clarification to your concerns as below. We appreciate it if you have any further questions or comments.
>
> Firstly, we will elaborate on the advantages of our theoretical scheme and results, and give a comparison with previous work.
>
> **Major Concerns**
>
> **1. Intuition and Summary**:
>
>    MBRL methods alternate between the two stages: model learning and policy optimization. This is a chain reaction of alternating two stages. Analyzing only the effect of one on the other is not enough. As is known, in MBRL, model accuracy often acts as the bottleneck to policy performance.
>    In previous works with the local view, they analyze the effect of model accuracy on policy performance by coarsely assuming an upper bound of model bias $\epsilon_m$.
>    However, a crucial complexity of MBRL is that besides that model accuracy affects policy quality, policy, in turn, does affect the outcome of model learning via the data collected from its interaction with the environment.
>
>    In a nutshell, the following problems are crucial in MBRL and have been less explored and not well guaranteed in the local-view works. "How does the policy affect model updating?  What is a indeed better model in MBRL? Can model-based RL algorithms be guranteed to improve the policy monotonically when considering model shifts? "
>
>    Our proposed global view monotonicity analysis focuses on the issues mentioned above and has several advantages:
>
>    * Our analysis considers how the policy exploration affects the model shifts, and then can help to improve model accuracy. However, the local view ignores the varying model shifts and crudely treats model learning as a supervised learning process independent from policy exploration.
>
>    * Our theory indicates that $M_{i+1}$ is better than $M_i$ when the performance of sub-optimal derived policy $\pi_{i+1}$ evaluated under the real environment is higher than that of $\pi_i$. In contrast, the local view  merely adopted the validation loss to measure the quality of a learned model, which is an isolated measurement ignoring the inherently entangled nature of MBRL.
>
>    * Our theory provides a monotonicity guarantee considering the varying model shifts. We will detail the drawbacks of the local view analysis due to the disregarding of model shifts as follows.

---

> ### Author Response · Authors · 2022-08-06
> **Thanks for your comment. We are willing to address further concerns.**
>
> Dear reviewer,
>
> We first thank you again for your comments and suggestions. We hope our last reply has resolved all your concerns. If you have any other questions, we are also pleased to respond. We sincerely look forward to your response.
>
> Best wishes!
>
> The authors.

---

> ### Comment · Reviewer_3kr9 · 2022-08-07
> **Thanks for the comment.**
>
> Thank the authors for the comment. It clarifies the difference between their work and previous ones. However, the advantage is still not clear. For example, regarding "when $\epsilon_m>0$, this approximation fails", $\epsilon_m$ is the model error of the policy w.r.t. a fixed (e.g. local) model, which already gives nice monotonic property when assuming access to optimization oracle (i.e. no need to consider future model, model shift etc). The proposed method seems to be the dual form of works such as DPI (i.e. not limiting policy, but limiting models).
>
> What I find interesting is that the proposed method might encourage exploring by not imposing constraints on policy updates. However, current results (monotonic improvement) are not enough for concluding efficient exploration, which can still get stuck at local optima. I suggest the authors take this into consideration, given the unclear theoretical advantage compared to existing methods.

---

> > ### Author Response · Authors · 2022-08-08
> > **Thanks! Further discussion about "effective exploration". （3/3）**
> >
> > > **Q2**: What I find interesting is that the proposed method might encourage exploring by not imposing constraints on policy updates. However, current results (monotonic improvement) are not enough for concluding efficient exploration, which can still get stuck at local optima. I suggest the authors take this into consideration, given the unclear theoretical advantage compared to existing methods.
> >
> > Thanks for your valuable suggestions!
> >
> > As far as we know, effective exploration does not mean a globally optimal solution, and the model-based RL algorithms that we have known cannot guarantee achieving a totally global optimal solution in complex scenarios.
> >
> > We illustrated in the experiments that our theory and algorithm can **promote exploration**, which is able to help us **improve local optima**. Our higher local optima can be seen from our outstanding learning curves. And here we provide a comparison in policy coverage to show our better exploration property. The policy coverage increasing with the stages means that the policy has new explorations at every stage and may not fall into a poor local optimum.
> >
> > | Env         | Algo | Stage1     | Stage2     | Stage3     | Stage4     | Stage5     |
> > | ----------- | ---- | ---------- | ---------- | ---------- | ---------- | ---------- |
> > | HalfCheetah | CMLO | 138.566126 | 182.281857 | 243.466268 | 302.816499 | 344.356213 |
> > | HalfCheetah | MBPO | 129.251405 | 173.085743 | 242.492030 | 264.853917 | 338.555574 |
> > | Ant         | CMLO | 354.154379 | 744.91538  | 849.473640 | 876.119479 | 909.798043 |
> > | Ant         | MBPO | 342.134362 | 729.295456 | 821.658472 | 864.933838 | 880.252964 |
> >
> > We hope the explanation could resolve your concerns and help to understand the advantages of our theory and algorithms. If you have further questions, we are glad to discuss them with you. Thanks again for your comments and we sincerely wish you would reconsider your score.

---

> > > ### Comment · Reviewer_3kr9 · 2022-08-08
> > > **Response to authors**
> > >
> > > Thank the authors for the clarification and detailed explanation! I decided to increase my score to borderline accept. But I still feel the studied problem is of less theoretical interest, as the single-step predictive model is relatively easy to learn in experiments (at least Mujoco), and the efficient exploration claim is not theoretically justified.

---

> > > > ### Author Response · Authors · 2022-08-08
> > > > **Thank you for your valuable suggestions!  We really appreciate the pleasant discussions!**
> > > >
> > > > Dear reviewer:
> > > >
> > > > Thank you for your reply and understanding. We will update our paper considering your and other reviewers' comments accordingly. We sincerely thank you for your comprehensive comments.
> > > >
> > > > Best wishes!
> > > >
> > > > The authors.

---

> > ### Author Response · Authors · 2022-08-08
> > **Thanks! Further discussion about "theoretical advantages". （2/3）**
> >
> > Regarding the problems mentioned above, we will show **the advantages of our theory in dealing with these problems**.
> >
> > * Towards issue 1:  When $\epsilon_m$ is large, it is useless to worry about whether performance can be improved after a single policy iteration, and it is difficult to meet the monotonicity requirements, as detailed in Issue1. At this point, model quality is the key to performance improvement.  Our theory focuses on the sub-optimal policy $\pi_i$ evaluated in each model $M_i$ and we guarantee the real performance $V^{\pi_i}(\mu)$ is non-decreasing. Thus, we can guarantee monotonicity across models, and help improve the model quality.
> >
> > * Towards issue 2:  Instead of giving a lazy upper bound of model error and then throwing out the important nature of model varying, our theory considers what the model shifts (might be seen as the difference in two model errors just for understanding) will affect the eventual corresponding policy performance.  Overall, our method models the impact of $\epsilon_m$ changes in performance and uses this impact to help optimization.
> >
> > * Towards issue 3: We provide a novel measurement of the learned model quality instead of the validation loss in previous work. As we detailed in our paper, Theorem 4.3 implies that if the model update $M_1\rightarrow M_2$ can shorten the divergence between the estimated dynamics and the true dynamics and improve the ceiling performance on the model, it may guarantee overall performance improvement under the true dynamics.  Then, we could say that $M_2$ is better than $M_1$.  Further, Proposition 4.7 guides when to update our model to keep improving the model quality and overall policy performance.
> >
> > Furthermore, note that considering the MBRL process, **the guidance of our theory to the algorithm** (especially in model optimization)is distinct and important, while previous work on local view failed to do so.  We list the guidance advantages as follows and the detailed explanation refers to our response to Q3:
> >
> > 1. Guide the model improvement.
> > 2. Guide the policy exploration.
> > 3. The policy optimization oracle allows using many local view results. Note that, we indeed have adopted the MBPO (the discrepancy bound class) results in main experiments, and the TRPO (the API class) results in the ablation study.
> >
> >
> > Besides, regarding your concern about DPI, it seems that we are in the dual direction intuitively. Our theoretical analysis is not only quite distinct from DPI, as we stated in our response to Q1, but we also have **considerable advantages over DPI**. For example, our theories are free of some strong assumptions of DPI. DPI requires that the reward function is a quadratic function, but our theory and algorithm work well under any reward function $r \in [-R, R]$.

---

> > ### Author Response · Authors · 2022-08-08
> > **Thanks! Further discussion about "the issues ignored by local view". （1/3）**
> >
> > > **Q1** "Thank the authors for the comment. It clarifies the difference between their work and previous ones. However, the advantage is still not clear. For example, regarding "when , this approximation fails",  is the model error of the policy w.r.t. a fixed (e.g. local) model, which already gives nice monotonic property when assuming access to optimization oracle (i.e. no need to consider future model, model shift etc). The proposed method seems to be the dual form of works such as DPI (i.e. not limiting policy, but limiting models)."
> >
> > Thank you very much for appreciating that our work is novel from previous works in problem modeling, theoretical structure, and algorithm design. To better address your concerns, we will explain the advantages of our theory and our algorithm based on the previous response more concisely.
> >
> > Our novel monotonicity analysis from the global view does not conflict with the local view (i.e., we marginate the monotonicity of the local view by policy optimization oracle). Notably, we aim to tackle the issues that the local view theory ignored.
> >
> > First of all, let us elaborate on **the issues ignored by local view**, and our theory tackles these problems.
> >
> > * Issue 1:
> >   * Firstly, when $\epsilon_m$ is large, the performance of the policy in the real environment $V^\pi(\mu)$ will be poor: $\epsilon_m$ means that model is quite different from the environment and the discrepancy $C(\epsilon_m, \epsilon_\pi)$ is large. Thus, even if $V_M^{\pi}(\mu)$ can be monotonic, the corresponding performance evaluated in the real environment $V^\pi(\mu)$ may be quite low for $V^\pi(\mu) \geq V^\pi_M(\mu) - C(\epsilon_m, \epsilon_\pi)$.  Secondly, when $\epsilon_m$ is large, local view analysis could not guarantee that the $V^\pi(\mu)$ is non-decreasing: it is difficult to satisfy that policy improvement in a single policy iteration is higher than $C(\epsilon_m, \epsilon_\pi)$, so $V^\pi(\mu)$ cannot be guaranteed to be monotonically increasing in the real environment either.
> >   * $\epsilon_m$ may not shrink upon model updating, as the local view cannot guide the model updating. Then, the policy performance still cannot grow in the future model, and the monotonicity across models fails.
> > * Issue 2:
> >   * An important fact, which we provided in our response to Major concerns 2.1, implies model shift or model error stays a substantial influence during the MBRL training process. Thus, it is important to consider the varying model shift. (i.e., considering future model, model shift, etc.)
> >   * Previous works gave an upper bound on the distribution shift of all models. These solutions would be very coarse if only the upper bound of the model shift given. Even worse, since the given upper bound is likely too large (refer to issue 1), it will fail to find a feasible solution considering monotonicity per policy iteration in practice, thus making the monotonicity guarantee fails.
> >
> > * Issue 3:
> >   * Model quality is the bottleneck of the MBRL algorithm performance, so it is crucial to improve model quality. However, local view analysis can only guide policy iteration but not for model updating. They crudely treat model learning as a supervised learning process independent from policy exploration.
> >   * Notably, a crucial complexity of MBRL is that besides that model quality affects policy quality, policy, in turn, does affect the outcome of model learning via the data collected from its interaction with the environment, so that a crude  supervised learning process is not enough.
> >
> > In a nutshell, it is useful to consider the future model and model shift because the monotonicity analysis given under a fixed model would fail and ignore some vital nature of MBRL.

---

### Official Review · Reviewer_99tJ · 2022-07-09

**Rating:** 6
**Confidence:** 4
**Soundness:** 2 fair
**Presentation:** 3 good
**Contribution:** 3 good

**Summary:**

The paper focuses on the monotonic improvement for model-based reinforcement learning which is an extremely important problem due to the inherently entangled nature of the multi-level optimization problem - policy optimization and model learning. Earlier research has not specifically considered the model shift which is considered in this research while proving the monotonic improvement. The primary objective is to show that $||V^{M1|\pi_1} - V^{M2|\pi_2}|| \geq C$ which can guarantee monotonic improvements under the updating dynamics. The primary reason for the model bias is a mismatch between the samples in the model learning stage and the policy optimization stage. To tackle the same, they formulate a constrained bi-level optimization framework for the MBRL problem and design an event-triggered strategy to decide when to update the model to guarantee monotonic improvement under changing dynamics. Empirical results show some improvements in sample efficiency from prior stable model-based RL methods and the ablation study supports the main hypothesis of the paper.


**Questions:**

1. Can you please illustrate how exactly is the volume of the of convex closure computed from the replay buffer?

2. Can you please point me to the equation where you derive an estimation for the model shift as a product of volume $\times$ avg prediction error?

3. Instead of going by the complicated event-triggered mechanism, if I simply try to obey the constraint in the optimization problem with a TRPO type regularized update in the model parameter space that should also do right? Given that we know that under the gaussian dynamics assumption, the total variation distance simply can be upper-bounded by the difference b/w the means and variances as well and hence it should be easy to be in the feasibility region.

4. Can you please show the derivation on how you obtain line 47 in Appendix where you mention that with Lipschitzness of $V^{\pi}_{M_1} V^{\pi}_{M_2}$ you derive the bounds for $|G^{\pi}_{M_1, M_2} (s,a)| \leq L |P_{M_2}(\cdot|s,a) - P_{M_1}(\cdot|s,a)|$.

5. Additionally in line 48 in Appendix, you derive an upper bound for value function difference in-terms of the $|P_{M_2}(\cdot|s,a) - P_{M_1}(\cdot|s,a)|$. However, it has been shown in [1] that there is a dependence of the maximum reward $R$ (and horizon $H$ which will be replaced by some factor of $\gamma$ here) intermingled/ associated with the Lipschitz constant. Basically, they prove that in the Lipschitz assumption of $||V1-V2|| \leq L ||P1- P2||$, $L$ has a dependence on $CHR_{max}$. I think that should be applicable here as well, then how does the analysis gets affected? As now there is an additional dependence on the max reward that will be added. Can you please share your thoughts on the same?

References :
[1]. Ying Fan and Yifei Ming. Model-based reinforcement learning for continuous control with posterior sampling. In Marina Melia and Tong Zhang, editors, Proceedings of the 38th International Conference on Machine Learning, volume 139 of Proceedings of Machine Learning Research, pages 3078–3087. PMLR, 18–24 Jul 2021.

**Limitations:**

The author mentions some points regarding the current applicability to certain environments and wants to scale with improved optimization methods which are sensible.

**Strengths And Weaknesses:**

The primary strength of the paper lies in the formulation of the bi-level constrained optimization objective from the lower bound objective with an event-triggered mechanism under a generative model assumption to update the model which is quite novel according to me. Earlier research has addressed the bias in the model learning and has also tried to guarantee monotonic improvement, but this paper explicitly talks about the monotonic improvement of the policy under shifted model dynamics and shows guarantees of monotonic improvements (with certain assumptions) which is interesting and novel.  The biggest challenge in guaranteeing a monotonic improvement in the policy is due to the changing dynamics which might occur due to a potential mismatch between the true trajectories and model-generated trajectories and is indeed an important challenge to overcome. The paper breaks down the source of error into 2 components with the assumption that we can always get an epsilon optimal policy under a given model which is a bit optimistic but a very common and frequently used assumption in RL. The components being 1. inconsistency gap between the model and the environment : $E_{s,a \sim d_{\pi}} TV(P(\cdot|s,a) || P_M(\cdot|s,a))$ and 2. Optimal returns under the true model and they hypothesize that the performance difference $||V^{M1|\pi_1} - V^{M2|\pi_2}||$ is lower bounded by the above two aspects. Finally, with the above notions, the authors formulate a constrained lower bound optimization problem as stated in Theorem 4.6 which is quite novel as the formulation of model-based RL as a bi-level optimization problem is quite natural as done in this paper. The strength of the paper lies in designing an event-triggered mechanism and providing a high probability bound for the value of $k$ for guaranteeing monotonic improvement under the generative model assumption. In Corollary 4.8, the authors derive a relation between the model training interval $k$, model bias, and state-space coverage which is quite unique to my knowledge and given by $k \propto \frac{2 \times vol(S)}{\epsilon^2}$ (just approximated and ignore other terms). In other words, when the model bias is less one needs to increase $k$ to increase state action coverage which is quite intuitive. Overall the ablation study seems interesting and the experimental results show some improvements over the past SOTA model-based RL methods.

The primary weakness lies in various assumptions that have been made to derive some of the primary results that are not very general. To begin with, the main result in Corollary 4.8 is shown with a generative model assumption which is quite restrictive. The primary novelty of the research from a solution perspective lies in designing the event-triggered mechanism and the theory is shown with a generative model assumption also with linear
quadratic regulator (which I still believe can be relaxed) and not for a general scenario. Secondly, it is not very clear how minimization of the constrained objective function in Proposition 4.7 turns out to be simple negative likelihood minimization. In Proposition 4.7, there is $P(s'|s,a)$ involved in the expression which can't be ignored since the $(s,a) ~ d^{\pi_2}$ and optimization variable involve $\pi_2$ as well. So, it's not clear how this is handled as we will never know $P(s'|s, a)$, and how this boils down to simple NLL minimization is not explicitly described. Also, I see in the derivations on lines 90, 97, 128, etc. the proofs involve summation over s,a which are relevant for Tabular or Discrete settings but the experiments are for continuous state-action space which causes a mismatch and some of the proofs might break when everything is expressed for continuous state-action spaces. The ablation study is interesting and the experimental results show improvement but not significant improvements. However, I feel the work addresses an important question and it might be helpful to understand this aspect for model-based RL.

---

> ### Author Response · Authors · 2022-08-02
> **Response to Reviewer 99tJ (Q7-Q9)**
>
>
> >**Q7:**  "Instead of going by the complicated event-triggered mechanism, if I simply try to obey the constraint in the optimization problem with a TRPO type regularized update in the model parameter space that should also do right? Given that we know that under the gaussian dynamics assumption, the total variation distance simply can be upper-bounded by the difference b/w the means and variances as well and hence it should be easy to be in the feasibility region."
>
> Thanks for your proposal!  We think that only applying the trust region approach is flawed.
> If we only try to solve the pure constrained optimization problem in Proposition 4.7, methods such as trust region and Lagrange relaxation could be tried.
> However, these methods are problematic when viewed in the context of MBRL's general algorithmic process and properties:
> The trust region approach directly optimizes the next model $M_2$, ignoring the inherently entangled nature of MBRL, and cannot guide policy exploration.
> Let's give two examples to illustrate this matter:
>
> * At the extreme, based on the $M_1$,  even if we do not perform any exploration, we could still perform trust region optimization and get $M_2$, which is not expected.
> * In the other case, when we collect too many novel samples, optimizing $M_2$ too late will make $M_2$ unable to adapt to the new data distribution, thus resulting in data waste.
>
> These two cases can be guided by the event-triggered approach. In addition, the event-triggered mechanism is not a very complex design, it is simple, effective, without additional training, and fits well with the MBRL process. We are happy to have further discussions if you have any other ideas and suggestions.
>
> > **Q8:** Can you please show the derivation on how you obtain line 47 in Appendix where you mention that with Lipschitzness of $V^\pi_{M_1}$, $V^\pi_{M_2}$ you derive the bounds for $\vert G^\pi_{M_1,M_2}(s,a)\vert \leq L\cdot \vert P_{M_2}(\cdot\vert s,a ) - P_{M_1}(\cdot\vert s,a)\vert $.
>
> From the definition: $G^\pi_{M_1,M_2}(s,a)= E_{{\tilde{s}'}\sim P_{M_2}(\cdot\vert s,a)}[V^\pi_{M_2}({\tilde{s}'})] - E_{s'\sim P_{M_1}(\cdot\vert s,a)}[V^{\pi}_{M_2}(s')]$.
>
> (1) Deterministic dynamics case: for clarity, we write $s' = M_i(s,a)$ instead of $s'\sim P_{M_i}(s'\vert s,a)$, then we rewrite $G_{M_1,M_2}(s,a)$ as: $G_{M_1,M_2}^\pi(s,a) = V^\pi_{M_2}(M_2(s,a)) - V^{\pi}_{M_2}(M_1(s,a))$.
>
> With the Lispchitzness, we have that $\vert G^\pi_{{M_1}, M_2}(s,a)\vert  \leq L\cdot \vert M_2(s,a) - M_1(s,a)\vert $.
>
> (2) Stochastic dynamics case:  when $L \geq \frac{R}{1-\gamma}$, we have
>
> $ \vert G^\pi_{M_1,M_2}(s,a)\vert = \vert  E_{{\tilde{s}'}\sim P_{M_2}(\cdot\vert s,a)}[V^\pi_{M_2}({\tilde{s}'})] - E_{s'\sim P_{M_1}(\cdot\vert s,a)}[V^{\pi}_{M_2}(s')]\vert $
>
> $= \vert \sum_{\tilde{s}'\in {\cal S}}(P_{M_2}(s'\vert s,a)-P_{M_1}(s'\vert s,a))V_{M_2}^\pi(s')\vert $
> $\leq \vert \max_{s'}V_{M_2}^\pi(s') \vert \cdot \vert P_{M_2}(\cdot\vert s,a ) - P_{M_1}(\cdot\vert s,a)\vert \leq L\cdot \vert P_{M_2}(\cdot\vert s,a ) - P_{M_1}(\cdot\vert s,a)\vert.$
>
> Here, we have previously checked that the $L\geq \frac{R}{1-\gamma}$ does not affect the results in our paper.
> Sorry for skipping some details of the proof here, we have refined it in the rebuttal revision.
>
> > **Q9:** "Additionally in line 48 in Appendix, you derive an upper bound for value function difference in-terms of the $|P_{M_2}(⋅|s,a)−P_{M_1}(⋅|s,a)|$. However, it has been shown in [1] that there is a dependence of the maximum reward R (and horizon $H$ which will be replaced by some factor of γ here) intermingled/ associated with the Lipschitz constant. Basically, they prove that in the Lipschitz assumption of $ \Vert V1−V2\Vert\leq L\Vert P1−P2\Vert$, $L$ has a dependence on CHRmax. I think that should be applicable here as well, then how does the analysis gets affected? As now there is an additional dependence on the max reward that will be added. Can you please share your thoughts on the same?
>
> Thanks for your exciting proposal!
>
> As discussed in our response to A8, in the case of stochastic dynamics case, $L$ indeed has a dependence on $\frac{R_{max}}{1-\gamma}$.  We agree that the analysis in [1] is applicable here as well. Our analysis is currently unaffected. As can be inferred from (R1), when $L$ varies, adjusting the threshold $\sigma_{M_1,M_2}$ will enable (R1) to be feasible.
>
> We thank the reviewer for pointing out this interesting work. We have added a discussion on the $L$ and cited this work in the rebuttal revision.
>
> [1]. Ying Fan and Yifei Ming. Model-based reinforcement learning for continuous control with posterior sampling. Proceedings of the 38th International Conference on Machine Learning, 2021.
>
> Thanks again for reading our article carefully and giving very constructive suggestions.  We hope that the above can resolve your concerns and we are glad to have further discussion.

---

> ### Author Response · Authors · 2022-08-02
> **Response to Reviewer 99tJ (Q4-Q6)**
>
> > **Q4:** “ I see in the derivations on lines 90, 97, 128, etc. the proofs involve summation over s,a which are relevant for Tabular or Discrete settings but the experiments are for continuous state-action space which causes a mismatch and some of the proofs might break when everything is expressed for continuous state-action spaces.”
>
> * Corollary 4.8 (Line 90, 97): It is derived upon the generative model setting, so it indeed need the Tabular or Discrete settings. Note that it is only used to give a feasible example of Proposition 4.7 for understanding but not guide the algorithm directly. Our algorithm is free of discrete settings.
> * Lemma C.2 (Line 128):  The proof here can turn to under the continuous state-action space setting.
>
>   $ V^\pi(\mu) - V_M^\pi(\mu) \geq  -\sum_{h=0}^{\infty} \gamma^h \vert E_{s,a\sim \rho_h^\pi(\mu;P)}[r(s,a)] - E_{s,a\sim \rho_h^\pi(\mu;P_M)}[\gamma^h r(s,a)] \vert$
> $\geq -\sum_{h=0}^{\infty} \gamma^h \int_{s\in \mathcal S}\int_{a\in {\cal A}} R\vert \rho_h^\pi(\mu; P) - \rho_h^\pi(\mu; P_M) \vert da\ ds$
>  $ =  -2R\cdot \sum_{h=0}^{\infty}\gamma^h \frac{1}{2}\int_{s\in \mathcal S}\int_{a\in {\cal A}} \vert \rho_h^\pi(\mu; P) - \rho_h^\pi(\mu; P_M)\vert da\ ds$
>
>   $= -2R \cdot \sum_{h=0}^{\infty} \gamma^h {\cal D}_{TV} (\rho_h^\pi(\mu; P)\Vert \rho_h^\pi(\mu; P_M)) $
>
> We choose the summation form for more friendly to readers. Besides, we follow the symbolic system of MBPO[1] partly, which also adopts a summation form for analysis and then experiments in continuous space. We checked our primary results will not be affected. We are pleased to discuss if there is something not well thought out.
>
> [1] Michael Janner, Justin Fu, Marvin Zhang, and Sergey Levine. When to trust your model: Model-based policy optimization. In Advances in Neural Information Processing Systems, 2019.
>
> >  **Q5:** "Can you please illustrate how exactly is the volume of the of convex closure computed from the replay buffer?"
>
> We detailed how to compute convex closure from the replay buffer in Appendix Line 181-187. As for the convex hull, we first perform Principal Component Analysis on the states to reduce the dimension and then leverage the Graham-Scan algorithm to construct a convex hull of N points which are sampled from the replay buffer.
>
> > **Q6:** "Can you please point me to the equation where you derive an estimation for the model shift as a product of volume × avg prediction error?"
>
> Our proposed constraint estimation is a practical design, so that we call it "practical overestimation" in the paper. We have provided an explanation for the overestimation in Appendix D.5, paragraph "Estimation on model shifts". Besides, Appendix Figure 3 demonstrates that our prediction is higher than the true estimation and their trends stand consistent.
>
> As discussed in Line 239-240, the constraint is based on an unobserved model $M_2$  so that we seek to construct a surrogate function and use current data to estimate it. More specifically, recall the constraint function $\mathcal D_{TV}(P_{M_1}(\cdot\vert s,a)\Vert P_{M_2}(\cdot\vert s,a)) \leq \sum_{s'\in {\cal S}}\frac{1}{2}[\vert P_{M_1}(s'\vert s,a)- P(s'\vert s,a)\vert +\vert P_{M_2}(s'\vert s,a)-P(s'\vert s,a)\vert]$.
>
> As the updated dynamics $P_{M_2}$ usually comes closer to the true dynamics $P$ than the previous one $P_{M_1}$, we can use $\sum_{s'\in {\cal S}}\vert P_{M_1}(s'\vert s,a)- P(s'\vert s,a)\vert$ to estimate it. And we use the volume to estimate the summation space of $s'$.

---

> ### Author Response · Authors · 2022-08-02
> **Response to Reviewer 99tJ (Q2-Q3)**
>
>
> >  **Q2:** “it is not very clear how minimization of the constrained objective function in Proposition 4.7 turns out to be simple negative likelihood minimization. ”
>
> It is not trivial to turn the objective function in Proposition 4.7 directly into a loss function. Although not directly derived, negative likelihood minimization (NLL) has an optimization objective consistent with it. The objective function encourages us to alleviate model error as much as possible. It motivates us to use NLL for implementation in our practice. NLL method is typically adopted[1] [2] and is effective in learning the transition dynamics of probabilistic models.
>
> We detailed the formulation of our model learning in appendix Line 172-178. To be specific, each dynamical model $f_{\phi_i}$ in the ensemble is a probabilistic neural network that outputs a Gaussian distribution with diagonal covariance , $ f_{\phi_i}(\cdot\vert s_t, a_t)  = {\cal N}(\mu_{\phi_i}(s_t, a_t), \Sigma_{\phi_i}(s_t, a_t))$. These models are trained independently via maximum likelihood. Thus the corresponding loss function is:
>
> ${\cal L}^H(\phi_i) = \sum\limits_{t}^{H}[\mu_{\phi_i}(s_t,a_t)-s_{t+1}]^T\Sigma_{\phi_i}^{-1}(s_t,a_t)[\mu_{\phi_i}(s_t,a_t)-s_{t+1}] + \log \det \Sigma_{\phi_i}(s_t,a_t)$
>
> And the prediction for these ensemble models is, $\hat s_{t+1} = \frac{1}{K}\sum_{i = 1}^{K} f_{\phi_i}(s_t, a_t)$.
>
> [1] K. Chua, R. Calandra, R. McAllister, and S. Levine. Deep reinforcement learning in a handful of trials using probabilistic dynamics models. In Advances in Neural Information Processing Systems (NIPS), pages 4754–4765, 2018.
>
> [2] M. Janner, J. Fu, M. Zhang, and S. Levine. When to trust your model: Model-based policy optimization. In Advances in Neural Information Processing Systems, 2019.
>
>
> > **Q3:** "In Proposition 4.7, there is $P(s′|s,a)$ involved in the expression which can't be ignored since the $(s,a) \sim d_{\pi_2}$ and optimization variable involve $\pi_2$ as well. it's not clear how this is handled as we will never know $P(s′|s,a)$, and how this boils down to simple NLL minimization is not explicitly described"
>
> Sorry for our insufficient explanation, we have refined it in the rebuttal revision.
>
> We cannot obtain $\pi_2$ directly because it is a suboptimal policy under $M_2$. At the time of optimization, $M_2$ is still not available, and $P(s'\vert s,a)$ is not a priori knowledge, thus it is not practical to obtain the true $d^{\pi_2}$. We agree on this point.
>
> Due to the impracticality of solving the optimization objective directly, we turn to design some approximation techniques in implementation. Below we describe the rationality of these designs in our implementation:
>
> * As inferred from the optimization objective, the minimization of the objective function can be achieved when we try to minimize the difference between $M_2$ and the real environment. To reduce model bias, we chose to use NLL as a loss function in our implementation, which has been shown an effective way to learn model dynamics.
>
> * Besides, we perform model learning on the current interaction tuples. The distribution mismatch indeed exists due to policy difference, this mismatch is somewhat tolerable (we explained it from the perspective of control theory in the Appendix). Off-policy reinforcement learning algorithms also adopt similar techniques by using existing interaction data for learning and optimization.

---

> > ### Comment · Reviewer_99tJ · 2022-08-07
> > **Discussion regarding Q2-Q3**
> >
> > Thanks for the comments and explanations. I agree with the justification given on Q2.
> >
> > However, the justification on Q3 is still unclear to me as to how the approximate method can tackle the distribution mismatch for this particular setting ? Can you provide a more detailed justification on why will the distribution mismatch be tolerable for this particular setting of MBRL. Thanks.

---

> > > ### Author Response · Authors · 2022-08-08
> > > **Thanks! Further discussion regarding Q3.**
> > >
> > > Thank you for your additional feedback! We respond to the concerns below:
> > >
> > > About Q3, since we can not obtain $d^{\pi_{2}}$ directly, we turn to approximate it using previously sampled data. And it does exist distribution mismatch.
> > >
> > > About why we say the distribution mismatch is tolerable, we verified it by experiments in our paper. In Figure 3 of Appendix E.5 (quickly see in https://anonymous.4open.science/r/Picture-246D/README.md), we compare the estimation of model shifts after updating (when we can obtain samples from $d^{\pi_{2}}$) and pre-estimation of model shifts before model updating (what we use for approximation). From this practical result, we can see their trends stand consistent, and the bias in the approximation is tolerable as it can be bridged by adjusting the constraint threshold $\alpha$.
> > >
> > > Before, we provided a proof that similar model derives similar distribution from the perspective of control theory under several assumptions in the Appendix.
> > >
> > > Below we try to explain from the general MBRL process perspective, but it is not the focus of our paper. With a limited model shift and rollout policy shift, the difference between the rollout data distributionsfrom two models is limited. Because the policies are derived from the rollout data and then collect interation data, the interaction data difference might also be limited. Specifically, let model shift $\epsilon_{M_1, M_2}^{\pi} = E_{s,a\sim P_{M_1}^\pi,\pi}[D_{TV}(P_{M_1}(\cdot\vert s,a)\Vert P_{M_2}(\cdot\vert s,a)]$,  rollout policy in $M_1$ be $\pi$, rollout policy in $M_2$ be $\pi'$, rollout policy shift be $\delta_{\pi, \pi'}^{M_1} = E_{s\sim d_{M_1}^{\pi}}[D_{TV}(\pi'(a\vert s)\Vert \pi(a\vert s))]$.
> > >
> > >
> > > Then, we have that $\Vert d_{M_2}^{\pi'} - d_{M_1}^{\pi} \Vert_1 \leq  \frac{2}{1-\gamma} (\delta_{\pi,\pi'} + \epsilon_{M_1, M_2}^\pi)$ , which is limited as well. We assume that the policies derived from these two distribution are similar as well. Then we have that $d^{\pi_2}$ similar to $d^{\pi_1}$ as well, i.e., $\Vert d^{\pi_2} - d^{\pi_1}\Vert_1 \leq \frac{2}{1-\gamma}\delta_{\pi_1,\pi_2}$
> > >
> > > Thank you again for your reply. Hope we have resolved your concerns and we are glad to have further discussions if you have any questions.

---

> ### Author Response · Authors · 2022-08-02
> **Response to Reviewer 99tJ (Q1)**
>
> Thank you for your valuable comments and suggestions, which are of great help to improve the quality of our work. We sincerely appreciate your positive comments on our proposed theory as a novel and interesting analysis to address an important question. We carefully answer each of your concerns as below.
>
> > **Q1:**  "The primary weakness lies in various assumptions that have been made to derive some of the primary results that are not very general. To begin with, the main result in Corollary 4.8 is shown with a generative model assumption which is quite restrictive. The primary novelty of the research from a solution perspective lies in designing the event-triggered mechanism and the theory is shown with a generative model assumption also with linear quadratic regulator (which I still believe can be relaxed) and not for a general scenario."
>
> There might be a misunderstanding.  We design the event-triggered mechanism based on Proposition 4.7 which doesn't depend on the generative model assumption and linear quadratic regulator.
>
> Corollary 4.8 is not a primary result but just an example. It is used to support the motivation of designing dynamically varying model training intervals but not to guide the algorithm design directly. To give Corollary 4.8, we followed assumptions from these papers [1] [2],  and these assumptions would not hurt the generalizability of the event-triggered mechanism. It is indeed exciting to give some other feasible solutions for Proposition 4.7 under weaker assumptions, but it is not the focus of this work yet.
>
> [1] Alekh Agarwal, Sham Kakade, and Lin F Yang. Model-based reinforcement learning with a generative model is minimax optimal. In Conference on Learning Theory, 2020.
>
> [2] Gen Li, Yuting Wei, Yuejie Chi, Yuantao Gu, and Yuxin Chen. Breaking the sample size barrier in model-based reinforcement learning with a generative model. In Advances in Neural Information Processing Systems, 2020.

---

> > ### Comment · Reviewer_99tJ · 2022-08-07
> > **Discussion regarding Q1**
> >
> > Thanks for the comments. I understand that 4.8 is an example and not the primary result. I could not find a general proof without the generative model assumption, can you please point me to the same.

---

> > > ### Author Response · Authors · 2022-08-08
> > > **Thanks! Further discussion regarding Q1.**
> > >
> > > Thank you for your reply. We really enjoy communicating with you and appreciate your efforts.
> > >
> > > It is hard to give a direct solution to solve the constrained optimization problem in Proposition 4.7 under a general setting.  A general proof for a feasible solution of Proposition 4.7 is quite exciting but is not the focus of the work yet.
> > >
> > > The event-triggered mechanism is a practical design to follow the inspiration from Proposition 4.7. Through the event-triggered mechanism, we can decouple the constraint and objective of this intractable constraint optimization problem and solve them asynchronously, by detecting the model shifts constraint in the policy exploration stage and optimizing the objective function in the model training stage. Besides, the special feasible solution example in Corollary 4.8 implies that the dynamically varying model training interval may help the monotonicity. This inspiration is consistent with the event-triggered mechanism.
> > >
> > > Hope we have resolved your concerns and we are glad to have further discussions if you have any questions.

---

> > > > ### Comment · Reviewer_99tJ · 2022-08-09
> > > > **Discussion regarding Q1.**
> > > >
> > > > Thanks for the comments. Yes, I agree with the authors on the decoupling aspect and I think that is correct. I acknowledge the fact that a general proof for 4.7 is hard but wanted to emphasize that is critical as the analysis is somewhat strongly relying on the proof of 4.7.  Hence, that will be important even for this work.
> > > > However, under some assumptions, the authors have shown the proof which I believe is sufficient given this work points to an interesting question.

---

> > ### Author Response · Authors · 2022-08-09
> > **Thank you for your responsible reply!**
> >
> > Dear reviewer,
> >
> > We thank the reviewer for your insightful and constructive comments and suggestions, which provide much helpful guidance to improve the quality of our paper!  We really enjoy communicating with you and appreciate your efforts!
> >
> > Best wishes!
> >
> > The authors.

---

### Official Review · Reviewer_iXge · 2022-07-10

**Rating:** 5
**Confidence:** 4
**Soundness:** 3 good
**Presentation:** 3 good
**Contribution:** 3 good

**Summary:**

This paper studies how to ensure optimization monotonicity of learning an accurate dynamics model for MBRL. They derive a lower bound for the derived policy performance improvement that depends on the one-step dynamics prediction error of the current model, constraint by the model shift.

Inspired by the theory, they propose an algorithm to dynamically alternate between policy exploration and model learning, with the aim to improve optimization monotonicity. They evaluate the proposed algorithm on a series of MoJoCo control tasks and compared the results against a few model-free baselines, and show that their model-based method is able to reach the same SOTA asymptotic performance while being more stable and sample efficient.

**Questions:**

1. In Figure 1, the performances on different tasks are capped at different timesteps. In several cases the learning curves have not stabilized yet, e.g. Walker2d and Swimmer. Could you please report the final performance where the learning curves stabilize or at 5M steps (as used to determine the “asymptotic performance” of the SAC and MBPO baselines).
2. What is the y-axis in Figure 3(b)? Could you please include the policy coverage and prediction error results for some of the baselines for a comparison?
3. Figure 4 looks a bit confusing to me. Do the MBRL baselines start with an initial 30k steps exploration followed by model learning (inferred from Figure 4)? Why only showing 4k steps per stage instead of the whole 60k? What is the y-axis?
4. Figure 4(b) shows that the model shifts estimation jumps drastically from stage 1 to stage 2, and that it doesn’t hit the threshold at all in stage 1 (first 4k steps). Could the authors please explain this unexpected observation?
5. The absolute performance of TRPO on HalfCheetah and Ant tasks shown in Figure 5(a) are much lower than the other baseline results shown in Figure 1. The variance also seems quite high. Is it expected?
6. Why choosing these specific training step caps in Figure 5(b)?
7. Could you briefly discuss the costs induced by estimating the volume of the convex closure, especially when applying to high-dimensional state space?

**Limitations:**

The paper mentioned briefly in the conclusion that finding the optimal threshold for each specific environment may be time consuming. Could you please elaborate on that? For example, how different are the thresholds for different environments and the amount of steps it takes for tuning.

**Strengths And Weaknesses:**

Strength:

The problem studied in the paper is important and relevant to practice. The paper is well-written and easy to follow. The proposed method is theoretically motivated.

Weakness:

My main concerns are around the experiments and potential limitations. Please see the section below for detailed comments.

---

> ### Author Response · Authors · 2022-08-02
> **Response to Reviewer iXge (Q4-Q8)**
>
> >  **Q4:**  "Figure 4(b) shows that the model shifts estimation jumps drastically from stage 1 to stage 2, and that it doesn’t hit the threshold at all in stage 1 (first 4k steps). Could the authors please explain this unexpected observation?"
>
> Sorry for the insufficient explanation of the results in Figure 4(b). The observation is indeed expected due to the Ant environment characteristics.  In the initial stage, exploration in the Ant environment is quite restricted and localized, then the sampled tuples are highly repetitive, which in turn results in the low value of model error as the model has fitted well. While in the second stage, the agent undergoes an epiphany and more fresh data are collected, resulting in a jump in the average return, along with the model error increasing due to these novel data. This observation is consistent with the average return, we find the learning curves (both CMLO and baselines) smooth and rise limitedly in the initial stage (Fig. 1 Ant).
>
> At the same time, this observation also supports our event-triggered mechanism, if we still perform frequent model updates when the sampling repetition is quite high, it is wasteful and may risk overfitting. (Line 250-253).
>
> > **Q5:** "The absolute performance of TRPO on HalfCheetah and Ant tasks shown in Figure 5(a) are much lower than the other baseline results shown in Figure 1. The variance also seems quite high. Is it expected?"
>
> Yes, it is expected, because the absolute performance of Dyna-style model-based algorithms highly depends on its model-free part. For example, in Halfcheetah, purely model-free TRPO achieves the asytomptic performance 4000 around 8M steps. So when using TRPO as the policy optimization oracle, it is expected to get such performance and variance. Our baseline paper SLBO[1] also reported a similar observation when adopting TRPO in its Appendix figure 4.
>
> [1] Luo Y, Xu H, Li Y, et al. Algorithmic Framework for Model-based Deep Reinforcement Learning with Theoretical Guarantees[C]//International Conference on Learning Representations. 2018.
>
> > **Q6:** "Why choosing these specific training step caps in Figure 5(b)?"
>
> This is not a specific pick. The performance in DKitty-Stand starts to converge at 1700 steps, while in Panda-Reaching starts to converge at 1000 steps, then we freely chose 2000 and 1500 steps as the training steps cap.
>
> > **Q7:** "Could you briefly discuss the costs induced by estimating the volume of the convex closure, especially when applying to high-dimensional state space?"
>
> Actual computation time: compared to the unconstrained cases, our total training time increased by an average of 3.24h in the HalfCheetah environment (300k steps, 17-dimensional state space) and 3.96h in the Ant environment (300k steps, 27-dimensional state space). And our computing infrastructure and CMLO computational time was listed in Appendix Table 5.  Thus, the additional costs are acceptable.
>
> When applying to the high-dimensional case, we detailed how to calculate state-space coverage in Appendix Line 181-187.  We first sample $N$ tuples from the replay buffer and then perform Principal Component Analysis to reduce the dimension, then we leverage the Graham-Scan algorithm to construct a convex hull of these $N$ points, which only takes $O(N \log N)$ for time complexity.
>
> > **Q8:** "The paper mentioned briefly in the conclusion that finding the optimal threshold for each specific environment may be time consuming. Could you please elaborate on that? For example, how different are the thresholds for different environments and the amount of steps it takes for tuning."
>
> We admit that $\alpha$ is a hyperparameter that needs manually tuning. Compared with MBPO, we only add the tuning cost of $\alpha$, yet we do not need to tune the fixed model training frequency in MBPO.
>
> In Table 3 of the appendix, we presented the thresholds used in 6 environments. $\alpha=2.0$ for Swimmer, HalfCheetah, and Ant, $\alpha=2.5$ for Humanoid, $\alpha=3.0$ for Walker2d, and $\alpha=1.2$ for Hopper.
>
> To further demonstrate the tuning process of this parameter, we present the tuning process for Humanoid and Ant environments as follows.
>
> Humanoid         |     |    |     |  |
> ----------|--------|-----------|-----------|---------
> alpha          | 1.0     | 2.0     | 2.5     | 3.0
> Average Return | 5480.53 | 6402.78 | 6775.67 | 6348.92
>
>
> Ant         |     |    |     |  |
> ------------|---------------|------------|------------|---------
> alpha          | 1.0     | 2.0     | 2.5     | 3.0
> Average Return | 5600.23 | 6810.42 | 6382.25 | 6523.86
>
>
> Thanks again for your comments. We will be continued to polish our statement for clarity in our revision. If you have any other questions, please post them and we are happy to have further discussions.

---

> > ### Comment · Reviewer_iXge · 2022-08-08
> > **Thank you for your response**
> >
> > Thanks the authors for their thorough response as well as their efforts to revise the paper to make it more clear. I have increased my score by 1.

---

> > > ### Author Response · Authors · 2022-08-08
> > > **Thank you for your inspiring reply!**
> > >
> > >
> > > Dear reviewer,
> > >
> > > Thank you for helping us improve the paper and for updating the score! We really appreciate your comments and suggestions!
> > >
> > > Best wishes!
> > >
> > > The authors.

---

> ### Author Response · Authors · 2022-08-02
> **Response to  Reviewer iXge (Q1-Q3)**
>
> Thank you for your comments and suggestions, the detailed responses regarding each problem are listed below.  We hope to resolve the misunderstandings caused by the imperfection of our presentation. If you have any other questions, please post them and we are happy to have further discussions.
>
> >  **Q1**:  "Figure 1, the performances on different tasks are capped at different timesteps. In several cases the learning curves have not stabilized yet, e.g. Walker2d and Swimmer. Could you please report the final performance where the learning curves stabilize or at 5M steps (as used to determine the “asymptotic performance” of the SAC and MBPO baselines)."
>
> Thank you for the comment. Actually, we did report  the maximum average return (a kind of asymptotic performance) in Appendix Table 2. Results show that our method has comparable asymptotic performance in Walker2d (350k) and Swimmer (350k) environments. We will refine our description on  this table in next revision for clarity.
>
> We observed that MBRL baselines (MBPO, AutoMBPO) show convergence at 300k, thus we choose the 300k as the capped steps for the sake of fairness. Besides, CMLO also starts to converge around 300k. We have extended  the plot by capping the curve at 350k in the rebuttal revision.
>
> >**Q2:**   "What is the y-axis in Figure 3(b)? Could you please include the policy coverage and prediction error results for some of the baselines for a comparison?"
>
> * The y-axis in Figure 3(b) is the prediction error as shown in the caption.
>
> * Prediction error: We performed the prediction error comparison to the baseline MBPO in Appendix Figure 4.
>
> * Policy coverage: Policy coverage represents the exploration ability of the policy. The policy coverage increasing with the stages means that the policy has new explorations at every stage and may not fall into a local optimum. Per the reviewer's suggestion, we further report the numerical comparison to MBPO here.
>
>
>   | Env         | Algo | Stage1     | Stage2     | Stage3     | Stage4     | Stage5     |
>   |-------------------|------------------|-------------------|------------------|-------------------|------------------|-------------------|
>   | HalfCheetah | CMLO | 138.566126 | 182.281857 | 243.466268 | 302.816499 | 344.356213 |
>   |   HalfCheetah          | MBPO | 129.251405 | 173.085743 | 242.492030 | 264.853917 | 338.555574 |
>   | Ant         | CMLO | 354.154379 | 744.91538  | 849.473640 | 876.119479 | 909.798043 |
>   |  Ant    | MBPO | 342.134362 | 729.295456 | 821.658472 | 864.933838 | 880.252964 |
>
>   Here, each stage $i$ contains $(60\times(i-1), 60\times i ]k$ steps. In HalfCheetah, we find that our policy achieves higher coverage especially in first 4 stages than MBPO. Consistently, we find that our policy enjoys higher performance, with an average return lead of about 1855.29 over MBPO in the first 300k steps. Likewise, the growth of policy coverage in Ant is also consistent with the rise in average return. The increase in policy coverage helps the policy to refrain from falling into a local optimum, thus improving performance.
>
> Again, thanks for your suggestions, we have incorporated these comparison results into the rebuttal revision.
>
> >  **Q3:** "Figure 4 looks a bit confusing to me. Do the MBRL baselines start with an initial 30k steps exploration followed by model learning (inferred from Figure 4)? Why only showing 4k steps per stage instead of the whole 60k? What is the y-axis?"
>
> * No, both CMLO and other MBRL baselines start with the initial steps within 5k steps.
> * For visual clarity, we arbitrarily chose 4k within each stage as an illustration, from which we can see how the trigger frequency varies with the stages. A clear display of 60k data requires a lot of space so we turn to only display 4k per stage.  We have added the full 60k figure in the appendix of the rebuttal revision for clarity.
> * The y-axis is our estimation of the triggered condition (the detailed formula seen in Appendix 202-).

---

> ### Author Response · Authors · 2022-08-06
> **We sincerely look forward to your reply.**
>
> Dear reviewer,
>
> We appreciate your comments and suggestions. We hope our last reply has resolved all your concerns. We have refined our explanation and added experiments as you suggested. If you have any other questions, we are also pleased to respond. We sincerely look forward to your response.
>
> Best wishes!
>
> The authors.

---

### Official Review · Reviewer_r1Xm · 2022-07-11

**Rating:** 6
**Confidence:** 5
**Soundness:** 2 fair
**Presentation:** 3 good
**Contribution:** 3 good

**Summary:**

This paper first demonstrates that the model shifts---the difference between the updated model and the model before updating---hinder the monotonic improvement of model-based RL. To tackle this problem, the paper proposes CMLO, which introduces an event-triggered mechanism to determine when to alleviate the model shifts. Experiments show the effectiveness of the proposed method.

**Questions:**

My suggestions are as follows.

1. The advantages of the proposed event-triggered mechanism are unclear. The authors may want to explain why the event-triggered mechanism is more effective than previous methods [1, 2].

2. The authors claim that the proposed constraint estimation is “the practical overestimation for the model shifts” in Lines 247-248. The authors may want to provide the theoretical analysis of the overestimation.

3. Several definitions are missing, such as $\sigma_{M_1, M_2}$ in Theorem 4.6 and $\Delta \mathcal{D}$ in Line 245.

4. The formulation for computing the proposed volume $vol(\mathcal{S}_\mathcal{D})$ in Lines 239-244 is missing.

5. The authors may want to provide the detailed motivation of Equation 5.1.

6. The experiment settings in Section 6.2 are missing, such as the settings of all methods in Figure 2 and that of the visualization in Figure 3. The authors may want to provide the detailed settings of all ablation studies.

[1] Michael Janner, Justin Fu, Marvin Zhang, and Sergey Levine. When to trust your model: Model-based policy optimization. In Advances in Neural Information Processing Systems, 2019.

[2] Hang Lai, Jian Shen, Weinan Zhang, Yimin Huang, Xing Zhang, Ruiming Tang, Yong Yu, and Zhenguo Li. On effective scheduling of model-based reinforcement learning. In Advances in Neural Information Processing Systems, 2021.


**Limitations:**

Yes, the authors adequately addressed the limitations and potential negative societal impact.

**Strengths And Weaknesses:**

Strengths:

1. The authors propose the theoretical analysis to show that the model shifts hinder the monotonic improvement of model-based RL, which provides a useful perspective in model-based RL.

2. Experiments demonstrate that the proposed method improves the performance and generalization of existing methods.


Weaknesses:
1. The authors propose a constraint estimation in their method to estimate the model shifts in Proposition 4.7. However, in Lines 245-247, the authors use the data from the current ensemble models and real environment to compute the constraint estimation, which is not consistent with Proposition 4.7.

2. The authors propose the event-triggered mechanism to determine when to update the model instead of the frequent model updating in MBPO [1]. However, the motivation to alleviate the model shifts by the event-triggered mechanism is unclear. The authors may want to explain why they introduce the event-triggered mechanism based on their proposed theoretical analysis.

[1] Michael Janner, Justin Fu, Marvin Zhang, and Sergey Levine. When to trust your model: Model-based policy optimization. In Advances in Neural Information Processing Systems, 2019.

---

> ### Author Response · Authors · 2022-08-02
> **Response to Reviewer r1Xm (Q8)**
>
>
>
> > **Q8:** The experiment settings in Section 6.2 are missing, such as the settings of all methods in Figure 2 and that of the visualization in Figure 3. The authors may want to provide the detailed settings of all ablation studies.
>
> Thanks for the suggestions, we have included the detailed settings of our ablation studies in the Appendix in the rebuttal revision. Note that other hyperparameters we do not mention below are the same as the hyperparameter settings in Appendix Table 3.
>
> * Figure 2 (a):  We compare to three unconstrained cases (given fixed model training interval), the fixed intervals are shown in the caption (w/o-n), and the number n means how many newly real interaction tuples have been collected. These experiments are averaged over 5 random seeds. (Sorry for the typo in Ant figure legend w/o-100, it should be w/o-150. We will fix it later on.)
> * Figure 2 (b):  It is used to present the triggers times during training of the experiments in (a) (Same environment in the same column figures). We compute the average triggered times over 5 random seeds per 10k steps. For clarity, only the mean values are shown here. And the y-axis represents the number of the model training times performed every 10k steps. Note that Figure 2(a) and Figure 2(b) share the legends.
> * Figure 3 (a):  The stage $i$  represents $[60\times(i-1), 60\times i]k$ steps. For visualization, we firstly sampled 6k tuples from the replay buffer in each stage, then we performed Uniform Manifold Approximation and Projection (UMAP) to get a visualization of policy coverage (state-space coverage). The data used here are from w/-ours experiments in Figure 2.
> * Figure 3 (b):  This figure shows the prediction error, ${\cal L}(\Delta{\cal D}) = \mathop{\mathbb{E}}\limits_{(s,a,s')\in \Delta{\cal D}} \big[\frac{1}{K}\sum\limits_{i=1}^K \Vert s' - \hat{f_{\phi_i}}(s,a) \Vert\big]$ (details in Appendix 188-190).  The data used here are from w/-ours experiments in Figure 2.  $K$ is the size of $\Delta{\cal D}$,  or to say the time steps interval between model updates.
>
> For other ablation studies,  we also provide detailed settings as follows:
>
> * Figure 4:   The y-axis is the estimation on our practical triggered condition (the detailed explanation and settings in Appendix 198-202),  $\sum_{i=0}^{[\tau/F]} \log \Big(\frac{vol(\mathcal S_{\mathcal D_t \cup \Delta \mathcal D(Fi)})}{vol(\mathcal S_{{\mathcal D}_t})} \cdot \mathcal L(\Delta \mathcal D(Fi)) + \beta \Big) \geq \alpha$.  The data used here are from w/-ours experiments in Figure 2. Here, $\beta=1.0$, $F=20$ for Hopper, and $F=50$ for the other five benchmarks, as listed in Appendix Table 3.
>
> * Figure 5 (a): For model network settings, we adopt the same as present in Appendix Table 3. About Legend: w/o-n, we use a data sampler with batchsize=20, thus we get 20*n real interactions during the model training interval. We compute the total triggered times and scale them to [0,1], which is shown in the bar plots.
>
>   * For the TRPO part, the key parameters are listed below:
>
>     Ant: horizon = 1000, $\gamma$=0.99, gae=0.97, step_size=0.01, iterations=40
>
>     HalfCheetah:  horizon = 1000, $\gamma$=0.99, gae=0.95, step_size=0.01, iterations=40
>
> * Figure 5 (b):
>
>   * For the dynamical models network: Gaussian MLP with 3 hidden layers of size 200, batch size is 64, and the learning rate is 0.0001.  For the iLQR part: LQR_ITER=10, R=0.001, Q=1, horizon=5. For legend: w/o-n, we get n real interactions during the model training interval. And $\alpha=0.5$ in w/-ours. We compute the total triggered times, and scale them to [0,1], which is shown in the bar plots.
>   * DKitty-Stand: This environment is from the DKittyStandFix in ROBEL, the environment parameters are the same as the forward setting in the original setting, we modified the task horizon as $T = 50$.
>   * Panda-Reaching: state space dimension 20, action space dimension 7, task description: Under the simulation conditions of Coppeliasim, the endpoint of the panda arm is required to reach a random target point in space from a fixed initial position. The reward is set as the negative of the $L_2$-norm distance between the current position of the end point and the position of the target point.  Range of target points is $[1.05, -0.25, 1.1] \times [1.2, 0.25, 1.4]$.  And task horizon $T = 50$.
>
> Finally, we hope we resolve all of your concerns. We have refined our explanations in the rebuttal revision according to your suggestions. And we will be continued to polish our language and the clarity in our revision. Thanks again for your suggestions.

---

> ### Author Response · Authors · 2022-08-02
> **Response to Reviewer r1Xm (Q4 - Q7)**
>
>
>
> > **Q4:** “The authors claim that the proposed constraint estimation is “the practical overestimation for the model shifts” in Lines 247-248. The authors may want to provide the theoretical analysis of the overestimation.”
>
> Our proposed constraint estimation is a practical design, so that we call it "practical overestimation" in the paper. We provided an explanation for the overestimation in Appendix D.5, paragraph "Estimation on model shifts".  Besides, we peformed an experiment to show the overestimation in figure 3 of the appendix, which demonstrates that our prediction (unobserved $M_2$) is higher than the estimation (ground truth $M_2$) , their trends stand consistent.
>
>
>
> > **Q5:** "Several definitions are missing, such as $\sigma_{M_1, M_2}$ in Theorem 4.6 and $\Delta {\cal D}$ in Line 245."
>
> Thanks for your kindly reminder. We have refined the missing definitions in the rebuttal revision.
>
> * $\sigma_{M_1, M_2}$denotes the model shift constraint threshold between model $M_1$ and $M_2$.
>
> * We denote replay buffer as ${\cal D}$, and  $\Delta {\cal D}$ is the newly encountered data that expands the replay buffer.
>
> > **Q6:** "The formulation for computing the proposed volume in Lines 239-244 is missing."
>
> We presented the details for computing the proposed volume in Appendix D.3 (Line 181-187). It is a practical design. We first sample $N$ Tuples from the replay buffer, then perform Principal Component Analysis for dimension reduction and then leverage the Graham-Scan algorithm to construct a convex hull of these $N$ points.
>
> > **Q7:** "The authors may want to provide the detailed motivation of Equation 5.1."
>
> Eq 5.1 is the event-triggered condition motivated by Proposition 4.7.  Proposition 4.7 implies that, when the model shift constraint boundary is violated according to the estimation, we need to pause data collecting and turn to solve the optimization objective.   Besides, although turning to train models once not to violate the constraint is theoretically reasonable, we avoid doing so in practice because performing an update on data with a minor shift in coverage and distribution is wasteful and may risk overfitting (Line 250-253).
>
> We further explain each item of Eq. 5.1 here. We adopt the fraction form $\frac{vol(\mathcal S_{ \mathcal D_t \cup \Delta \mathcal D(\tau )})}{vol(\mathcal S_{\mathcal D_t})} \cdot \mathcal L(\Delta \mathcal D(\tau)) $ for the triggered condition. Denominator $vol(\mathcal S_{{\mathcal D_t}\cup \Delta {\mathcal D}(\tau)})\cdot {\cal L}(\Delta {\mathcal D}(\tau))$  is used to obtain an estimation for the model shift, as detailed in Line 239-248. The numerator $vol (\mathcal S_{D_t})$, on the one hand, is to reduce numerical errors; on the other hand, this fraction reflects the relative change of the policy coverage and model shift if we turn to train $M_2$ under different $\tau$  starting from $M_1$. This fraction reflects the current ability to digest new data. It can facilitate the setting of threshold, for we do not need to tune $\alpha$ once the policy coverage updates.
>
> Sorry for the unsufficient explanation on it. And we have refined the description in the rebuttal revision according to your suggestion.

---

> ### Author Response · Authors · 2022-08-02
> **Response to Reviewer r1Xm (Q3)**
>
> >  **Q3:** “The advantages of the proposed event-triggered mechanism are unclear. The authors may want to explain why the event-triggered mechanism is more effective than previous methods [1, 2].”
>
> Thanks for your suggestions. Let us first summarize the advantages of the event-triggered mechanism:
>
> * It is theoretically motivating which helps monotonicity, as detailed in A2 (response to Q2).  Event-triggered mechanism focuses on the impact of model shifts. And drastic model shifts will hurt the analysis in previous work.
>
> * It is a simple, effective, and training-free method to solve the constrained problem in Proposition 4.7.
>   * Compared to MBPO [1], it only introduces one hyperparameter, and it does not introduce additional training.
>
>   * Compared to AutoMBPO [2], it does not need to construct a bilevel deep RL problem. So that it does not need to cost too much time in training.
>
> We have reflected the above explanations to address the reviewer's concern in the rebuttal revision.
>
> Below we provide some detailed analysis and comparison:
>
> * MBPO performs a fixed model training frequency.  Our event-triggered mechanism instead provides a dynamically varying training frequency.  The advantage of dynamically adjusting the model training interval has been demonstrated in Corollary 4.8.  Besides, our event-triggered mechanism accounts for the impact of model shifts on performance, which is neglected in MBPO. We can infer from MBPO that  $\epsilon_m$ will be large when model shifts are too large, thus the monotonicity will be destroyed because it could hardly get an updated policy which increases the return under a certain model above $2R[\frac{\gamma^{k+1} \epsilon_{\pi}}{(1-\gamma)^2} + \frac{\gamma^k \epsilon_\pi}{(1-\gamma) }+ \frac{k}{1-\gamma}(\epsilon_m')]$.
> * AutoMBPO is not meant to propose a practical MBRL algorithm but to do hyperparameter tuning. It did not touch on the model shifts issue. Although it can learn the model training frequency along with many other coupled hyperparameters, it is quite time-consuming and introduces more hyperparameters to tune in the outer RL. Our algorithm does not have such a heavy computational cost, for example, we spent an average 33.31h for Humanoid while AutoMBPO needs 245.33h.
>
> [1] Michael Janner, Justin Fu, Marvin Zhang, and Sergey Levine. When to trust your model: Model-based policy optimization. In Advances in Neural Information Processing Systems, 2019.
>
> [2] Hang Lai, Jian Shen, Weinan Zhang, Yimin Huang, Xing Zhang, Ruiming Tang, Yong Yu, and Zhenguo Li. On effective scheduling of model-based reinforcement learning. In Advances in Neural Information Processing Systems, 2021.

---

> ### Author Response · Authors · 2022-08-02
> **Response to Reviewer r1Xm (Q1, Q2)**
>
> Thanks for your comments and suggestions. We provide clarification to your questions and concerns as below. If you have any additional questions or comments, please post them and we would be happy to have further discussions.
>
> > **Q1:**  "The authors propose a constraint estimation in their method to estimate the model shifts in Proposition 4.7. However, in Lines 245-247, the authors use the data from the current ensemble models and real environment to compute the constraint estimation, which is not consistent with Proposition 4.7."
>
> There is probably some misunderstanding here. Our implementation is consistent with Proposition 4.7. We did give a detailed explanation on the consistency in Appendix D.5 (Line 243-255) and performed an experiment to visualize the connection between them in Figure 3 of the appendix.
>
> It is consistent with Proposition 4.7 considering the general MBRL procedure because $M_2$ is trained based on the data from the current ensemble models and real environment (Line 235-238).  More specifically, recall the constraint function
> $\mathcal D_{TV}(P_{M_1}(\cdot\vert s,a)\Vert P_{M_2}(\cdot\vert s,a)) \leq \sum_{s'\in {\cal S}}\frac{1}{2}[\vert P_{M_1}(s'\vert s,a)- P(s'\vert s,a)\vert +\vert P_{M_2}(s'\vert s,a)-P(s'\vert s,a)\vert]$.  As the updated dynamic $P_{M_2}$ usually comes closer to the true dynamics $P$ than the previous one $P_{M_1}$, we can use $\sum_{s'\in {\cal S}}\vert P_{M_1}(s'\vert s,a)- P(s'\vert s,a)\vert$ to estimate it.
>
> Moreover, the results in Appendix Figure 3 demonstrate that our constraint estimation is higher than the true value and their trends stand consistent.
>
>
> >  **Q2:** “The authors propose the event-triggered mechanism to determine when to update the model instead of the frequent model updating in MBPO[1]. However, the motivation to alleviate the model shifts by the event-triggered mechanism is unclear. The authors may want to explain why they introduce the event-triggered mechanism based on their proposed theoretical analysis. ”
>
> There could be a misunderstanding. The event-triggered mechanism does not aim to alleviate model shifts but to detect model shifts and choose suitable occasions to train the model, as described in Line 249-253.
>
> About the motivation of the event-triggered mechanism:
>
> * The event-triggered mechanism is proposed to handle the constraint optimization problem of Proposition 4.7.  Through it, we can decouple the constraint and objective of this intractable constraint optimization problem and solve them asynchronously, by detecting the model shifts constraint in the policy exploration stage and optimizing the objective function in the model training stage.
> * Besides, considering the general MBRL procedure, $M_2$ is updated based on newly collected data, so it is hard to satisfy (R2) condition when the distribution of these newly collected data differs drastically from that of training $M_1$. Thus it is natural to call off the policy exploration when the novelty of the newly collected data reaches the threshold. This is an intuitive explanation of our event-triggered condition.

---

> ### Author Response · Authors · 2022-08-06
> **We sincerely look forward to your reply.**
>
>  Dear reviewer,
>
> We first thank you again for your valuable comments and suggestions. In the previous replies, we think we have addressed your concerns point by point and refined details in the rebuttal revision as you suggested. We sincerely look forward to your reply to our response.
>
> Best wishes!
>
> The authors.

---

> > ### Comment · Reviewer_r1Xm · 2022-08-07
> > **Thanks for the authors' rebuttal.**
> >
> > Thanks for the authors' rebuttal. I have read the response and all the other reviewers' comments. However, I still have some concerns of the constraint estimation, which have not been properly addressed.
> >
> > 1. The authors use $\sum\_{s' \in \mathcal{S}} | P\_{M\_1}(s'|s,a) - P(s'|s,a) |$ to estimate the model shifts in the response to Q1, which is different from $vol( \mathcal{S}\_{\mathcal{D}}) \cdot \mathcal{L} ( \Delta \mathcal{D})$ in Line 247. The authors may want to provide the detailed derivation from $\sum\_{s' \in \mathcal{S}} | P\_{M\_1}(s'|s,a) - P(s'|s,a) |$ to $vol( \mathcal{S}\_{\mathcal{D}}) \cdot \mathcal{L} ( \Delta \mathcal{D})$.
> >
> > 2. The authors provide the details for computing the proposed volume $vol(\mathcal{S}\_\mathcal{D})$ in the response to Q6. The authors may want to further provide the detailed formulation of $vol(\mathcal{S}\_\mathcal{D})$.
> >
> > 3. The estimation of the model shifts in Line 247 is $vol( \mathcal{S}\_{\mathcal{D}}) \cdot \mathcal{L} ( \Delta \mathcal{D})$, which is conflict with $vol(\mathcal{S}\_{\mathcal{D} \cup \Delta \mathcal{D} (\tau)}) \cdot \mathcal{L}(\Delta \mathcal{D} (\tau))$ in the response to Q7.

---

> > > ### Author Response · Authors · 2022-08-08
> > > **Thanks! Response to the concerns of the constraint estimation.**
> > >
> > > Thank you for your reply! We respond to the concerns below:
> > >
> > > > **Q1:** The authors use $\sum_{s'\in\mathcal S}|P_{M_1}(s'|s,a)-P(s'|s,a)|$  to estimate the model shifts in the response to Q1, which is different from $vol(\mathcal S_{\mathcal D})\cdot \mathcal L(\Delta \mathcal D)$ in Line 247. The authors may want to provide the detailed derivation from  $\sum_{s'\in\mathcal S}|P_{M_1}(s'|s,a)-P(s'|s,a)|$ to $vol(\mathcal S_{\mathcal D})\cdot \mathcal L(\Delta \mathcal D)$.
> > >
> > > Here, $vol(\mathcal S_{\mathcal D})\cdot \mathcal L(\Delta \mathcal D)$ is a practical design.
> > >
> > > Firstly, $\sum_{s'\in\mathcal S}|P_{M_1}(s'|s,a)-P(s'|s,a)|$ is a intermediate approximation result, from $D_{TV}(P_{M_1}(\cdot|s,a)||P_{M_2}(\cdot|s,a))$ to $vol(\mathcal S_D)\cdot \mathcal L(\Delta \mathcal D)$.
> > >
> > > Then, $\sum_{s'\in\mathcal S}|P_{M_1}(s'|s,a)-P(s'|s,a)|$ is incalculable directly because the whole state space $\mathcal S$ is unknown and inaccessible. During the training process, we can not access all state $s'$ covering $\mathcal S$. And, the current state coverage $\mathcal S_{\mathcal D}$ would be varying, as shown in Figure 3(b). Thus, $vol(\mathcal S_{\mathcal D})$ should be taken into consideration.
> > >
> > > Hence, we use $vol(\mathcal D)$ to estimate the summation space size and adopt the average predictive error $\mathcal L(\Delta \mathcal D)$ for the disagreement on newly encountered data $\Delta \mathcal D$.
> > >
> > > > **Q2:** The authors provide the details for computing the proposed volume $vol(\mathcal S_{\mathcal D})$ in the response to Q6. The authors may want to further provide the detailed formulation of $vol(\mathcal S_{\mathcal D})$.
> > >
> > > $vol(S_D)$ is a notation in our algorithm for the state(policy) coverage, which means the range of state spaces that the current policy $\pi_i$ can explore in the real environment.  In our implementation, we adopt $vol(S_D)$ to estimate the size of the state space $\mathcal S_{pc}^{\pi_i}$ so that we use $\mathcal S_{\mathcal D}$ for denotation. Here, we give a more formal definition about the state(policy) coverage, we use $\mathcal S_{pc}^{\pi_i}$ instead of $\mathcal S_{\mathcal D}$ here for clarity. For a given policy $\pi$, the corresponding state(policy) coverage $\mathcal S_{pc}^{\pi_i}$ can be formulated as  $\mathcal S_{pc}^{\pi_i}: \forall s\in \mathcal S_{pc}^{\pi_i}, a\sim\pi_i(\cdot|s), s'\sim P(\cdot|s,a)\in\mathcal S_{pc}^{\pi_i}$. And, $vol(\mathcal S_{pc}^{\pi_i})$ is the size of the state space $\mathcal S_{pc}^{\pi_i}$.
> > >
> > > > **Q3:** The estimation of the model shifts in Line 247 is $vol(\mathcal S_{\mathcal D})\cdot \mathcal L(\Delta \mathcal D)$, which is conflict with $vol(\mathcal S_{\mathcal D}\cup\Delta \mathcal D(\tau))\cdot \mathcal L(\Delta \mathcal D(\tau))$ in the response to Q7.
> > >
> > > $vol(\mathcal S_{\mathcal D_t}\cup\Delta \mathcal D(\tau))\cdot \mathcal L(\Delta \mathcal D(\tau))$  is a temporal formulation of $vol(\mathcal S_{\mathcal D})\cdot \mathcal L(\Delta \mathcal D)$ so that these two terms are not in conflict.
> > >
> > > In the term $vol(\mathcal S_{\mathcal D})\cdot \mathcal L(\Delta \mathcal D)$, $vol(\mathcal S_{\mathcal D})$ means the policy coverage of the learned model $P_{M_2}$, and $\mathcal L(\Delta \mathcal D)$ represents the disagreement on newly encountered data $\Delta \mathcal D$ in the interval from $P_{M_1}$ to $P_{M_2}$.
> > >
> > > For the term $vol(\mathcal S_{\mathcal D_t}\cup\Delta \mathcal D(\tau))\cdot \mathcal L(\Delta \mathcal D(\tau))$, we get the learned model $P_{M_1}$ at timestep $t$ so the replay buffer is $\mathcal S_{\mathcal D_t}$. When the derived policy interacts with the environment for a period of time $\tau$, the newly encountered data is $\Delta \mathcal D(\tau)$. Thus, we adopt $vol(\mathcal S_{\mathcal D_t}\cup\Delta \mathcal D(\tau))$ and $\mathcal L(\Delta \mathcal D(\tau))$ for the estimation of model shift at  timestep $t+\tau$. Sorry for missing the subscript $t$ in $\mathcal S_{\mathcal D_t}$ in our previous response to Q7.
> > >
> > > Thank you again for helping us improve the paper! We will continue to polish our paper according to your careful suggestions. Hope we have resolved your concerns and we are glad to have further discussions if you have any questions.

---

> > > > ### Comment · Reviewer_r1Xm · 2022-08-08
> > > > **Discussion regarding the volume.**
> > > >
> > > > Thanks for the authors' response. I have a further concern for $vol(\mathcal{S}\_\mathcal{D})$, and one of my concerns remains the same.
> > > >
> > > > 1. The authors use $vol(\mathcal{S}\_\mathcal{D})$ to estimate the size of the state coverage $\mathcal{S}^{\pi\_i}_{pc}$ of the current policy $\pi\_i$. However, the replay buffer $\mathcal{D}$ stores all samples from the initial policy $\pi\_0$ to the current one $\pi\_{i}$, which is not the subset of $\mathcal{S}^{\pi\_i}\_{pc}$. The authors may want to explain why they use $\mathcal{D}$ to estimate the state coverage of the current policy as a practical implementation.
> > > >
> > > > 2. The authors may want to provide the detailed formula for computing $vol(\mathcal{S}\_\mathcal{D})$.

---

> > > > > ### Author Response · Authors · 2022-08-08
> > > > > **Thanks very much for your reply! Further discussion regarding the volume.**
> > > > >
> > > > > Thank you for quickly response!  We really enjoy communicating with you and appreciate your efforts.
> > > > >
> > > > > > **Q1:** The authors use $vol(S_D)$ to estimate the size of the state coverage $S_{pc}^{\pi_i}$ of the current policy $\pi_i$. However, the replay buffer $\mathcal D$ stores all samples from the initial policy $\pi_0$ to the current one $\pi_i$, which is not the subset of $\mathcal S_{pc}^{\pi_i}$. The authors may want to explain why they use D to estimate the state coverage of the current policy as a practical implementation.
> > > > >
> > > > > Sorry for the unclear statement in the previous response. We use $vol(S_{\mathcal D})$ to estimate “the full range of state space can be explored until current", or  $\bigcup_{i} S_{pc}^{\pi_i}$  for clarity.
> > > > >
> > > > > In $\sum_{s'\in\mathcal S}|P_{M_1}(s'|s,a)-P(s'|s,a)|$ , as long as the tuple $(s,a,s')$ is the real interation data, it could be used for computing, so that we do not need to care about these tuples are collected by what policy.  As the $s'$ we have access to are not limited to the current policy collection,  it is reasonable for us to estimate them by replay buffer $\mathcal D$.
> > > > >
> > > > > Thank you for helping us improve the paper!
> > > > >
> > > > >
> > > > >
> > > > > > **Q2:** The authors may want to provide the detailed formula for computing $vol(\mathcal S_{\mathcal D})$.
> > > > >
> > > > > Thank you for your suggestions!
> > > > >
> > > > > About computing the $vol(S_{\mathcal D})$ of the convex closure $S_{\mathcal D} = \{ \sum_{s_i \in \mathcal D} \lambda_i s_i: \lambda_i \geq 0, \sum_i \lambda_i = 1\}$ constructed on the replay buffer D:
> > > > >
> > > > > We first sample $N$ Tuples from the replay buffer $\mathcal D$, then perform Principal Component Analysis for dimension reduction, and then leverage **the Graham-Scan algorithm** to construct a convex hull and output the convex hull volume of these $N$ points.
> > > > >
> > > > > The area of the convex hull of 2-D points $P_1, \ldots, P_n$ is:
> > > > >
> > > > > $\int_{x=-\infty}^{\infty}\int_{y=-\infty}^{\infty} \mathbb 1_{\sum_{i=1}^{n} \lambda_i P_i : \lambda_i \ge 0, \sum_{i=1}^{n} \lambda_i = 1 } (x,y) dx dy$
> > > > >
> > > > > Where $\mathbb{1}_{A}(x,y)$ is the indicator function of 2-D set $A$.
> > > > >
> > > > > Specifically, for computing the area of the 2-D convex hull, we use the Graham scan algorithm described as follows:
> > > > >
> > > > > ---
> > > > >
> > > > > Input: $n$ 2-D points $P_1, \ldots, P_n$
> > > > >
> > > > > Output: the area of the convex hull of $P_1, \ldots, P_n$
> > > > >
> > > > > Steps:
> > > > >
> > > > > 1. Find the left-then-lowest point $P_s$
> > > > > 2. Initialize a point-stack $S$ containing only $P_s$
> > > > > 3. Sort the remaining points in counterclockwise order around $P_s$
> > > > > 4. For each point $P_i$ in the sorted list of remaining points:
> > > > >    Repeat
> > > > >        Denote the top point of $S$ as $P_j$, the second-to-top point as $P_k$
> > > > >        If $P_i$ is **to the left** of the line $\overrightarrow{P_k P_j}$:
> > > > >            break
> > > > >        Else:
> > > > >            Pop $P_j$ off $S$
> > > > >    End Repeat
> > > > >    Push $P_i$ onto $S$
> > > > > 5. Return the area of the polygon defined by the points in $S$
> > > > >
> > > > > Notes:
> > > > >
> > > > > 1. Point $A$ is "to the left" of the line $\overrightarrow{B C}$, if the cross product of $\overrightarrow{B C}$ and $\overrightarrow{B A}$ is positive.
> > > > > 2. The area of the polygon defined by points $P_1, \ldots, P_n$ is $\frac{1}{2} \sum_{i=1}^{n} (P_{i} \times P_{(i+1)\mod n})$
> > > > >
> > > > > ---
> > > > >
> > > > >
> > > > > Thank you again for your comments! We will continue to improve the paper according to your careful suggestions. Hope we have resolved your concerns and we are glad to have further discussions if you have any questions.

---

> > ### Comment · Reviewer_r1Xm · 2022-08-09
> > **Thanks for the authors' rebuttal.**
> >
> > Thanks for the authors' rebuttal and their efforts to improve this work. As the response has addressed my concerns, I would like to increase my score by 1 (6, weak accept). Best wishes.

---

> > > ### Author Response · Authors · 2022-08-09
> > > **Thank you for your inspiring reply!**
> > >
> > > Dear reviewer,
> > >
> > > Thank you for helping us improve the paper and for updating the score! We really enjoy communicating with you and appreciate your valuable suggestions!
> > >
> > > Best wishes!
> > >
> > > The authors.

---

### Meta-Review · Area_Chair_i2eh · 2022-08-27

**Recommendation:** Accept
**Confidence:** Certain

**Metareview:**

This paper studies the relation between model shift and the performance of model-based reinforcement learning. The paper proposes a new algorithm that leads to empirical improvement over certain data sets. All the reviewers agree that the paper provides useful theoretical insights into model-based reinforcement learning, and the experiments are also consistent with the theory.

**Award:**

No

---

### Decision · Program_Chairs · 2022-09-14

Accept